# Tuning apicobasal polarity and junctional recycling in the hemogenic endothelium orchestrates the morphodynamic complexity of emerging pre-hematopoietic stem cells

Léa Torcq[1,2], Sara Majello[1], Catherine Vivier[1], Anne A Schmidt[1]*[†]

[1]Department of Developmental and Stem Cell Biology, Institut Pasteur, Université Paris Cité, Paris, France; [2]Sorbonne Université, Paris, France

*For correspondence: anne.schmidt@pasteur.fr

[†]Lead contact

Competing interest: The authors declare that no competing interests exist.

**Abstract** Hematopoietic stem cells emerge in the embryo from an aortic-derived tissue called the hemogenic endothelium (HE). The HE appears to give birth to cells of different nature and fate but the molecular principles underlying this complexity are largely unknown. Here we show, in the zebrafish embryo, that two cell types emerge from the aortic floor with radically different morpho-dynamics. With the support of live imaging, we bring evidence suggesting that the mechanics underlying the two emergence types rely, or not, on apicobasal polarity establishment. While the first type is characterized by reinforcement of apicobasal polarity and maintenance of the apical/luminal membrane until release, the second type emerges via a dynamic process reminiscent of trans-endothelial migration. Interfering with Runx1 function suggests that the balance between the two emergence types depends on tuning apicobasal polarity at the level of the HE. In support of this and unexpectedly, we show that Pard3ba – one of the four Pard3 proteins expressed in the zebrafish – is sensitive to interference with Runx1 activity, in aortic endothelial cells. This supports the idea of a signaling cross talk controlling cell polarity and its associated features, between aortic and hemogenic cells. In addition, using new transgenic fish lines that express Junctional Adhesion Molecules and functional interference, we bring evidence for the essential role of ArhGEF11/PDZ-RhoGEF in controlling the HE-endothelial cell dynamic interface, including cell-cell intercalation, which is ultimately required for emergence completion. Overall, we highlight critical cellular and dynamic events of the endothelial-to-hematopoietic transition that support emergence complexity, with a potential impact on cell fate.

## eLife assessment

This **important** study presents a detailed characterization of two distinct cellular morphologies of haematopoietic stem cells undergoing endothelial to haematopoietic transition in zebrafish. It brings new information on how regulation of apico-basal polarity influences cellular behaviour, shape, and interaction with neighbouring cells. The evidence supporting the existence of these two distinct morphologies is **convincing**, using state-of-the-art confocal microscopy and image analysis of 2D-cartography.

**eLife digest** In mammals and other animals with backbones, the cells that will make up blood and immune cells are generated during a very narrow timeframe in embryonic development. These cells, called hematopoietic stem cells and progenitors (or HSPCs for short), emerge from tissue known as hemogenic endothelium that makes up the floor of early blood vessels.

For HPSCs to eventually specialise into different types of blood and immune cells, they require diverse migratory and homing properties that, ultimately, will determine the specific type of functions they exert. An important question for scientists studying the development of different blood and immune cell types is when this commitment to functional diversity is established. It could, for example, arise due to cells in the hemogenic endothelium having different origins. Alternatively, the signals that generate hemogenic endothelium cells could be responsible. It is also possible that both explanations are true, and that having different mechanisms involved ensures diversity in populations of HSPCs.

To investigate differences between the HSPCs emerging from the hemogenic endothelium, Torcq et al. studied zebrafish embryos that had been modified so that one of the proteins involved in sensing cell polarity – where the top and bottom of the cell are located – was fluorescent. Live imaging of the embryos showed that two types of cells, with striking differences in morphology, emerge from the hemogenic tissue. In addition, one cell type displays the same polarity as the other vessel cells, whereas the other does not. Torcq et al. also present evidence suggesting that the signals responsible for controlling this cell polarity are provided by surrounding blood vessel cells, supporting the idea of an interplay between the different cell types.

The finding that two different cell types emerge from the hemogenic endothelium, reveals a potential new source of diversity in HSPCs. Ultimately, this is expected to contribute to their functional complexity, resulting in both long-term stem cells that retain their full regenerative potential into adulthood and more specialized blood and immune cells.

## Introduction

Hematopoietic stem cells (HSCs) endowed with full regenerative potential in adult vertebrates are generated during a narrow-time window of few days during embryonic development. These cells, at the root of all blood and immune cells in the body, emerge from intra-embryonic aortic vessels and, more specifically, from a specialized type of vascular tissue called the hemogenic endothelium (HE, *Wu and Hirschi, 2021*). A series of seminal studies have evidenced the autonomous production of repopulating HSCs within an intraembryonic region called the Aorta-Gonad-Mesonephros (AGM), before their appearance in other hematopoietic organs (*Medvinsky et al., 1993*; *Müller et al., 1994*; *Garcia-Porrero et al., 1995*; *Medvinsky and Dzierzak, 1996*; *Cumano et al., 1996*). Thereafter, the endothelial origin of HSCs was evidenced (*Jaffredo et al., 1998*; *de Bruijn et al., 2002*; *North et al., 2002*). The direct visualization, in real-time, of the emergence of precursors of hematopoietic stem and progenitor cells (HSPCs) from the dorsal aorta, termed the Endothelial-to-Hematopoietic Transition (EHT), was finally achieved in vitro (*Eilken et al., 2009*), ex vivo from mouse sections (*Boisset et al., 2010*), and in vivo in the zebrafish embryo (*Bertrand et al., 2010*; *Kissa and Herbomel, 2010*), using live microscopy. This opened the way to a more detailed analysis of the characteristics of the HE at the transcriptional level, revealing its transient nature and its early hematopoietic commitment (*Swiers et al., 2013*).

The HE is characterized by inherent heterogeneity and is not only contributing to the formation of precursors to long-term HSCs (pre-HSCs) but also to more restricted progenitors (*Hadland and Yoshimoto, 2018*). These progenitors can be born from HE sub-types that are found in extra-embryonic source such as the yolk-sac – in which case it gives rise to erythro-myeloid progenitors that will sustain erythro-myelopoiesis until birth (*Frame et al., 2016*) – or both from extra and intra-embryonic sources (the yolk sac and the dorsal aorta) in which cases progenitors are biased, such as for example in the mouse, toward T- and B- innate-type lymphoid progenitors (*Yoshimoto et al., 2011*; *Yoshimoto et al., 2012*) or toward less restricted, multipotent progenitors (*Hadland and Yoshimoto, 2018*; *Dignum et al., 2021*). Importantly, while some of the non-HSC derived progenitors born during embryonic life only support the functions of the immune system during embryonic/ fetal

life, others can persist in the adult to exert tissue-resident functions, as has been mainly described in the mouse (*Ghosn et al., 2019*). However, studies in human embryos support the idea that developmental hematopoiesis is highly similar in mice and humans (*Ivanovs et al., 2011*; *Ivanovs et al., 2017*).

A key question regarding the issue of the capacity of the HE to give birth to cells endowed with different hematopoietic potential is whether this comes from HE cells of distinct origin (in which case the HE would be a mosaic of precursors with distinct potential) or if this results from extrinsic environmental cues that impose variability in the constantly evolving developmental context (for a discussion, see *Barone et al., 2022*).

To address this type of question, high-resolution experimental settings need to be developed, if possible at the single-cell resolution. Along this line, recent developments of transcriptomics, including single-cell RNAseq, spatially resolved in situ transcriptomic approaches, and in situ barcoding, have been invaluable (*Weijts et al., 2021*). Among those high-resolution approaches, cell fate mapping and lineage tracing approaches that include imaging technologies have been very instrumental, particularly when performed with the zebrafish model that reproduces many aspects of developmental hematopoiesis in higher vertebrates (*Orkin and Zon, 2008*). Recently, work performed in the zebrafish embryo has brought evidence for heterogeneity of hematopoietic stem cell precursors being born from the HE, in the ventral floor of the dorsal aorta and independently from HSCs, including a wave of transient T-lymphocytes (using temporally spatially resolved fate-mapping, *Tian et al., 2017*) and lympho-myeloid biased progenitors born from a myeloid-lymphoid biased Spi2+HE that appears to co-exist with an erythroid-biased HE (using single-cell RNA-sequencing, *Xia et al., 2023*).

While light starts to be shed on the molecular and signaling cues that appear to regulate HE subspecification and a continuum from arterial endothelium, HE maturation, and subsequent heterogeneity in HSPCs (*Zhu et al., 2020*), the essential molecular and cell biological properties that support HE functional plasticity remain to be determined.

Here, using the zebrafish embryo as a model, we complement our previous work describing essential molecular and mechanistic features of EHT cell emergence (*Lancino et al., 2018*). With the support of high-resolution live imaging and the generation of new transgenic fish lines that express a functional marker of cell polarity (Podocalyxin, a sialomucin of the CD34 family *Nielsen and McNagny, 2008*), we show that the HE is giving birth to two cell types with distinct cell polarity status; these cells also emerge with radically different morphodynamic characteristics, which raises the intriguing possibility of an incidence of EHT emergence complexity on downstream fate, after release from the aortic wall.

To substantiate the significance of apico-basal polarity control in the EHT, we investigate on the potential involvement of Pard3 proteins. These proteins, recruited by transmembrane receptors via their PSD-95/Dlg/ZO-1 (PDZ) domains (*Buckley and St Johnston, 2022*), are at the root of apicobasal polarity initiation and are essential for the maintenance of apical membrane functional properties (*Román-Fernández and Bryant, 2016*). Among the four Pard3 proteins expressed in the zebrafish (encoded by four different genes and non-including splicing variants), we describe the specific expression of Pard3ba, in the aorta, during the EHT time-window. We show that the expression of Pard3ba is sensitive to interference with the transcription factor Runx1 whose function is essential for EHT completion (*Kissa and Herbomel, 2010*) and for regulating HSC number (*Adamo et al., 2009*; *North et al., 2009*). In link with apico-basal polarity and to address its incidence on the biomechanics of the emergence of the two EHT cell types that we describe, we generated new fish lines that express Junctional Adhesion Molecules (JAMs) fused with eGFP (eGFP-JAMs). The JAMs belong to tight junction complexes (*Garrido-Urbani et al., 2014*) and recruit Pard3 via direct interaction with their carboxy-terminus (*Ebnet et al., 2003*), thereby allowing initiation of apico-basal polarity. With our eGFP-JAMs fish lines, we investigate on the function of ArhGEF11/PDZ-RhoGEF that was shown to be involved in disrupting junctional integrity in the context of the epithelial-to-mesenchymal transition as well as in the growth and migration of invasive cancer cells (more specifically one splicing variant of its c-terminus conserved in mammalian species, *Itoh et al., 2017*; *Lee et al., 2018*). We reveal the function of ArhGEF11/PDZ-RhoGEF in the dynamic interplay between HE/EHT cells and their endothelial neighbors, particularly in cell-cell intercalation which is essential for EHT completion and the proper sealing of the aortic floor.

## Results

### Apicobasal polarity determines emergence types

Our previous work describing the morphodynamic evolution of cells emerging from the aortic floor through the EHT (hereafter designated as 'EHT cells'), in the zebrafish embryo, revealed the unusual feature of a cell extruding from the plane of a tissue while maintaining its luminal/apical membrane until the very end of the release process, thus contributing to its peculiar crescent-shaped morphology (*Kissa and Herbomel, 2010*; *Lancino et al., 2018*, and see the cartoons in *Figure 1A*). However, to our knowledge, the polarity status of EHT cells has not been investigated so far and the maintenance of a bona fide apical domain has never been proven (with the luminal membrane enriched in apically targeted proteins and physically delimited by apical polarity factors and tight junction complexes *Rodriguez-Boulan et al., 2004*; *Buckley and St Johnston, 2022*). Importantly, the fate of this apical-luminal membrane, after the release, may lead to cells potentially endowed with specific functional features. For example, this membrane surface may be directly and fully exposed to the extracellular space or released in the cytoplasm of EHT cells for recycling and/or degradation, after emergence completion (for examples of different scenarios, see *Figure 1A*, and the legend for more details). Overall, this could lead to precursors of hematopoietic stem cells that, ultimately, may be differentially fated.

To address the polarity status of EHT cells, we raised transgenic fish lines that express endogenous Podocalyxin (Podxl2, *Herwig et al., 2011*). Podocalyxin was shown to take part in the formation of the preapical domain during polarization and in the regulation of its oriented organization, in tissue culture (*Meder et al., 2005*; *Bryant et al., 2014*). Its contribution to lumenization in vivo, in the mouse aorta, has been described and it involves negative charge repulsion induced by their glycosyl residues (*Strilić et al., 2009*; *Strilić et al., 2010*).

We first attempted to express transiently, in the vascular system, the full-length Podxl2 zebrafish protein fused to eGFP at its extreme N-terminus. We failed to detect the fusion protein at any time point relevant for observing easily the EHT process (a time window ranging from 48 to 72 hpf [hours post-fertilization]); therefore, we designed a N-ter truncated form deleted of the mucin domain and that retains amino-acids 341–587 fused to eGFP at its N-terminus (*Figure 1B* and see Materials and methods). Transient transgenesis revealed that this truncated version is detected and is targeted to the luminal membranes of EHT cells. We then raised two transgenic (*Tg*) fish lines that express the N-ter truncated form of Podxl2 fused to either eGFP (thereafter abbreviated eGFP-podxl2) and under the control of the Kdrl:Gal4 driver *Tg(Kdrl:Gal4;UAS:RFP; 4xNR:eGFP-podxl2)*, or to mKate2 and under the control of the Kdrl promoter *Tg(Kdrl:mKate2-podxl2)*. We observed that eGFP-podxl2 is enriched at the luminal side of crescent-shaped EHT undergoing cells (see *Figure 1C*, *Figure 1—figure supplement 1A* and *Figure 1—video 1* (z-stack at t=0) and their legends for the details of the luminal/apical membrane evolution through time). Thereafter, these cells will be referred to as EHT pol+ cells.

We also followed the cell after emergence and observed the evolution of the luminal/apical membrane appearing as internal pseudo-vacuoles. We illustrate the reduction of their volume via membrane retrieval and, ultimately, their remanence as an intracellular membrane compartment which we define as a post-EHT signature (*Figure 1—figure supplement 1A* and *Figure 1—video 2* [z-stack at t=80 min]); we propose a model for the intracellular evolution of the luminal/apical membrane (*Figure 1—figure supplement 1B*) which, unfortunately, cannot be traced after 2–3 hr post-emergence because of the apparent short half-life of eGFP-podxl2 and of the drop in activity of the Kdrl promoter. Of notice, the pseudo-vacuoles are reminiscent of the cystic formations observed in EHT cells in the mouse and visualized by electron-microscopy (*North et al., 1999*; *Marshall and Thrasher, 2001*) and also of the vacuolar structures recently described in EHT cells in avian embryos (*Sato et al., 2023*; for more details, see *Discussion*).

While imaging the EHT using the Podxl2 expressing lines that clearly delimitate the luminal membrane, we unambiguously identified a second type of emergence. This second cell type is primarily characterized by a round-to-ovoid morphology (cells never bend as crescent-shaped EHT pol+ cells, see *Figure 1D* and *Figure 1—video 3* for a time-lapse sequence). Importantly, these cells do not show any enrichment of eGFP-podxl2 at the luminal membrane and will be referred to as EHT pol- cells. EHT pol- cells were observed in all other *Tg* fish lines that we are routinely imaging, including the *Tg(Kdrl:Gal4;UAS:RFP)* parental line that was used for transgenesis, thus excluding the

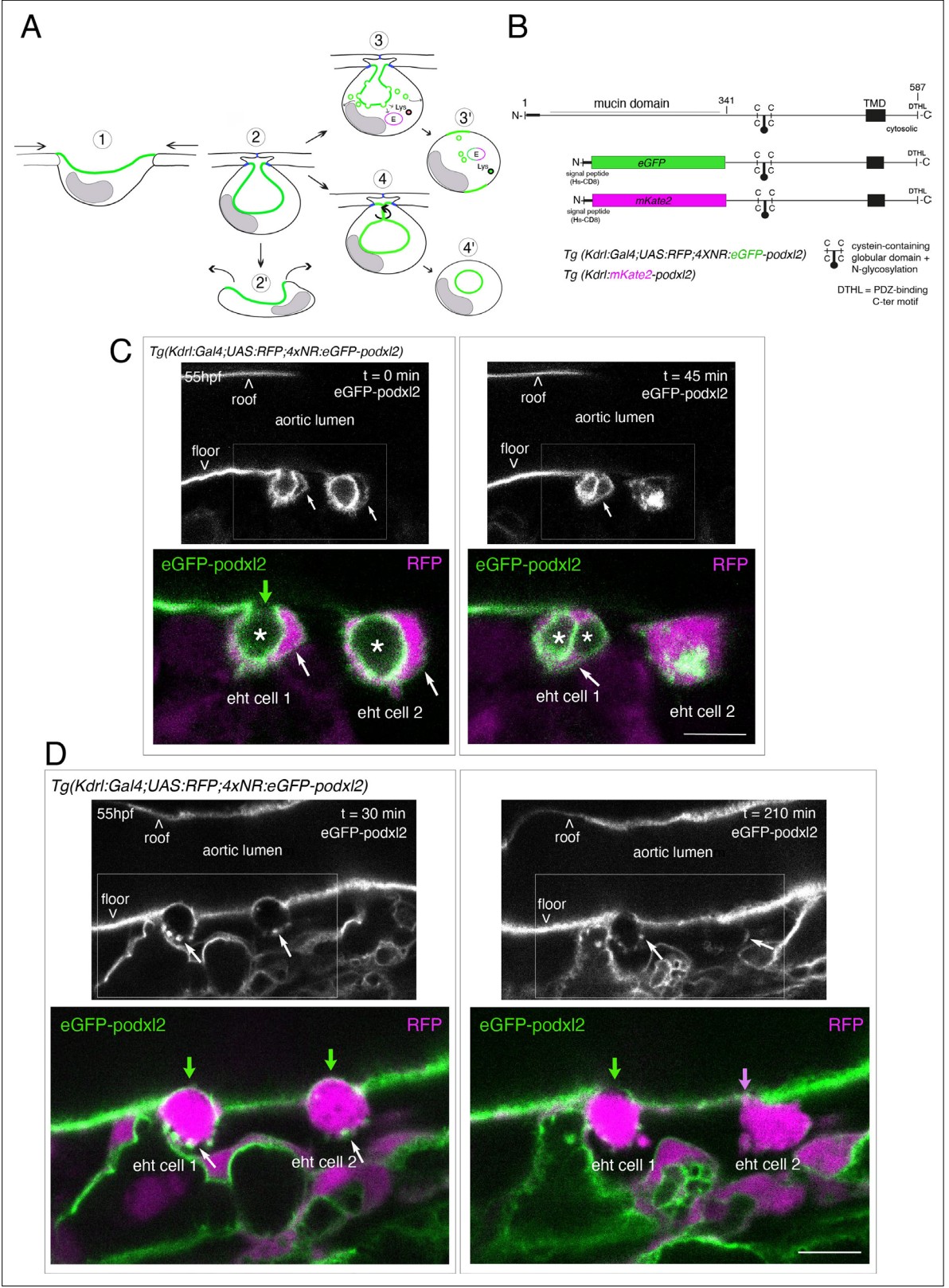

**Figure 1.** Tracing the evolution of the luminal membrane using the polarity marker Podocalyxin points at the biological significance of apicobasal polarity establishment in EHT cell emergence complexity. (**A**) Cartoons depicting the early and late steps of EHT cells emerging from the aortic floor (steps 1 and 2, as previously described in the zebrafish embryo, see *Kissa and Herbomel, 2010*; *Lancino et al., 2018*) and with hypothetical evolution of the luminal membrane (in green) before (3 and 4) and after the release 2', the cell detaches from the endothelial layer via junction downregulation

*Figure 1 continued on next page*

*Figure 1 continued*

leading to exposure of the luminal membrane with le extracellular milieu; 3', the luminal membrane is consumed via endocytic recycling (E) and/or lysosomal degradation (Lys) prior to detachment; 4', the luminal membrane in 4 is released inside the cell (twisted arrow) before detachment. Grey area = nucleus. (**B**) PodocalyxinL2 (Podxl2) construct designed to establish transgenic fish lines. Cartoons representing full length (top drawing) and deleted Podxl2 (amino-acid sequence 341–587) in which the mucin domain (serine/threonine-rich O-glycosylation domain) is replaced by either eGFP or mKate2. The tetracystein-containing globular domain (subjected to N-glycosylation) was kept as favoring apical membrane retention. TMD, transmembrane domain; DTHL, (C)-terminal peptidic motif involved in partnership with PDZ domain containing proteins. (**C, D**) EHT performing cells visualized using *Tg(Kdrl:Gal4;UAS:RFP;4xNR:eGFP-podxl2)* embryos and time-lapse sequences initiated at 55 hpf obtained with spinning disk confocal microscopy (imaging was performed at the boundary between the most downstream region of the AGM and the beginning of the caudal hematopoietic tissue). Top grey panels show the green, eGFP channels for eGFP-podxl2 only. Bottom panels show the merge between green and red (soluble RFP, in magenta) channels. Scale bars = 8 μm. (**C**) Single plane images of 2 EHT pol+ cells extracted from a time-lapse sequence at t=0 and t=45 min, with the right cell (eht cell 2) more advanced in the emergence process than the left one (eht cell1). Note the enrichment of eGFP-podxl2 at the luminal membrane (surrounding the cavity labeled with an asterisk) in comparison to the basal membrane (white arrow). Note also the evolution of the luminal membranes with time, with the aortic and eht cell 1 lumens still connecting at t=0 (green arrow), the apparent fragmentation of the cytosolic vacuole (2 asterisks for eht cell 1 at t=45 min) and the compaction of Podxl2-containing membranes for eht cell 2 at t=45 min. More details on the evolution of the connection between the aortic/eht cell lumens are shown in *Figure 1—figure supplement 1A*. (**D**) Single plane images of 2 EHT pol- cells extracted from a time-lapse sequence at t=30 min and t=210 min (see *Figure 1—video 3* for the full-time lapse sequence), with the right cell (eht cell 2) slightly more advanced in the emergence than the left one (eht cell 1, with the latest attachment point between the emerging cell and the aortic floor (pink arrow)). Note, in comparison with the cells in panel (**C**), the ovoid shapes of cells, the absence of enrichment of eGFP-podxl2 at luminal membranes (green arrows) and the accumulation of eGFP-podxl2 at basal membrane rounded protrusions (white arrows).

The online version of this article includes the following video and figure supplement(s) for figure 1:

**Figure supplement 1.** Evolution of the apical/luminal membrane throughout time.

**Figure supplement 2.** EHT pol+ and EHT pol- cells recover their respective morphology after mitosis.

**Figure supplement 3.** EHT pol+ and EHT pol- cells express eGFP driven by the CD41 promotor.

**Figure 1—video 1.** EHT pol+ cells at early timing.

https://elifesciences.org/articles/91429/figures#fig1video1

**Figure 1—video 2.** EHT pol+ cells at late timing.

https://elifesciences.org/articles/91429/figures#fig1video2

**Figure 1—video 3.** Emergence of EHT pol- cells.

https://elifesciences.org/articles/91429/figures#fig1video3

possibility that these cells result from an artefact due to the expression of a deleted form of Podxl2 and/or to its overexpression (see also for example our new *Tg(kdrl:eGFP-Jam3b)* fish line, Figure 5C and Figure 6B).

Finally, we have estimated that the ratio between EHT pol+and EHT pol- cells is of approximately 2/1, irrespective of the imaging time window and of the localization of emergence along the aortic antero-posterior axis (starting from the most anterior part of the AGM (at the limit between the balled and the elongated yolk) down to the caudal part of the aorta facing the CHT). We observed that both EHT pol+and EHT pol- cells divide during the emergence and remain with their respective morphological characteristics after completing abscission (*Figure 1—figure supplement 2*); hence they appear as pairs of cells that exit the aortic wall sequentially (as shown *Figure 1C and D*). We also observed that both EHT pol+and EHT pol- cells express reporters driven by the hematopoietic marker CD41 (*Figure 1—figure supplement 3*), which indicates that they are both endowed with hematopoietic potential.

Altogether, our results show that hematopoietic precursor cells emerging from the aortic floor do so with heterogeneity in their morphodynamic characteristics. They also suggest that the control of apico-basal polarity may be at the root of these specific emergence types.

## The immature HE is not polarized

EHT pol+ and EHT pol- cells appear to emerge from the hemogenic endothelium (HE), the latter constituting the virtually exclusive cell population of the aortic floor just prior to the initiation of the EHT time-window (around 28 hpf, see *Zhao et al., 2022*). In this context, we addressed the polarity status of HE cells and its evolution throughout the EHT (*Figure 2*). Surprisingly, confocal microscopy using our *eGFP-podxl2* and *mKate2-podxl2* expressing fish lines revealed that HE cells do not appear to be polarized, based on the absence of Podxl2 enrichment at luminal membranes, at the initiation

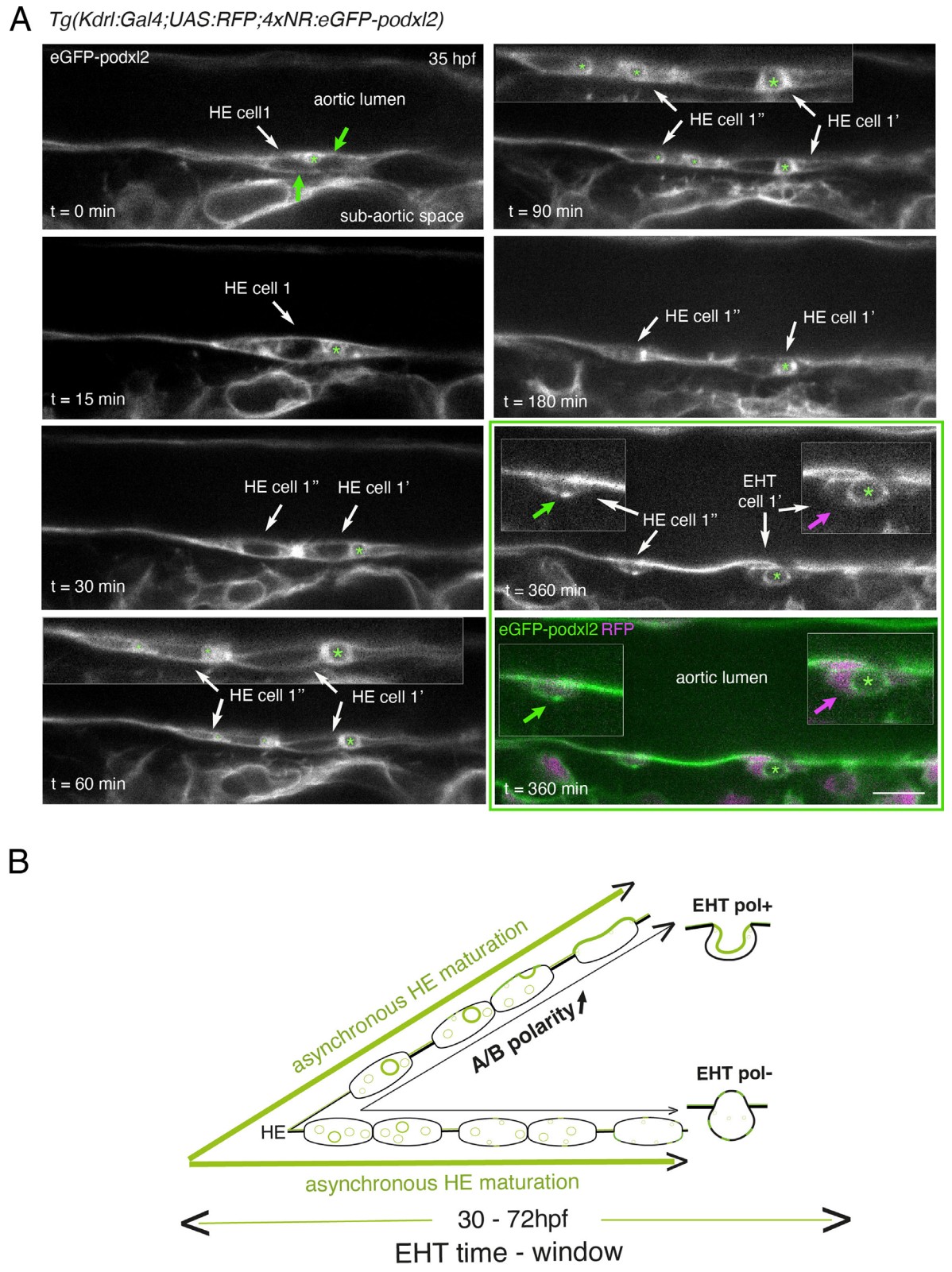

**Figure 2.** Immature HE is not polarized and controls membrane delivery of intra-cytosolic vesicular pools. (**A**) *Tg(Kdrl:Gal4;UAS:RFP;4xNR:eGFP-podxl2)* embryo imaged using spinning disk confocal microscopy. Black and white images show eGFP-podxl2 only. Images (single z-planes) were obtained from a time-lapse sequence (initiated at 35 hpf) lasting for 435 min (7.25 hr), with intervals of 15 min between each z-stack. Example of an HE cell with equal partitioning of eGFP-podxl2 between luminal and abluminal membranes (at t=0 min), with eGFP-podxl2 containing intra-cytosolic vesicles (one

*Figure 2 continued on next page*

*Figure 2 continued*

labeled with a green asterisk) and undergoing mitosis at t=30 min (HE cell 1' and HE cell 1" are daughter cells). Note the inheritance of the largest micropinocytic-like vacuole by HE cell 1' and its maintenance over time until EHT emergence initiation at t=180 min (green asterisk in 1.5 x magnified areas at t=60 and 90 min). At t=360 min (green box) EHT is proceeding and both fluorescence channels are shown; bottom panel: green (eGFP-podxl2), magenta (soluble RFP). The magenta arrow points at the basal side of the EHT pol+ cell (EHT cell 1', on the right) that does not contain any detectable eGFP-podxl2; on the contrary, eGFP-podxl2 is enriched at the luminal/apical membrane (note that exocytosis of the large vacuolar structure may have contributed to increase the surface of the apical/luminal membrane [the green asterisk is surrounded by the apical/luminal membrane of the EHT pol+ cell]). The green arrow points at the abluminal membrane of the EHT cell derived from HE cell 1" (EHT cell 1") and that contains eGFP-podxl2 (with no evidence of a significant expansion of a luminal/apical membrane); this indicates that this cell is more likely to be an EHT pol- cell that did not sort the vesicular cargo to the luminal/apical membrane. Scale bar = 10 μm. (**B**) Hypothetical model summarizing the evolution of HE cells involving the tuning of apicobasal polarity to lead to cells competent for giving birth to either EHT pol+ or EHT pol- cells (including the release of large vesicular macropinocytic-like vacuoles preferentially toward the luminal membrane of future EHT pol+ cells). The polarity status of HE cells is proposed to evolve asynchronously throughout the entire EHT time window, leading to place-to-place ability to give birth to EHT cells (emergence of EHT pol+ and EHT pol- cells are both observed until 72 hpf, see main text).

The online version of this article includes the following figure supplement(s) for figure 2:

**Figure supplement 1.** HE cells are not polarized at 30 hpf.

**Figure supplement 2.** Evolution of non-polarized HE cells throughout emergence.

of the EHT time-window and later (at approximately 28–30 hpf, *Figure 2—figure supplement 1* see also at 35 hpf *Figure 2A*, top left panel, green arrows and *Figure 2—figure supplement 2* and at 48–55 hpf *Figure 3—figure supplement 2A*). Interestingly, the cytoplasm of characteristic elongated HE cells located on the aortic floor is filled with more-or-less large membrane vesicles that carry eGFP-podxl2 (the largest vesicles reaching approximately 30 μm in diameter). This suggests that HE cells contain a reservoir of eGFP-podxl2 membranes that may be subjected to exocytose; as such, HE cells may be comparable to endothelial cells organizing a vascular lumen and that have been proposed to exocytose large intracellular macropinocytic-like vacuoles when cultured in 3D extracellular matrices (*Bayless and Davis, 2002*; *Davis et al., 2002*) or, in vivo, in the zebrafish model (*Kamei et al., 2006*; *Lagendijk et al., 2014*). This finding is unexpected since HE cells are assumed to possess aortic cell characteristics (i.e exhibit an apicobasal polarity) as they are supposedly integrated in the aortic wall contemporarily to aortic precursors (*Jin et al., 2005*) and may have been taking part in the lumenization of the ancestral vascular cord, a process that takes place around 18–20 hpf. Consequently, loss of apicobasal polarity features of HE cells at 28–30 hpf may be part of the programme that initiates the EHT process.

Although technically difficult for long hours (because of important variations in the volume of the balled yolk that trigger drifting of embryos), we have been able to follow over time non-polarized HE cells and to visualize the evolution of their vesicular content, starting at the initiation of the EHT time-window (around 35 hpf, see *Figure 2* and *Figure 2—figure supplement 2*). Interestingly, we could follow a dividing HE cell for which the vesicular content labeled with eGFP-podxl2 appeared to partition unequally between daughter cells (*Figure 2A*, t=30 min, HE cell 1' inherits the largest macropinocytic-like vacuole [green asterisk] and emerges unambiguously as an EHT pol+ cell [t=360 min, EHT cell 1'], with eGPF-podxl2 enriched at the apical/luminal membrane [surrounding the green asterisk] and virtually undetected at the basal membrane [magenta arrow]). This suggests that asymmetric inheritance of cytosolic vesicles containing apical proteins may contribute, presumably after delivery to the luminal membrane, to specify the apical membrane of EHT pol+ cells.

Altogether, these results support the idea that the HE, at the initiation of the EHT time-window, is not polarized. Subsequently, HE cells establish – or not – apical and basal membrane domains, which characterizes EHT pol+or EHT pol- cell types, respectively (see our hypothetical model *Figure 2B*). In the case of EHT pol+ cells and while emergence is proceeding, apicobasal polarity is maintained (if not reinforced) until the release.

## Interfering with Runx1 function alters HE maturation, the EHT progression, and the balance between EHT cell types

To provide functional support to our findings and hypotheses on apico-basal polarity control in the gradual maturation of the HE throughout the EHT time-window, we thought of interfering with the

activity of the transcription factor Runx1 whose expression is sensitive to fluid sheer stress (*Adamo et al., 2009*).

To interfere with Runx1 function, we generated a *Tg* fish line that expresses a truncated form of Runx1 (dt-runx1; fish line *Tg(Kdrl:Gal4;UAS:RFP;4xNR:dt-runx1-eGFP)* thereafter abbreviated *Tg(dt-runx1)*) deleted from its transactivation domain and carboxy-terminus but retaining its DNA-binding domain (the Runt domain aa 55–183, *Kataoka et al., 2000*; *Burns et al., 2002*; *Kalev-Zylinska et al., 2002*, see *Figure 3C*). Importantly, in this *Tg* fish line, dt-runx1 expression is restricted to the vascular system, hence excluding expression in the brain region that express Runx1 endogenously (see the aforementioned articles) and preventing biases owing to potential interference with neuronal functions. In addition, owing to the expression of eGFP concomitantly to dt-runx1 (the C-terminal eGFP is cleaved from dt-runx1 via a T2a site for endopeptidase, see cartoons at the top of panel C), this fish line allows for the easy selection of embryos for imaging and for phenotypic analysis. In preliminary experiments aimed at addressing the localization of dt-runx1 as well as its stability, we expressed it transiently and measured its proper targeting to the nucleus (the construct also contains a double HA (2xHA) epitope for immuno-detection, see *Figure 3C*).

To characterize further the *Tg(dt-runx1)* fish line, we addressed the incidence of dt-Runx1 expression on HE and EHT cell populations during the EHT time-window as well as later on, at 5 days post-fertilization (5dpf) in the thymus, as a readout of hematopoiesis efficiency. We made the following observations:

Firstly, using 48–55 hpf *Tg(dt-runx1)* embryos and live imaging we observed, in comparison to controls, a significant accumulation of HE cells (see *Figure 3A*, top and bottom panels (HE, arrow-heads), see also the quantifications *Figure 3B*, top graph, median value of 17 cells for controls and 25 cells for mutants). In the mutant condition, the bulk of HE cells also include uncharacterized EHT cells (*Figure 3A*, middle panel, blue arrows); hypothetically, these may be EHT cells at an early phase of their emergence, including cells that may have evolved as EHT pol- cells in unperturbed conditions, owing to the absence of luminal membrane invagination. We also observed the unusual tendency for the accumulation of EHT pol+ cells (*Figure 3A*, green and pink arrows), with a maximum value of 3 cells per embryo for controls that reached the unusual - never observed in control conditions -, values of 11, 9, and 8 cells for 3 mutants out of 7 (*Figure 3B*, middle graph). In addition to the increase in the proportion of EHT pol+ cells in the ventral side of the aorta (*Figure 3B*, right panel, with a value of 4% for controls and reaching 7% for mutants, consistently with the increase in the absolute count per embryo), we also observe an increase in the proportion of EHT pol+ cells in lateral sides of the dorsal aorta (*Figure 3A*, pink arrows and *Figure 3B*, right graph, with a value of 4% for controls and reaching 9% for mutants). Finally, we did not observe any significant modification in the number of EHT pol- cells (same median values for both controls and mutants, the values oscillating between 0 and 1 cell per embryo, with only one example of an embryo exhibiting two apparent EHT pol- cells [*Figure 3B*, bottom graph]).

Secondly, using 48–55 hpf *Tg(dt-runx1)* embryos and controls *Tg(Kdrl:eGFP)* we confirm, by whole mount in situ hybridization performed with an RNAscope probe specific for myb (*Figure 3—figure supplement 1*), that the cells that accumulate in the aortic floor of dt-Runx1 mutants are indeed of hematopoietic nature (see the cells with RNAscope spots in magenta or marked by blue arrows, *Figure 3—figure supplement 1A*, zoomed areas). Unfortunately, chemical fixation impaired the maintenance of EHT pol+and EHT pol-specific cell shape, preventing us from classifying emerging cell populations. However, cell counts after segmentation using the Imaris software confirmed, based on cell localization, the specific increase of hemogenic/EHT cells in aortic floors of mutants median values of 34 and 16.5 for mutants and controls, respectively, *Figure 3—figure supplement 1B* (right panel) while no difference was measured for endothelial roof cells between mutants and controls (left panel). Quantification of myb-positive cells confirmed the increase of hemogenic cells in the aortic floors of mutants in comparison to controls, albeit with slightly smaller numbers (median values of 20.5 and 8.5 in mutants and controls, respectively *Figure 3—figure supplement 1C*, middle panel), most probably because myb becomes detectable only when the EHT process has initiated. In comparison to floor cells, very few roof cells express myb (median values of 2.0 and 1.5 in mutants and controls, respectively *Figure 3—figure supplement 1C*, left panel consistently with the idea that the aortic roof is not hemogenic *Zhao et al., 2022*). Importantly, the increase of pre-hematopoietic cells in the aortic floor in the mutants correlates with a decrease in hematopoietic cells lodging in the sub-aortic region,

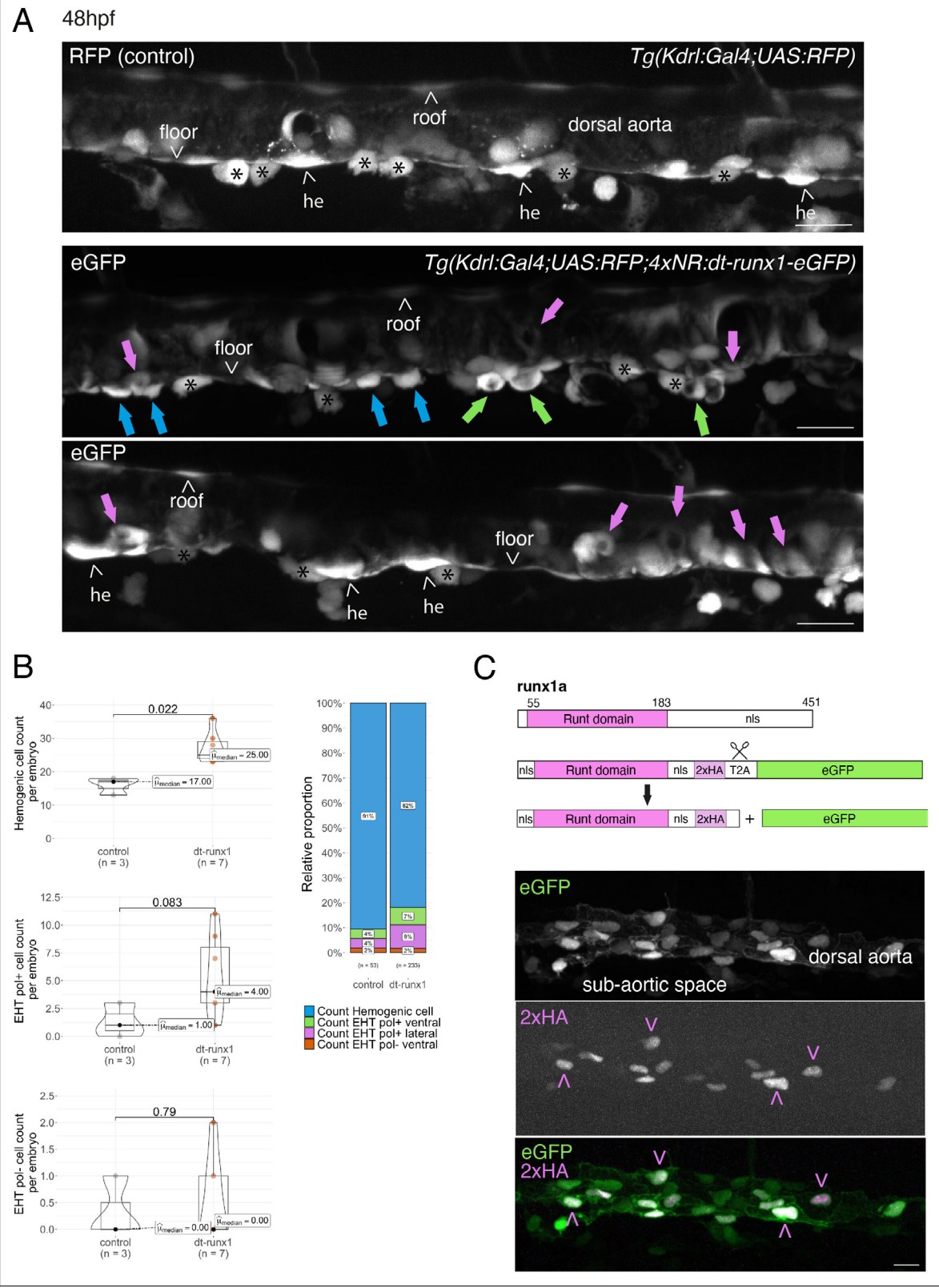

**Figure 3.** Interference with Runx1 function alters emergence efficiency and induces a bias toward EHT pol+ cells. (**A**) *Tg(kdrl:Gal4;UAS:RFP;4xNR:dt-runx1-eGFP)* mutant embryos imaged using spinning disk confocal microscopy and analyzed in the AGM/trunk region. Z-projections of the dorsal aorta obtained from 52 to 55 hpf embryos. Top panel: fluorescence from the red channel is shown for the *Tg(Kdrl:Gal4;UAS:RFP)* control. Bottom panels: fluorescence in the green channel only is shown for the mutants (eGFP, released from the dt-runx1-eGFP cleavage). The black asterisks point at emerged

*Figure 3 continued on next page*

*Figure 3 continued*

cells that are in close contact with the aortic floor. Green arrows: EHT pol+ cells on aortic floor; magenta arrows: EHT pol+ cells in the lateral aortic wall; blue arrows: uncharacterized emerging cells; he: hemogenic cells. Scale bars = 20 µm. (B) Quantitative analysis of the dt-runx1 mutant phenotype. The analysis was carried out on *Tg(Kdrl:Gal4;UAS:RFP)* control embryos (n=3) and on Tg(*kdrl:Gal4;UAS:RFP;4xNR:dt-runx1-eGFP*) mutant embryos (n=7). Left top: hemogenic cell count (comprising hemogenic cells, labeled by 'he' on panel (A) as well as uncharacterized emerging cells, labeled by blue arrows on panel (A)). Left center: EHT pol+ cell count (sum of EHT pol+ cells emerging ventrally and laterally, respectively labeled by green and magenta arrows on panel (A)). Left bottom: EHT pol- cell count not shown on panel (A). Right: percentage of cell types in control and dt-runx1 embryos (proportions relative to total number of hemogenic and EHT undergoing cells). Statistical comparisons have been performed using two-sided unpaired Wilcoxon tests, all p-values are displayed. (C) Top: cartoons representing the zebrafish full-length runx1a amino acid sequence and the dt-runx1 mutant deleted from the trans-activation domain and of the C-terminus note that the construct encodes for a C-terminal fusion with eGFP that is released upon expression via a cleavable T2A peptide (introduced between the 2xHA tag and the N-terminus of eGFP, to prevent from potential steric hindrance). nls, nuclear localization signal. Bottom: image of an anti-HA tag immunofluorescence obtained after z-projection of the dorsal aorta of a 50 hpf *Tg(dt-runx1)* embryo. Note the localization of the 2xHA-tagged dt-runx1 protein in nuclei (some of them are pointed by red arrowheads) and of eGFP in nuclei and the cytosol of aortic cells. Scale bar = 25 µm. Raw images (z-stacks) for this figure (dt-runx1 phenotype analysis) are available at https://doi.org/10.5281/zenodo.10932245.

The online version of this article includes the following video and figure supplement(s) for figure 3:

**Figure supplement 1.** Expression of dt-runx1 triggers the accumulation of myb + cells in the aortic floor.

**Figure supplement 2.** Phenotypic analysis of dt-runx1 expressing mutants: evidence for apicobasal polarity of hemogenic cells.

**Figure supplement 3.** Phenotypic analysis of dt-runx1 expressing mutants: expansion of the thymus.

**Figure 3—video 1.** Cellular expansion in the thymus of a dt-runx1 embryo compared to control.

https://elifesciences.org/articles/91429/figures#fig3video1

reinforcing the idea of impairment in EHT cell emergence (*Figure 3—figure supplement 1C*, right panel). Finally, quantification of the number of myb RNAscope spots per segmented hemogenic cells highlights the important increase in expression by hemogenic cells in the floor of dt-Runx1 mutants, in comparison to controls (with median values of 4 and 0, respectively) (*Figure 3—figure supplement 1D*, right panel). In comparison, virtually none of the roof cells contain more than 2 spots (*Figure 3—figure supplement 1D*, with 2/162 and 1/220 cells for the mutants and controls, respectively, with median values of 0 for both conditions). Altogether, these last results confirm that pre-hematopoietic cells accumulate in the aortic floor of dt-Runx1 mutants; the very substantial increase in the detection of myb mRNAs suggests an increase in the residential time in the aortic floor, which reinforces the idea that these cells are impaired in their release from the aortic floor.

Thirdly, between 30 and 55 hpf, using live confocal microscopy and contrarily to what we observed with our fish lines expressing either eGFP-podxl2 (*Figure 2* and *Figure 2—figure supplements 1 and 2*) or mKate2-podxl2 (*Figure 3—figure supplement 2A*) in which HE cells appear to be non-polarized, embryos obtained from outcrossing *Tg(dt-runx1) and Tg(Kdrl:mKate2-podxl2)* fishes appear to contain polarized HE cells, based on enrichment of the polarity marker at luminal membranes (*Figure 3—figure supplement 2B*).

Fourthly and occasionally, in live embryos, we observe - for EHT cells exhibiting invagination of the luminal membrane - scattered cytosolic and sub-plasmalemmal pools of Podxl2-containing membranes and, consistently, the apparent decrease of Podxl2 enrichment at the apical/luminal membrane (see *Figure 3—figure supplement 2C* and compare with *Figure 1C*). In addition and occasionally as well, we observe the reversion of apparent EHT pol+ cells into apparent EHT pol- cells (data not shown). These last two observations suggest that perturbing the control of apicobasal polarity in the HE, in the context of the dt-Runx1 mutant, alters emergence morphodynamics, hypothetically and possibly more specifically in the case of EHT pol- cells whose biomechanical features appear to require turning down the establishment of apical and basal membrane domains.

Fifthly, hematopoiesis is affected far downstream of emergence, as attested by the significant increase, at 5 dpf, in the number of hematopoietic cells in the thymus of dt-Runx1 mutants, in comparison to control siblings (see *Figure 3—figure supplement 3A* for images after cell segmentation with the Imaris software accompanied by *Figure 3—video 1* and *Figure 3—figure supplement 3B* for quantifications showing significant increase in thymus volume [left panel] owed to the increase in cell number [middle panel] rather than in cell volume [right panel]).

Altogether, the results obtained upon expression of dt-runx1 show impairment of hematopoiesis and suggest that, for both EHT cell types, the progression throughout EHT is perturbed, and

so until the release. The accumulation of morphologically characterized EHT pol+like cells that we observe may result from the sustained apicobasal polarity of the HE at early and later time points. Conversely, characterized EHT pol- cells did not accumulate, although we observed cells of uncharacterized morphology that could correspond to EHT pol- cells that did not evolve properly. In addition, our results suggest that precursors of HE cells are polarized, as is expected to be the case for non-hemogenic aortic cells, and that Runx1 is involved in controlling the molecular events that are tuning apico-basal polarity, starting at the initiation of the EHT time-window.

## Interfering with Runx1 activity unravels its function in the control of Pard3ba expression and highlights heterogeneous spatial distribution of Pard3ba mRNAs along the aortic axis

We then explored the potential involvement of proteins of the Pard3 family during the EHT. We anticipated that interfering directly with Pard3 proteins would hamper the development of the aorta (in addition to other functions of polarized tissues that are essential for embryonic development) and rather searched for correlative evidence for the differential expression of Pard3 gene products in the hemogenic endothelium, EHT cells and the aortic endothelium, both in wild type and dt-Runx1 expressing embryos. To achieve this task, we analyzed the expression of mRNAs encoding for the four Pard3 proteins expressed in the zebrafish (*Figure 4A*) by combining qRT-PCR on FACS-sorted endothelial cells and whole-mount in situ hybridization at the cellular resolution, using RNAscope. All four Pard3 proteins are composed of three PDZ domains and two conserved regions upstream and downstream of these domains (CR regions, *Figure 4A*).

We first sought to measure expression of these different gene products in the vascular system of 48 hpf embryos and, more restrictively, in the trunk region. To do so, we dissociated and FACS-sorted cells of either whole embryos, or dissected trunks, using the *Tg(Kdrl:Gal4; UAS:RFP)* fish line in which soluble RFP is expressed in endothelial tissues (*Figure 4—figure supplement 1A*). RFP+ vascular cells were found the be slightly enriched in the trunk region, in comparison to whole embryos (*Figure 4—figure supplement 1A* see the table with 2.10 vs 0.95% of RFP+ cells in the trunk and the whole embryo, respectively). Using qRT-PCR on the FACS-sorted cells, we found that the hematopoietic markers myb and runx1 were heavily detected in RFP+ cell extracts in comparison to RFP-cells (*Figure 4—figure supplement 1B*); this suggests that part of the isolated RFP+ cells contain hemogenic and hematopoietic cells (at 48 hpf, it is expected that some newly born hematopoietic cells are retained in the sub-aortic space). When we measured the expression levels of the four Pard3, we observed, in comparison to control RFP- cells, the enrichment of Pard3aa and Pard3ba mRNAs in extracts from RFP+ cells isolated from whole embryos, and the opposite for Pard3ab and Pard3bb (*Figure 4—figure supplement 1C*). These differences were conserved with extracts from RFP+ cells isolated from trunk regions, except for Pard3ba for which important variations in mRNA quantities were obtained, blurring conclusions (we shall come back to this variability beneath). These results suggest that, at least based on the vascular system of whole embryos, Pard3aa and Pard3ba exert functions in endothelial cells, with the possibility of having more specific functions in aortic and HE cells. To discriminate between cell types expressing Pard3aa and Pard3ba (and in particular vascular cell types), we used RNAscope on whole mount *Tg(Kdrl:eGFP)* embryos, at 48 hpf. Confocal images focused on trunk regions show that Pard3aa is strongly expressed in cells of the spinal cord (potentially neuronal cells and/or radial glia), as well as in the notochord (*Figure 4—figure supplement 2A*). Interestingly, Pard3ba mRNAs are expressed in more restricted areas, in majority in the spinal cord, albeit to a much lesser extent than for Pard3aa as well as in the region of the pronephric duct and the gut epithelium (*Figure 4—figure supplement 2B*). By curiosity and because it appears to be the most highly expressed in the whole embryo (and hence may be ubiquitously expressed, including in the trunk region and in accordance to the Daniocell resource at https://daniocell.nichd.nih.gov/, *Farrell et al., 2018*; *Sur et al., 2023*), we also investigated on the expression of Pard3ab mRNAs (*Figure 4—figure supplement 2C*). We confirmed its relatively ubiquitous expression in comparison to Pard3aa and Pard3ba, with a massive expression in the spinal cord. To visualize more precisely expression levels in the vascular system in the trunk (more specifically in the aorta and the underlying vein), we zoomed in this region (*Figure 4—figure supplement 2D*) and observed a homogeneous localization of Pard3aa and Pard3ab mRNAs as well as, strikingly in the case of Pard3ba, a quite heterogeneous localization of mRNAs unequally concentrated along the aortic axis (*Figure 4—figure supplement*

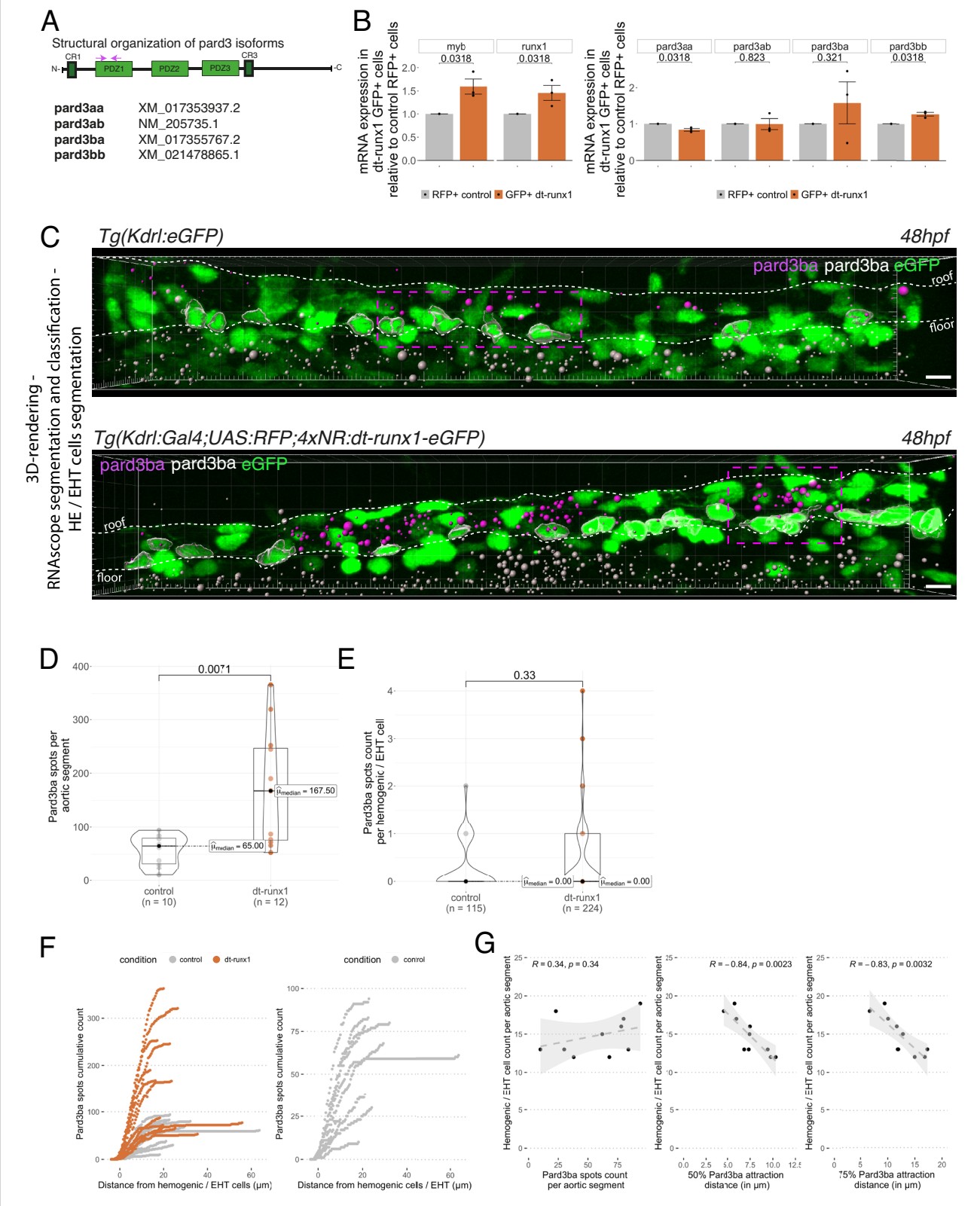

**Figure 4.** Pard3ba expression is highly sensitive to interference with Runx1 activity. (**A**) Pard3aa, ab, ba, and bb zebrafish gene products, with their accession numbers. The cartoon represents the common structures of the four proteins, with the three sequential PDZ domains (PDZ1-3), and the CR1 and CR3 conserved regions involved in oligomerization and atypical protein kinase C (aPKC) binding, respectively. The magenta arrows show the positioning of qRT-PCR primers, located in the PDZ1 domain (see Materials and ethods for primer sequences). (**B**) qRT-PCR analysis of genes expression

*Figure 4 continued on next page*

*Figure 4 continued*

levels of (left) the hematopoietic markers myb and runx1, (right) the four Pard3 mRNAs (encoding for Pard3aa, ab, ba, and bb) in cell populations isolated from FACS-sorted trunk vascular cells of 48–50 hpf control and mutant embryos (see *Figure 4—figure supplement 1*, *Figure 4—figure supplement 2* and Materials and methods for cell isolation procedures). Graphs show the measured mean fold changes relative to the expression of ef1α and to the expression in RFP+ control cells. Statistical tests: two-sided unpaired two samples Wilcoxon test, all p-values are displayed. Analysis was carried out on n=3 for control and mutant conditions. (**C**) Representatives images (Imaris 3D-rendering) of RNAscope in situ hybridizations for Pard3ba in 48–50hpf *Tg(Kdrl:eGFP)* control embryos and *Tg(Kdrl:Gal4;UAS:RFP;4xNR:dt-runx1-eGFP)* mutant embryos. The aorta is outlined with the white dashed lines. The RNAscope signal was segmented into spots and classified based on its localization: in aortic endothelial and hemogenic cells (magenta spots) or in extra-aortic tissues (grey spots, essentially residing in the sub-aortic space). Magenta dashed boxes delineate examples of mRNA Enriched Regions (MERs) in control and mutant embryos. Scale bars = 10 μm. (**D**) Pard3ba spots count per aortic segment. (**E**) Pard3ba spots count per hemogenic/EHT cells. (**F**) Cumulative Pard3ba spots count relative to hemogenic/EHT cell distance, averaged for each aortic segment, in control embryos (grey) and dt-runx1 mutant embryos (orange). (**G**) HE/EHT cell count relative to number of spots, all expressed per aortic segment (left), 50% Pard3ba spots attraction distance (in μm) (middle) or 75% Pard3ba spots attraction distance (in μm) (right). 50%/75% spots attraction distance corresponds to the distance from hemogenic/EHT cells within which 50%/75% of Pard3ba spots are located. Grey dash lines correspond to regression lines and light grey background correspond to the confidence interval (95%) of the regression model. Linear correlation coefficient (R, Pearson correlation) is displayed for each condition. For (**D** and **E**), statistical tests: two-sided unpaired two samples Wilcoxon test, all p-values are displayed. Analysis was carried out on n=5 control embryos and n=6 mutant embryos, 2 aortic segments per embryos. For (**G**) p-values were calculated using a t-test. Raw images (z-stacks) for this figure and *Figure 4—figure supplement 3* (Pard3 mRNA expression in control and mutant conditions) are available at https://doi.org/10.5281/zenodo.10937428.

The online version of this article includes the following figure supplement(s) for figure 4:

**Figure supplement 1.** Expression levels of Pard3 mRNAs in FACS-sorted endothelial cells.

**Figure supplement 2.** Localization of Pard3 mRNAs using RNAscope.

**Figure supplement 3.** Expression of Pard3ba is upregulated by dt-Runx1.

**Figure supplement 4.** Expression of Pard3aa and Pard3ab are insensitive to dt-Runx1.

*2D* middle panel and see also *Figure 4C*, top panel; these will be referred thereon as Pard3ba mRNA enriched regions (MERs)). Of notice, in the whole aorta, we observed more RNAscope signals for Pard3ab in comparison to Pard3aa and Pard3ba, including in some HE/EHT cells (*Figure 4—figure supplement 2D*, with positive HE/EHT cells indicated by blue arrows). This indicates that Pard3ab, consistently with the absence of variation upon dt-Runx1 expression (see beneath), is constitutively expressed in some HE/EHT cells (this may be also the case for Pard3aa but this should be taken with caution since its mRNAs are more in the limit of detection). We attempted to detect Pard3 proteins expressed in the vascular system, in the trunk region of the embryo, with available antibodies raised against other species and failed, possibly because of inter-species recognition limitation.

We then analyzed Pard3 mRNA levels in the context of dt-Runx1 expression. As we did for *Tg(Kdrl:Gal4;UAS:RFP)* embryos, we isolated by FACS GFP+ and double positive GFP+/RFP + cells dissociated from dissected trunks of *Tg(dt-Runx1)* 48 hpf embryos. For controls, we also isolated RFP + cells from dissected *Tg(Kdrl:Gal4;UAS:RFP)* embryos (see Materials and methods and *Figure 4—figure supplement 3A*). qRT-PCR on isolated eGFP+ cells that express dt-Runx1 revealed an increase of approximately 50% of the myb and runx1 mRNAs, in comparison to RFP+ control cells (*Figure 4B*, left panel), which is consistent with the accumulation of hemogenic/EHT cells that we observe in *Figure 3* and in *Figure 3—figure supplement 1*. Interestingly, we obtained very little to no variation in the expression levels of Pard3aa, ab and bb mRNAs in dt-Runx1 expressing eGFP+ cells in comparison to control RFP + cells but, again, a high variability in the case of Pard3ba mRNAs (*Figure 4B*, right panel). Using RNAscope, the non-homogeneous localization of Pard3ba mRNAs observed in controls, along the aortic axis, was not only conserved but became even more obvious to the eye for dt-Runx1 mutants, owing to the increase in the density of RNAscope spots (*Figure 4C*, compare top and bottom panels). Confocal images on several controls and dt-Runx1 mutants highlight the heterogeneity in Pard3ba mRNA detection as attested by the variability in the localization and number of RNAscope spots (compare in particular the five mutants shown in *Figure 4—figure supplement 3*, right panels). Quantification of RNAscope spots per aortic segment and per hemogenic/EHT cell using Imaris revealed an increase of almost 300% for the former when comparing controls and mutants (median values of 65.00 and 167.50 for controls and dt-Runx1 mutants, respectively, *Figure 4D*) and that virtually no Pard3ba mRNA was detected in the latter (median values of 0.00 spots per hemogenic/EHT cell for both controls and dt-Runx1 mutants, *Figure 4E*). A precise representation showing

the hemogenic/EHT cell count with RNAscope spots ranging from 0 to 4 reveals, for mutants in comparison to controls, a significant increase in the number of hemogenic/EHT cells not expressing Pard3ba (*Figure 4—figure supplement 3C* first panel from the left), as well as a slight tendency for an increase in the number of cells with 1–4 Pard3ba spots (*Figure 4—figure supplement 3C* second to fifth panels), with more significance for the population with 2 spots (middle panel). Overall, in either control or mutant conditions, we detected the presence of RNAscope spots in very few hemogenic/ EHT cells (15/115 for controls and 21/224 for dt-Runx1 mutants, with a maximum for 2 spots for controls and 4 spots for dt-Runx1 mutants, see *Figure 4E* and *Figure 4—figure supplement 3C*).

Again, we obtained a large variability in the number of Pard3ba RNAscope spots per aortic segment (the number of spots per aortic segment ranged from 10 to 94 for controls and from 53 to 366 for the mutants, see *Figure 4D*); this mirrors what was obtained by qRT-PCR, particularly for dt-Runx1 mutants (*Figure 4B*, right panel). This is not the case for Pard3aa and Pard3ab (see RNAscope images and quantifications *Figure 4—figure supplement 4*), which is also consistent with qRT-PCR results (*Figure 4B*, right panel). Importantly, expression of dt-Runx1 neither influenced the expression of Pard3aa, nor of Pard3ab (*Figure 4—figure supplement 4*).

Finally and importantly, in wild-type embryos, the Pard3ba mRNAs are preferentially localized in the direct vicinity of hemogenic/EHT cells (*Figure 4F and G*). Indeed, more than 50% of Pard3ba spots are localized within 5–10 μm of hemogenic/EHT cells (*Figure 4G*, middle panel), and more than 75% of spots are localized within a 5–20 μm distance (*Figure 4G* right panel). Interestingly, the number of hemogenic/EHT cells does not correlate with the absolute number of spots (*Figure 4F*, left panel, R=0.34), but rather it strongly correlates with the spatial densification of spots around hemogenic/ EHT cells: the shorter the distance of the majority of spots are of hemogenic/EHT cells (*Figure 4F*, middle and right panel, representing 50% and 75% of spots, respectively), the higher the hemo-genic/EHT cell count (R=–0.84 and R=–0.83, respectively). This shows that AECs expressing Pard3ba mRNAs are mostly in direct contact with cells committed to undergo EHT and suggests a signalling crosstalk occurring between them. This densification of spots in MERs is reinforced upon expression of dt-Runx1 (*Figure 4F*, left panel): we found that virtually all Pard3ba spots are localized within a 10–15 μm distance from hemogenic/EHT cells.

Altogether, these results unveil the Runx1-dependent control of Pard3ba specifically, during the EHT. In the aorta, AECs appear to be the main cell type that expresses this Pard3 protein and, unex-pectedly, expression in these cells is sensitive to interference with our dt-Runx1 mutant. One plausible explanation is that AECs respond to Runx1-mediated signaling taking place in HE/EHT cells (see Discussion for other possibilities and further details). This suggests that a specific signalling axis is taking place between these different cell types that would control the expression of Pard3ba mRNAs and, consequently, downstream apico-basal polarity associated features carried out by the Pard3ba protein (example: the recycling mode of junctional complexes between EHT pol+ cells and endothe-lial neighbors, in comparison to EHT pol- cells, see beneath and Discussion). This potential intercel-lular signaling is supported by the proximity of Pard3ba mRNAs expressed in AECs to HE/EHT cells in MERs. These results also strengthen the importance of the interplay between HE/EHT cells and their direct endothelial neighbors (this interplay has been highlighted in *Lancino et al., 2018* see also beneath).

## Junctional recycling between EHT cell types and endothelial neighbors is differentially controlled

To investigate the functional links between apicobasal polarity and the peculiarities of EHT pol+ and EHT pol- emergence processes, we sought to follow the dynamics of Junctional Adhesion Mole-cules (JAMs) that belong to tight junction complexes (*Garrido-Urbani et al., 2014*). During apico-basal polarity establishment in epithelial and endothelial tissues, these molecules recruit the Pard3/ aPKC complex, via a PDZ-binding peptide located in their extreme carboxy-terminus (*Figure 5A*, left cartoons and *Itoh et al., 2001*; *Ebnet et al., 2001*; *Ebnet et al., 2018*). In addition, JAMs are expressed in the vascular system and, in relation to this work, JAM3b is expressed in the aorta of the zebrafish embryo and promotes endothelial specification and hematopoietic development (*Kobayashi et al., 2020*).

We envisaged a scenario whereby EHT pol+ cells, whose longitudinal interface with endothelial neighbors shrinks along the X axis of the X, Y 2D-plane, may have less mobile junctional pools than

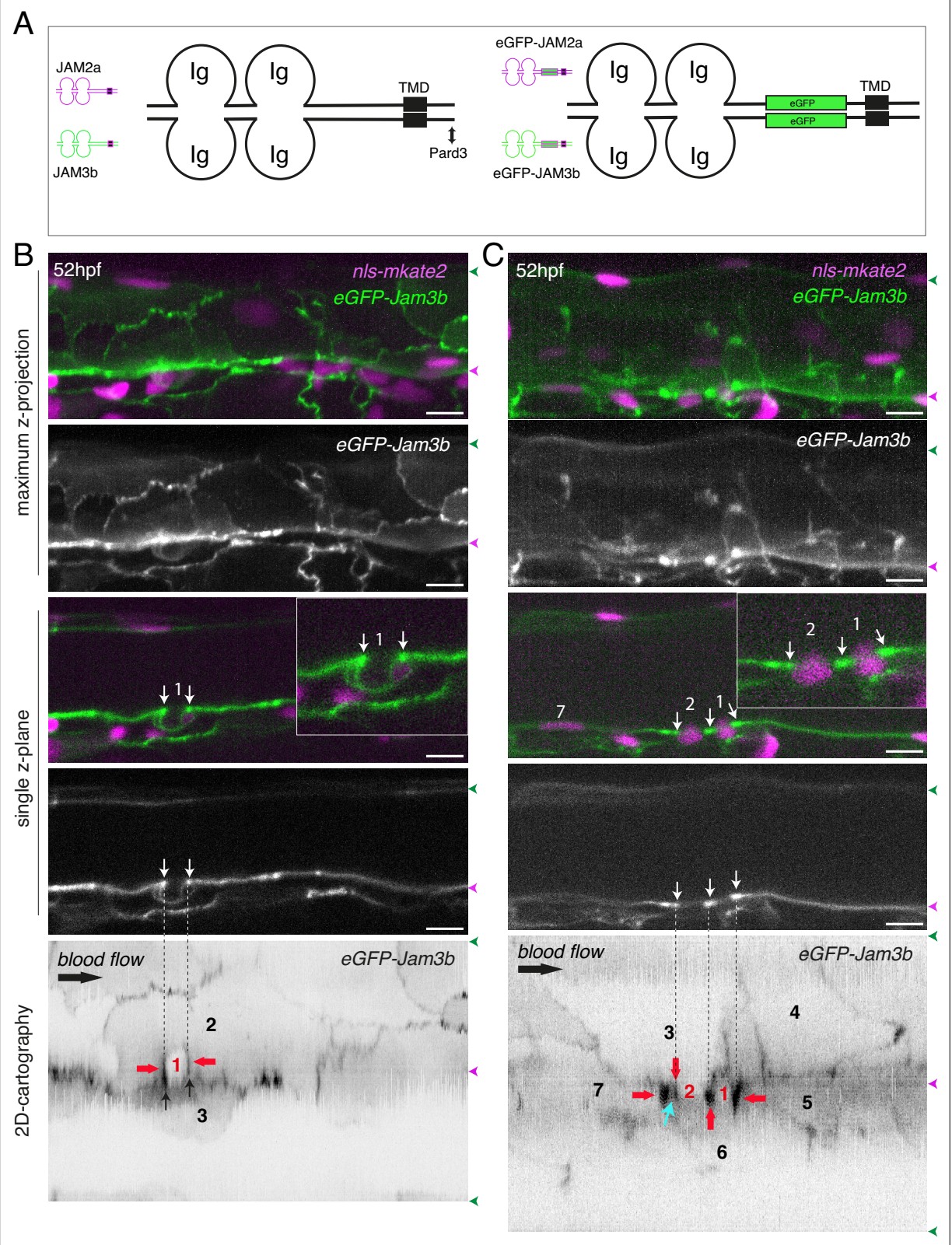

**Figure 5.** eGFP-Jam3b localization is reinforced at antero-posterior sites of the endothelial/EHT interface and at tri-cellular junctions. (**A**) Cartoons representing homodimers of full-length JAMs with the C-terminal cytosolic part interacting with Pard3 (JAMs interact with the first PDZ domain of Pard3) as well as the constructs generated in this study and containing eGFP inserted between the Immunoglobulin-like (Ig) domains and the trans-membrane region (TMD). The constructs were obtained for zebrafish JAM2a and JAM3b. (**B, C**) 52 hpf *Tg(kdrl:eGFP-Jam3b; kdrl:nls-mKate2)* embryos were imaged

*Figure 5 continued on next page*

Figure 5 continued

in the trunk region (AGM) using spinning disc confocal microscopy. The panels are either maximum z-projections (top two) or single plane z-sections (bottom two, focusing on the aortic floor) of aortic segments, with either the merged nls-mkate2 and eGFP-Jam3b fluorescence signals (magenta and green) or the eGFP-Jam3b signal only (black and white images). Bottom of the figure: 2D-cartographies obtained after deploying aortic cylinders and showing the eGFP-Jam3b signals only. The white/black vertical dashed lines show the correspondence of the antero-posterior junctional reinforcements in the single z-sections (white arrows) on the respective 2D-cartographies resulting from the deployment of the aortic walls (obtained from the z-stacks). (**B**) Example of an EHT pol+ cell (cell 1, white arrows point at reinforcement of signal at antero-posterior junctions). On the 2D cartography, cell 1 (red) is contacting endothelial cells 2 and 3; note the reinforcement of eGFP-Jam3b signals along antero-posterior membrane interfaces perpendicular to blood flow (red arrows) as well as at the two tri-cellular junctions visible between cells 1, 2 and 3 (black arrows). (**C**) Example of two EHT pol- cells (cells 1 and 2, white arrows point at reinforcement of signal at antero-posterior junctions). On the 2D cartography, cells 1 and 2 (red) are contacting endothelial cells 3, 4, 6 and 3, 6 respectively; note the reinforcement of eGFP-Jam3b signals along antero-posterior membrane interfaces perpendicular to blood flow (red arrows) and endothelial cell 6 that has intercalated between endothelial cell 7 and EHT pol- cell 2 (blue arrow). In right margins, magenta and green arrowheads designate the aortic floor and roof, respectively. Scale bars = 10 µm.

The online version of this article includes the following figure supplement(s) for figure 5:

**Figure supplement 1.** Model depicting the evolution of junctional interfaces and of the differential mobility of antero-posterior junctional complexes for EHT pol+ and EHT pol- cell emergence types.

**Figure supplement 2.** JAM2a and JAM3b expression and localization in diverse embryonic tissues.

EHT pol- cells whose entire junctional interface moves along the X, Y, Z 3D-axes (*Figure 5—figure supplement 1*). In the case of EHT pol- cells, the consumption of the junctional interface with adjoining endothelial cells appears to result from the converging migration of endothelial neighbors crawling over the luminal membrane, based on interpretation of our time-lapse sequences (*Figure 1—video 3*). In this context, we favoured the analysis of junctional pools localized at antero-posterior sites of emerging cells as we have shown that they are enriched with tight junction components (*Lancino et al., 2018*). In addition and in the case of EHT pol+ cells, it is conceivable that these adhesion pools – spatially restricted owing to apicobasal polarity – contribute to anchoring the emerging cell in the 2D-plane (*Figure 5—figure supplement 1*, top panel).

To achieve our goal, we designed two constructs in which eGFP is introduced in the extracellular domains of the two JAM2a and JAM3b molecules (*Figure 5A*, right cartoons). To investigate their localization and proper targeting at junctional interfaces, these constructs were expressed transiently and ubiquitously, using the *Hsp70* heat shock promoter. We observed that the two fusion proteins are efficiently targeted at cellular contacts (*Figure 5—figure supplement 2B, C, F*, orange and white arrows) as well as at the apical side of polarized epithelia such as for example cells of the ependymal canal (*Figure 5—figure supplement 2A, E*, white arrows). They were also observed at protrusions of epithelial cells and at T-tubules of muscle fibres (for JAM2a and JAM3b, respectively, see *Figure 5—figure supplement 2D, G*). We then established *Tg* fish lines expressing eGFP-Jam2a and eGFP-Jam3b under the control of the vascular kdrl promoter (*Tg(kdrl:eGFP-Jam2a)* and *Tg(kdrl:eGFP-Jam3b)*). Using these fish lines and spinning disk confocal microscopy, we observed a remarkable efficiency of targeting to intercellular junctions, for both proteins (data not shown for eGFP-Jam2a; for eGFP-Jam3b, see *Figure 5B and C*; in the maximum z-projections, the green signal is enriched at intercellular junctions; in single z-plane images, an EHT pol+ cell and 2 EHT pol- cells are visible in the left and right panels, respectively). Deployment of the aortic wall into 2D cartographies using an algorithm described previously (*Lancino et al., 2018*) allows to point precisely at junctional pools established at the interface between endothelial and EHT cells. This emphasizes on the enrichment of eGFP-Jam3b at antero-posterior poles of EHT pol+ and EHT pol- cells (*Figure 5B and C*, bottom panels, EHT and aortic endothelial cells (red and black numbers, respectively) are visualized as if the eye is located inside the aortic lumen; red arrows point at junctional reinforcements located at antero-posterior poles of EHT cells, black arrows in the left panel point at tri-cellular junctions and the blue arrow in the right panel points at intercalation of endothelial cells 3 and 6 between the EHT pol- cell (cell 2) and the endothelial neighbor 7).

Using double transgenic *Tg(kdrl:eGFP-Jam3b; kdrl:nls-mKate2)* embryos at 48–55 hpf, we addressed the recycling capacity of junctional pools using Fluorescence Recovery After Photobleaching (FRAP; for more details, see Materials and methods). The labelling of nuclei with nls-mKate2 allowed to point at EHT pol+ and EHT pol- cells unambiguously because eGFP-Jam3b is essentially targeted to junctional interfaces and not labelling luminal/abluminal membranes (*Figure 6A and B*),

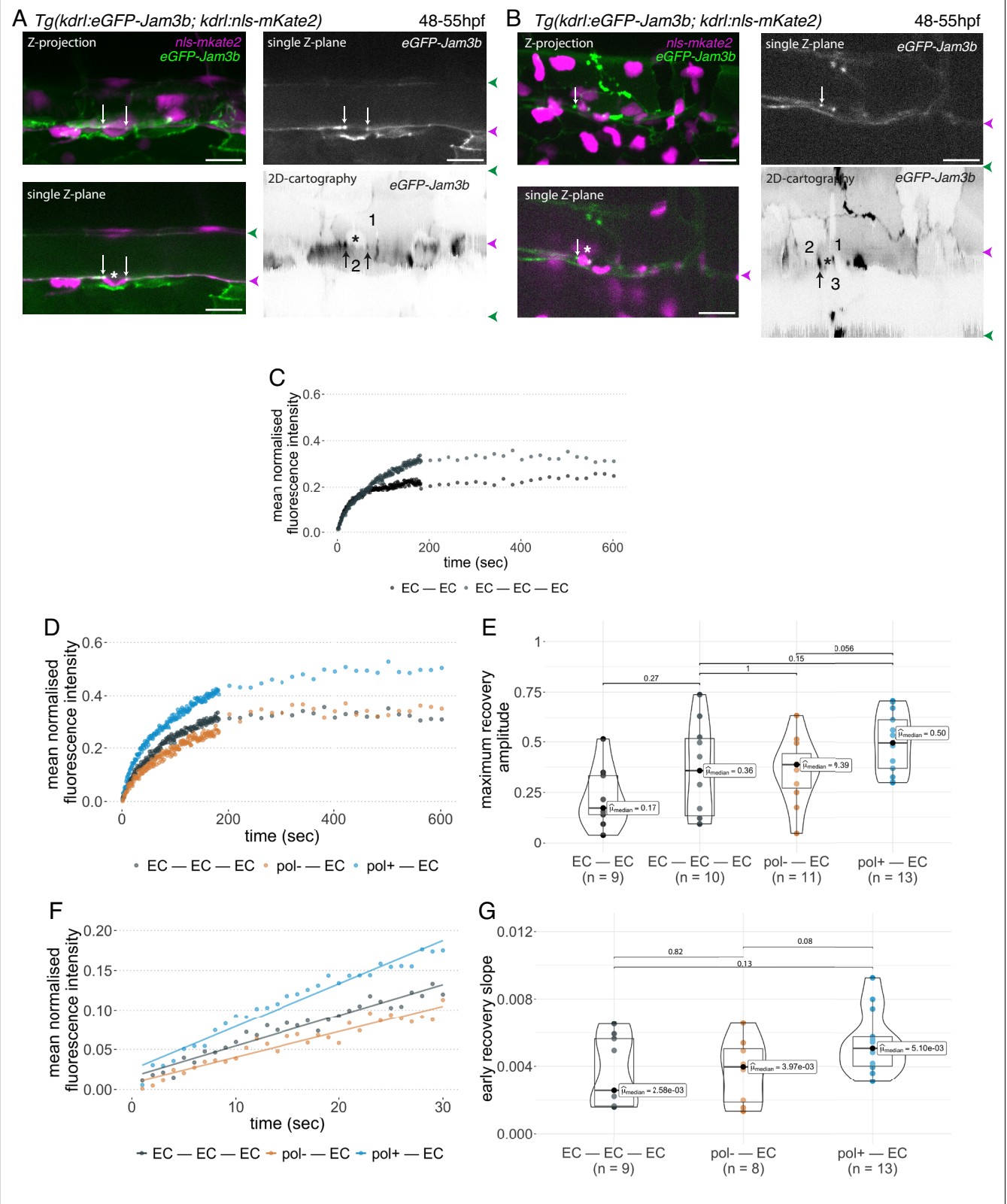

**Figure 6.** Junctional recycling at tri-cellular contacts is differentially controlled between the two EHT types. Forty-eight to 55 hpf *Tg(kdrl:eGFP-Jam3b; kdrl:nls-mKate2)* embryos were imaged using spinning disc confocal microscopy and illuminated for Fluorescence Recovery After Photobleaching (FRAP) in the trunk region (AGM, Aorta Gonad Mesonephros). (**A, B**) Panels are either maximum z-projections (top left) or single plane z-sections (bottom left and top right, focusing on the aortic floor) of aortic segments, with either the merged nls-mkate2 and eGFP-Jam3b fluorescence signals (magenta and

*Figure 6 continued on next page*

*Figure 6 continued*

green) or the eGFP-Jam3b signal only (black and white images). White arrows point at reinforcement of signal at antero-posterior junctional pools of an EHT pol+ cell (**A**) or of an EHT pol- cell (**B**), both marked by asterisks. Bottom right: 2D-cartographies obtained after deploying aortic cylinders and showing the eGFP-Jam3b signals only. Black arrows point at antero-posterior junctional pools, in particular at tri-junctional regions that exhibit increase in signal density (well visible in **A** black arrows). 2 and 3 endothelial cells are contacting the EHT pol+ cell (**A**) and the EHT pol- cell (**B**), respectively. In right margins, magenta and green arrowheads designate the aortic floor and roof, respectively. Scale bars = 20 μm. (**C—G**) FRAP analyses. EGFP-Jam3b junctional pools corresponding to the brightest spots inside junctional regions of interest were bleached for FRAP measurements (these high intensity pools were localized at the level of bi- and tri-junctions for endothelial cells (EC) and in tri-junctional regions for EHT pol+ and EHT pol- cells; all these junctional pools were systematically visualized by deploying each aortic segment before bleaching as shown in the 2D-cartographies in A and B as well as in *Figure 6—figure supplement 1*, see also Materials and methods). FRAP analysis concerned three types of junctional interfaces: between endothelial cells (EC – EC, black and grey), EHT pol- and endothelial cells (pol- – EC, brown), EHT pol+ and endothelial cells (pol+ – EC, blue). (**C, D**) Evolution of mean fluorescence intensity for each type of junctional interface over time (10 min), after photobleaching (t=0 s). (**E**) Median maximum amplitude of recovery of all determinations and for each type of junctional interface (maximum of simple exponential fitted curves). (**F, G**) Early fluorescence recovery. Early evolution (over the first 30 s) of the mean fluorescence intensity for each type of junctional interface over time after photobleaching (t=0 s). (**F**) The fitted lines correspond to linear regressions of mean fluorescence intensities. (**G**) Median values of fluorescence recovery slopes (linear regressions) of all determinations and for each type of junctional interface. (**E, G**) The number of biological replicate (n) is stated on the plots. Statistical tests: two-sided unpaired two samples Wilcoxon test.

The online version of this article includes the following figure supplement(s) for figure 6:

**Figure supplement 1.** Examples of junctional contacts targeted by FRAP in the aortic landscape.

except in few cases (*Figure 5B*, for an EHT pol+ cell); additionally, nuclei of EHT pol+ cells have a crescent shape (see *Figure 6A*).

First and to set up our protocol, we spotted bi- and tri-junctional contacts between endothelial cells (*Figure 6—figure supplement 1*). Recycling parameters (fluorescence intensity recovery with time and maximum recovery amplitude (that addresses the mobile pool), *Figure 6C*) showed that bi-junctional contacts are less mobile than tri-junctions, with a higher dispersion of maximum recovery amplitude values for the latter (*Figure 6E*); this introduced a clear limitation for statistical significance of the results, although clear tendencies were observed for mean fluorescence intensity recovery (*Figure 6C*) and median values for maximum recovery amplitude, when recycling reached its equilibrium (*Figure 6E*). We then focused at antero-posterior sites of EHT pol+ and EHT pol- cells and more specifically in the region of tricellular junctions (shared by one EHT cell and two endothelial neighbors (*Figure 6A, B* and 2D-cartographies, black arrows) and for more examples in 2D-cartographies, see *Figure 6—figure supplement 1*, black arrows) that are clearly the most enriched with eGFP-Jam3b. In each experiment, eGFP-Jam3b pools at tricellular junctions between endothelial cells (EC-EC-EC) were also spotted for a comparative analysis. Measurements of fluorescence recovery intensities revealed a tendency of increase in the mobile fraction of eGFP-Jam3b at EHT pol+ – EC versus EHT pol- – EC junctional interfaces (see *Figure 6D* for fluorescence recovery curves and *Figure 6E* for median values [50% and 39% maximum recovery amplitudes after 10 min, respectively]). Although smaller, differences in recovery were also measured between EHT pol+ – EC and EHT pol- – EC junctional interfaces when focusing on the earliest time points (the first 30 s, *Figure 6F*), with the median value of early recovery slopes for the EHT pol+ - EC versus EHT pol- – EC junctional interfaces increased by 128% (*Figure 6G*).

Altogether and unexpectedly regarding our initial scenario, these results indicate that tri-junctional pools localized at antero-posterior poles of EHT cells and enriched with eGFP-Jam3b molecules are significantly more dynamic for EHT pol+ cells in comparison to EHT pol- cells. Since EHT pol+ cells by virtue of apicobasal polarity establishment possibly assemble an apical endosome, this should favour a faster recycling of eGFP-Jam molecules (see Discussion for more details).

## ArhGEF11/PDZ-Rho GEF plays essential functions during EHT progression

Junctional maintenance and recycling are dependent on intracellular membrane trafficking, supported by sub-cortical actin remodelling and actomyosin contractility, which are controlled mainly by GTPases of the Rho family (*Ridley, 2006*; *Olayioye et al., 2019*). Owing to the significance of apicobasal polarity control on EHT features, as suggested by our work, we investigated which proteins may be essential for actin/actomyosin regulation and focused on regulators of Rho GTPases, in particular Rho GEFs

that catalyse exchange of GDP for GTP to positively regulate their activity (*Rossman et al., 2005*). As for Pard3 proteins, several of these Rho GEFs contain one or several PDZ domain(s) that target most proteins to complexes acting at the apical side therefore interlinking actin/actomyosin regulation with cell polarity (*Mack and Georgiou, 2014*; *Ebnet and Gerke, 2022*). We focused on ine PDZ-domain containing Rho GEFs, all encoded by different genes in the zebrafish (*Figure 7—figure supplement 1*): ArhGEF11/PDZ-RhoGEF (thereafter shortened as ArhGEF11), ArhGEF12a, ArhGEF12b, PRex1, PRex2, Tiam1a, Tiam1b, Tima2a, Tiam2b. We first investigated their expression by Whole mount In Situ Hybridization (WISH) and found that all nine mRNAs are detected in the aorta, and for the vast majority at 30–32 and 48–50 hpf (*Figure 7—figure supplement 2*). Then, using qRT-PCR (*Figure 7—figure supplement 1B*), we measured and compared their expression levels in the trunk region (at 35 and 48 hpf), for dt-runx1 expressing embryos and controls. We found that, in comparison to controls, ArhGEF11, ArhGEF12b, Tiam1b and Tiam2a are significantly reduced upon dt-runx1 expression at 48 hpf, hence being positively controlled in the wild-type condition, when the emergence of EHT cells is peaking. This is consistent with a functional link between these Rho-GEFs, actomyosin and the control of junctional dynamics during EHT progression. We finally decided to focus on ArhGEF11 for the following reasons: (i) in comparison to Tiams that act on Rac1, ArhGEF11, and ArhGEF12 (which are close relatives and can form heterodimers *Chikumi et al., 2004*), are mostly acting on RhoA which is controlling apical constriction via the RhoA-Myosin II signalling axis; as we have shown previously (*Lancino et al., 2018*), EHT progression requires the constriction of circumferential actomyosin; (ii) ArhGEF11 was shown to regulate the integrity of epithelial junctions by interacting directly with the scaffolding protein ZO-1, hence connecting inter-cellular adhesion with RhoA-Myosin II (*Itoh et al., 2012*); (iii) ArhGEF11-mediated apical constriction is essential during tissue morphogenesis such as, for example, the neural tube formation in which epithelial cells, submitted to mediolateral contractile forces, constrict at their apical side thus triggering inward bending of the neural plate which leads to the formation of the neural tube (*Nishimura et al., 2012*). The EHT may share features with this process, that isthe anisotropic distribution of contractile forces controlling the plane of emergence; (iv) mammalian ArhGEF11 exhibits alternative splicing in its C-terminal region that controls tight junction stability via the regulation of RhoA activity (*Lee et al., 2018*) as well as cell migration and invasion (*Itoh et al., 2017*).

To confirm the potential function of ArhGEF11 at the junctional interface between HE/EHT and endothelial cells, we first investigated its intracellular localization. We attempted to detect the full-length form upon expression of a GFP fusion protein, in the vascular system, and failed to do so. We then generated a truncated form that retains the N-terminal fragment encompassing the PDZ and RGS domains (see *Figure 7—figure supplement 3A*) fused with eGFP in its C-terminus. Upon transient expression in the vascular system, we visualized its localization at the interface between endothelial and hemogenic cells progressing throughout the EHT, with an apparent increase in density at antero-posterior regions between adjoining EHT cells (*Figure 7—figure supplement 3B*).

In line with the formerly described function of a splicing variant of ArhGEF11 in controlling tight junction integrity, particularly during egression of cells from the skin epithelium in the mouse (*Lee et al., 2018*), we questioned the potential role of such variant in EHT. This variant, referred to as the mesenchymal form, results from the inclusion of a peptide encoded by exon 37 or exon 38 in mouse and human, respectively (*Shapiro et al., 2011*; *Itoh et al., 2017*; *Lee et al., 2018*). This insertion locates in the degenerated C-terminal region of the protein, which is predicted to be relatively poorly organized in its 3D structure. Upon amplifying the full-length sequence of zebrafish ArhGEF11 for investigating its localization, we cloned fragments encoding – or not – for the insertion of a peptide of 25 amino-acid residues (75 base pairs corresponding to exon 38). Although variable when compared with other species (see *Figure 7—figure supplement 4C*, bottom panel), this peptide is inserted in the same region as it is for mammals and may correspond to an ArhGEF11 variant functionally equivalent and involved in the regulating of junctional stability (see Discussion).

To investigate the function of ArhGEF11 on the junctional interface between HE/EHT and endothelial cells, and more specifically of the isoform containing the exon 38 encoded peptide, we used both morpholino (MO) and CRISPR-based strategies. We designed a splicing MO at the 3-prime exon/intron boundary of exon 38 that interferes with its inclusion in the encoding mRNA (*Figure 7—figure supplement 4A*). This MO did not trigger any deleterious effect on the gross morphology of zebrafish embryos (*Figure 7—figure supplement 4B*) and blood circulation was normal in comparison

to control embryos. We attempted to generate CRISPR-based genomic deletion mutants of exon 38, both using Cpf1 and Cas9 enzymes and failed (see Materials and methods). However, using CRISPR/Cas9, we obtained a deletion mutant triggering a frame shift downstream of exon 38 and introducing a premature stop codon few amino-acid residues downstream thus leading to a sequence encoding for an ArhGEF11 C-terminal deleted form (see Materials and methods and *Figure 7—figure supplement 4C*). Unlike the variant skipping exon 38 induced by the splicing MO, expression of the CRISPR/Cas9 C-ter deletion mutant triggered, around 24 hpf, a severe retardation of blood circulation initiation in approximately 80% of the embryos obtained from incrossing ArhGEF11$^{CRISPR-Cterdel-/+}$ heterozygous mutant fishes. From 24–48 hpf, approximately 50% of these embryos recovered a normal blood circulation (suggesting that these embryos are probably heterozygous for the mutation), approximately 35% remained with a severe phenotype (characterized by a large pericardiac oedema) and approximately 15% died, see *Figure 7—figure supplement 4D* and Materials and methods. This indicates essential functions of the C-terminal region of ArhGEF11, in agreement with previously published data on the mammalian protein (*Chikumi et al., 2004*).

We then characterized more in depth the MO and CRISPR phenotypes (*Figure 7*) and performed a quantitative analysis of the number and morphology of HE, EHT and adjoining endothelial cells, based on confocal images and subsequent segmentation of cell contours (using 2D deployment of the aortic cylinder).

For the morphants, we found approximately a double amount of hemogenic cells in the aortic floor in comparison to controls (altogether localized on the floor and on the lateral sides of the aorta, see *Figure 7A*, panels in (a) for 2D-cartographies to visualize cell shapes, with the accumulation of hemogenic cells delineated in yellow for one morphant on the right and panels in (a') for cell counts of endothelial, hemogenic and EHT cells either in the entire aorta in trunks [left] or, more precisely, the aortic floor and lateral aortic side and the roof[right]; see also *Figure 7—figure supplement 5* for additional examples). Increase in the number of hemogenic cells in morphants in comparison to controls also concerned cells oriented more perpendicularly to the blood flow axis (toward the Y-axis, *Figure 7A*, see panels in (a")), consistently with variation in emergence angle, as we have shown previously (*Lancino et al., 2018*). This is accompanied by a significant reduction in the number of morphologically characterized EHT cells (*Figure 7A* and *Figure 7—figure supplement 5*). Similar results were obtained for the CRISPR homozygous deletion mutants, particularly regarding the accumulation of hemogenic cells, although with a smaller amplitude (*Figure 7B*, with the panels (b - b") mirroring the presentations of results obtained for morphants and internal controls in *Figure 7A*; see also *Figure 7—figure supplement 6* for additional examples). Consistently, a significant increase in the aortic perimeter was found, at least for the morphants (*Figure 7C*). Of notice also, hemogenic cells in morphants are on average less elongated in comparison to control and to the CRISPR homozygous mutants (as well as less dispersed according to this parameter, compare *Figure 7A*, panel (a''') with *Figure 7B*, panel (b'''), with visualization of the dispersion on right panels). Importantly, we observed for the morphants an accumulation of a less elongated cell population standing below the median value (*Figure 7A*, panel (a'''); to some extent, this is reproduced as well when measuring hemogenic cell area, compare *Figure 7A*, panels in (a'''') with *Figure 7B*, panels in (b'''') for morphants and CRIPR mutants, respectively). Finally, the population of small cells in the morphants are of uncharacterized morphology, which means that they appear neither as EHT pol+nor as EHT pol- cells and may represent abortive emergences (*Figure 7A* and *Figure 7—figure supplement 5*, cells in cyan boxes).

Additionally, we assessed the impact of MO interference on ArhGEF11 exon 38 splicing on hematopoiesis using the *Tg(CD41:eGFP)* fish line. By performing live imaging on *Tg(CD41:eGFP) X Tg(Kdrl:nls-mKate2)* outcrosses (see *Figure 7—figure supplement 7A and B* for control embryos and morphants, respectively), we quantified the number of hemogenic/EHT cells in the aortic floor, based on fluorescence intensities as well as the number of newly generated hematopoietic stem and progenitor cells (HSPCs, that continue to express the hematopoietic eGFP reporter but stopped to express the vascular mKate2 reporter, *Figure 7—figure supplement 7C*). We show that, in agreement with our morphometric analysis, an increase in the number of hemogenic/EHT cells in the aortic floor in morphants in comparison to control embryos (*Figure 7—figure supplement 7D*, with median values of 14.0 and 22.5 hemogenic/EHT cells for controls and morphants, respectively). Additionally, we observe, despite the high variability in both controls and morphants, a relative diminution in the number of newly generated HSPC precursors (*Figure 7—figure supplement 7E*, with median values

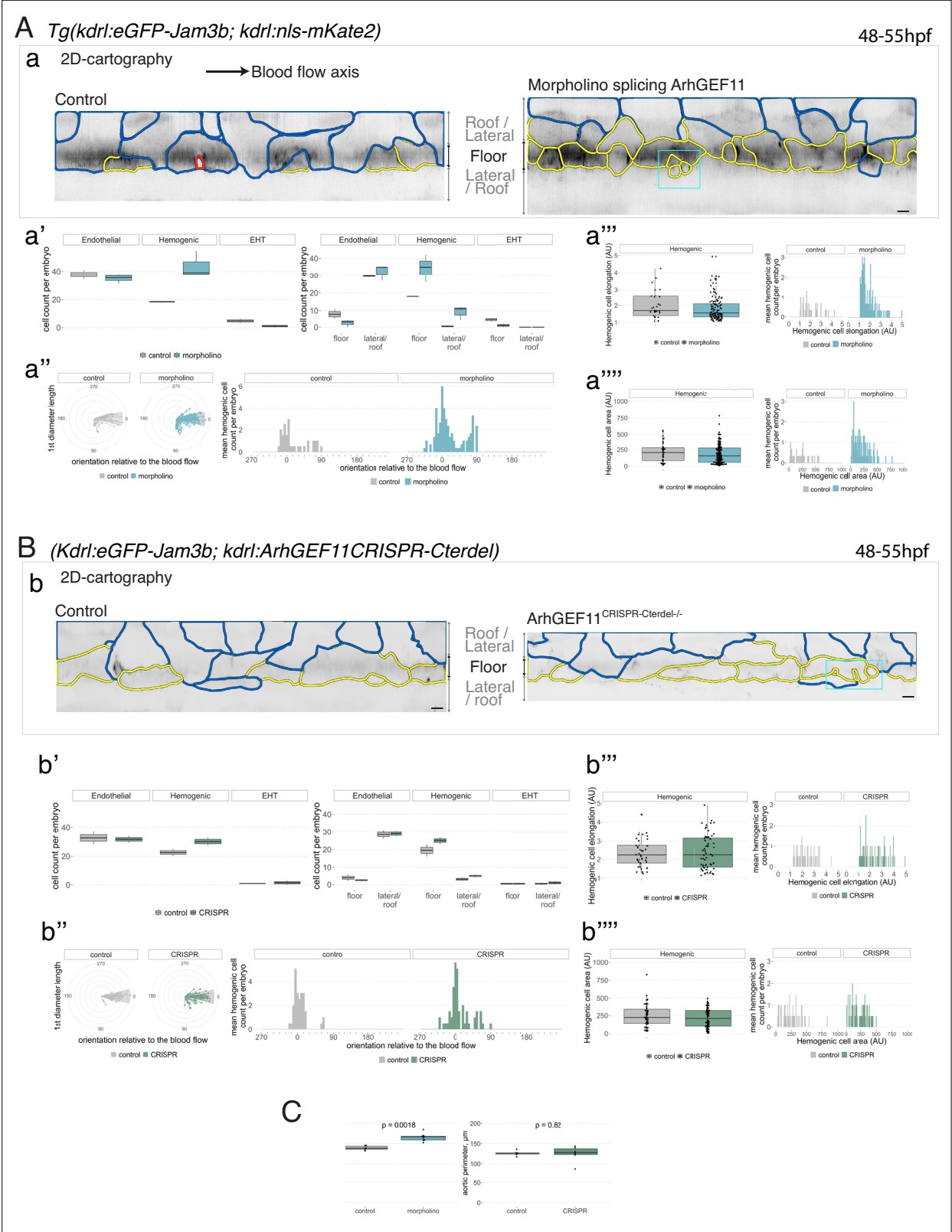

**Figure 7.** Interfering with ArhGEF11/PDZ-RhoGEF function leads to the accumulation of hemogenic cells and impairs EHT progression. (**A-C**) Numeration and morphometric analyses of aorta and cell types for *Tg(kdrl:eGFP-Jam3b; kdrl:nls-mKate2)* ArhGEF11 exon 38 splicing morpholino-injected and control embryos (**A**), or for (*Kdrl:eGFP-Jam3b; kdrl:*ArhGEF11CRISPR-Cterdel+/+) homozygous ArhGEF11 C-ter deletion mutants and control siblings. (**B**) Forty-eight to 55 hpf embryos were imaged using spinning disk confocal microscopy. (**Aa, Bb**) 2D-cartographies obtained after deploying

*Figure 7 continued on next page*

*Figure 7 continued*

aortic cylinders and showing the eGFP-Jam3b signals only with cell contours delineated either in blue (endothelial cells), yellow (hemogenic cells, see Materials and methods for their morphological definition), red (morphologically characterized EHT cells, for controls), and small cells delineated by cyan boxes (morphologically uncharacterized EHT cells and putative post-mitotic cells remaining as pairs, included in the numeration as hemogenic cells). Cellular contours have been semi-automatically segmented along the cellular interfaces labeled with eGFP-Jam3b (see Materials and methods). Scale bars: 10 μm. (**Aa′, Bb′**) Left: numeration of endothelial, hemogenic and EHT-undergoing cells according to the position of their geometrical center (either on the aortic floor, or on the roof, or on the lateral side), for each condition; right: number of endothelial, hemogenic and EHT-undergoing cells in each condition calculated from the segmentation of 3x2D-projections per embryo and covering the entire aortic regions in the trunk. (**Aa″, Bb″**) Left: length of hemogenic cells (in the longest axis) in function of their orientation (°) relative to the blood flow axis (0–180°); right: distribution of the orientation of hemogenic cells relative to the blood flow axis, displayed as a mean distribution of cells per embryo. (**Aa‴, Bb‴**) Hemogenic cell elongation factors in arbitrary Units (scale factor given by the ratio between the first- and the second-best fitting ellipse diameters, the minimum value being 1 for a non-elongated object) represented as boxplot distribution of all segmented cells (left) or as the distribution of cell elongation factor per embryo (right), for controls and for interfering conditions as indicated. (**Aa⁗, Bb⁗**) Hemogenic cell area represented as boxplot distribution of all segmented cells (left) or as the distribution of cell area per embryo (right), for controls and for interfering conditions as indicated. (**C**) Aaortic perimeter (in μm) for controls and mutant conditions as indicated. Statistical tests: two-sided unpaired two samples Wilcoxon test. For the ArhGEF11 exon 38 splicing morpholino condition, analysis was performed on 2 x control (non-injected embryos) and 3 x embryos injected at the one-cell stage; for the CRISPR mutant condition, analysis was performed on 2 x wild-type siblings for control and 2 x homozygous mutant embryos whose DNA mutation was confirmed by sequencing. Three consecutive aortic segments per embryo were analyzed to cover the whole length of the dorsal aorta, in the trunk region (covering a distance of 990 μm per embryo). Raw images (z-stacks and 2D cartographies) for panel (**A**) and *Figure 7—figure supplement 5* (morphometric analysis of aortic cells in control and ArhGEF11 morpholino splicing conditions) are available at https://doi.org/10.5281/zenodo.10937430. Raw images (z-stacks and 2D cartographies) for panel (**B**) and *Figure 7—figure supplement 6* (morphometric analysis of aortic cells in control and ArhGEF11 CRIPSR mutant conditions) are available at https://doi.org/10.5281/zenodo.10937434.

The online version of this article includes the following figure supplement(s) for figure 7:

**Figure supplement 1.** Searching for PDZ-domain containing RhoGEFs potentially involved in the EHT.

**Figure supplement 2.** Whole mount in situ hybridizations (WISH) performed on 30–32 hpf and 48–50 hpf embryos, with probes specific for all nine PDZ-domain containing RhoGEFs that were investigated in this study.

**Figure supplement 3.** A N-terminal fragment of ArhGEF11/PDZ-RhoGEF localizes at junctional membranes with enrichment at antero-posterior sites of EHT cells.

**Figure supplement 4.** MO and CRISPR approaches to investigate the function of ArhGEF11/PDZ-RhoGEF in the EHT.

**Figure supplement 5.** Supplementary data on the ArhGEF11/PDZ-RhoGEF exon 38 splicing morpholino phenotype.

**Figure supplement 6.** Supplementary data on the ArhGEF11/PDZ-RhoGEF CRISPR C-ter deletion phenotype.

**Figure supplement 7.** MO interference with ArhGEF11 exon 38 splicing leads to the accumulation of CD41 positive cells in the aortic floor and negatively impacts on the generation of hematopoietic precursors.

---

of 22.0 and 16.5 HSPCs for controls and morphants, respectively). This is consistent with our previous results showing an accumulation of HE/EHT cells in aortic floors of morphants and highlights the negative impact of this accumulation on the generation of HSPCs.

Altogether, the morphants and CRISPR mutant phenotypes show that the progression throughout EHT is significantly impaired supporting the idea that ArhGEF11 exerts important functions in the process. Increase in the frequency of elongated cells for the CRISPR mutants in comparison to morphants suggests that interference triggered by the C-terminal deletion of ArhGEF11 is more effective at early time points. Interference with inclusion of exon 38 in morphants triggers an increase in the number of hemogenic cells of smaller area and reduced elongation indicating a more efficient progression throughout emergence in comparison to the CRISPR mutants, albeit with impairment at later stages. This may be due to alteration in the dynamics of endothelial cell intercalation which is required for emergence completion (which is also compatible with the accumulation of pairs of cells that may be post-mitotic and that do not spread apart because this also requires aortic cell intercalation).

## ArhGEF11/PDZ-RhoGEF and its variant containing the exon 38 encoded sequence control junctional dynamics to support reduction of the hemogenic/endothelial interface and cell-cell intercalation

To investigate how ArhGEF11 controls the dynamic interplay between HE/EHT and endothelial cells, we characterized further the MO and CRISPR phenotypes, at the junctional level. To do so, we performed FRAP experiments using the *Tg(Kdrl:eGFP-Jam3b)* fish line that was injected at the one

cell stage with the exon 38 splicing MO, or incrossing (*Kdrl:eGFP-Jam3b x* ArhGEF11[CRISPR-Cterdel-/+]) heterozygous fishes to obtaining homozygous mutants and wild-type alleles for control siblings.

FRAP experiments were focused on the hemogenic (HE)/endothelial (EC) membrane interface, with the support of images obtained after 2D deployments of aortic segments (*Figure 8A*). Pools of *eGFP-Jam3b* molecules localized at bi-junctional (HE-EC, green arrows in panel A) and at tri-junctional interfaces (HE-EC-EC [magenta arrows in panel A] and HE-HE-EC [black arrows in panel A]) were bleached and subsequently imaged for FRAP analysis. In the case of the MO treatment (in which case HE cells accumulate on the aortic floor, see the 2D cartography (*Figure 8A*)) we measured, in comparison to controls, an increase in FRAP recovery parameters for *eGFP-Jam3b* pools at the tri-junctional HE-HE-EC interface, with a prominent tendency for an increase in the recovery speed at early time points (*Figure 8B*, panel (b''')), values obtained from fluorescence intensity recoveries whose mean values are represented by curves in panel (b''); compare with maximum recovery amplitudes (b'), with values obtained from the plateaus of fluorescence recovery intensities whose mean values are represented in panel (b)). Results are different for the ArhGEF11[CRISPR-Cterdel+/+] homozygous mutants for which we did not observe any obvious effect at HE-HE-EC tri-junctions (*Figure 8C*, with the panels (c - c''') mirroring the presentations of results obtained for morphants and internal controls in *Figure 8B*) but, rather, a small tendency for an increase in the mobile pool at HE-EC-EC tri-junctions (that has the particularity to concern 2 endothelial cells that contact the longitudinal membrane of one HE cell, see *Figure 8A and D*).

Altogether and including our results presented *Figure 7*, these results reinforce the idea that ArhGEF11 and its variant encoded by exon 38 are involved in critical steps of EHT progression, partly by controlling the dynamics of junctional pools at the level of tri-junctions established between adjoining hemogenic and endothelial cells (see the model *Figure 8D* and its legend for a detailed analysis of mechanistic issues). The results also suggest that the +exon 38 peptide encoding variant is prominently involved in controlling the HE-HE-EC interface and more specifically during intercalation, consistently with the accumulation, upon MO interference, of EHT cells having progressed throughout the emergence but appearing to be slowed down when reaching completion (cells of smaller area and less elongated, see *Figure 7*). Finally, ArhGEF11 is controlling RhoA and, most probably, its function in EHT progression also involves actin and acto-myosin activities that are involved in the contraction of HE and EHT cells (see *Lancino et al., 2018*; this is coupled with the consumption of longitudinal and transversal membrane interfaces, see also the model *Figure 8D* and its legend).

## Discussion

Here, we bring new insight into the cell biology and mechanics of the EHT, using live confocal microscopy, RNAscope (to reveal the expression of mRNAs at the cellular resolution), and genetic interference. We show that the emergence of EHT cells is more complex than previously shown, with two types of EHT cells of highly different morphodynamics and apico-basal polarity status. We bring evidence suggesting that these differences take root in HE maturation that is primarily characterized by loss of apico-basal polarity that is then tuned by the activity of the transcription factor Runx1. This leads to two types of EHT cells whose respective emergence characteristics appear to rely, partly at least, on controlling the recycling of junctional complexes directly linked to apico-basal polarity features (the JAMs). Also, we reveal essential aspects of the interplay between HE/EHT cells and adjoining aortic endothelial cells. The first is based on a potential signalling cross talk, whose molecular components remain to be identified, controlling apico-basal polarity and its associated features, in space (and possibly also in time, because of EHT asynchrony over the EHT time-window). The second involves the control of the dynamics of junctional interfaces to reduce contacting surfaces between aortic and HE/EHT cells and to promote cell-cell intercalation for the ultimate release of EHT cells and proper sealing of the aortic floor.

### EHT morphodynamics in the zebrafish embryo in comparison to amniotes

EHT cell emergence in the zebrafish embryo is relatively unique; cells extrude from the aortic floor toward the sub-aortic space and not in the aortic lumen, probably because of mechanical constraints on the aortic wall (discussed in *Lancino et al., 2018*). However, as a common feature in amniotes

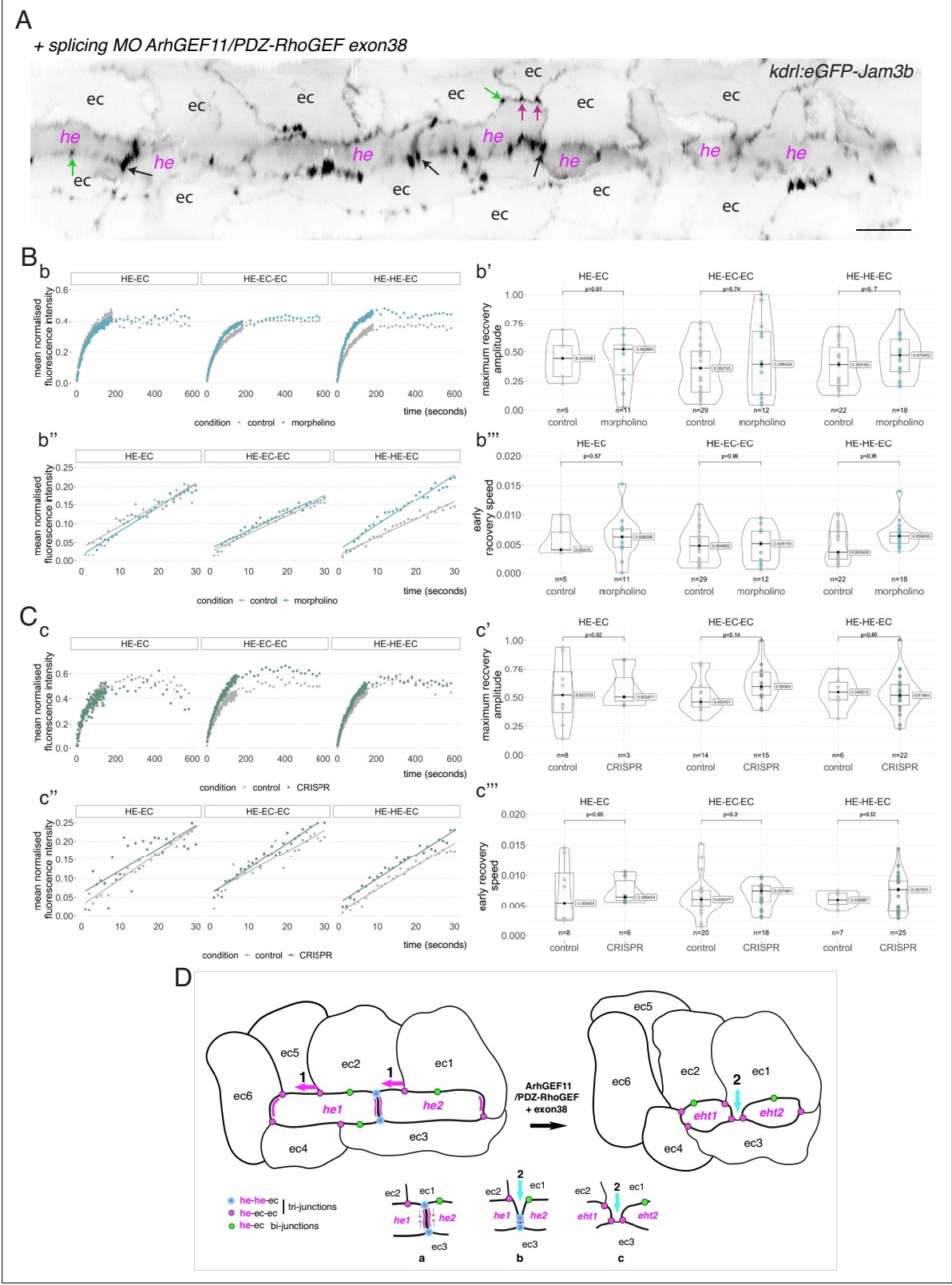

**Figure 8.** Interfering with the function of ArhGEF11/PDZ-RhoGEF suggests an activity at the interface between endothelial and hemogenic cells that relies on restricting the mobility of junctional pools at tri-cellular junctions, with a prominent role of the +exon 38 variant during endothelial cell intercalation. (**A**) 2D-cartography of an aortic segment of a *Tg(kdrl:eGFP-Jam3b; kdrl:nls-mkate2)* embryo injected at the one cell stage with the ArhGEF11 exon 38 splicing morpholino. All the arrows pointing at reinforced junctional contacts between cells were bleached and imaged for

*Figure 8 continued*

FRAP analysis (the bleached areas correspond to the spots of highest intensities in the regions of interest, as visualized at the fluorescent confocal microscope, see Materials and methods). Black and Magenta arrows point at he-he-ec and he-ec-ec tri-cellular junctions, respectively (he: hemogenic cell; ec: endothelial cell). Green arrows point at he-ec bi-cellular junctions. ec: endothelial cell, he: hemogenic cell. Scale bar = 20 µm. (**B, C**) FRAP analysis of bleached eGFP-Jam3b localized in regions of interest in controls (grey) and ArhGEF11 exon 38 splicing morpholino injected embryos (blue) or homozygous ArhGEF11$^{CRISPR-Cterdel+/+}$ mutants (green). The number of biological replicates (n) is stated on the plots. Statistical tests: two-sided unpaired two samples Wilcoxon test. Bb, Cc, evolution, after photobleaching (at t=0 s), of the mean fluorescence intensity per condition at HE-EC, HE-EC-EC and HE-HE-EC bi- and tri-junctions over time (10 min). Bb', Cc', median values for maximum amplitude of recovery (maximum of simple exponential fitted curves, see Materials and methods). Bb", Cc", early evolution, after photobleaching (at t=0 s), of the mean fluorescence intensity per condition over time (for the first 30 s). The fitted lines correspond to the linear regression of the mean fluorescence intensities. Bb''', Cc''', median fluorescence recovery slopes (linear regression). (**D**) Model (2D deployment of the aortic wall) representing the endothelial/hemogenic dynamic interplay and the proposed function of ArhGEF11 and of its + exon 38 peptide encoding variant at junctional and membrane interfaces. This interplay involves 2 essential dynamic events requiring junctional remodeling: 1 (left cartoon, magenta arrows), the mobility of he-ec-ec tri-junctional contacts accompanying the movement of endothelial cells (ex: for ec1 and ec2) along lateral sides of hemogenic cells which is required to decrease the number of adjoining cells (see *Lancino et al., 2018*). This takes place contemporarily to - or in alternance with -, the contraction of HE and EHT cells as they are progressing throughout the emergence and reducing contacting membrane surfaces along the longitudinal axis (the reduction in contacting membrane surfaces also involves membrane retrieval, hypothetically via endocytosis). Our data obtained with the CRISPR deletion mutants (a slight tendency to increase, on average, the turnover and mobile pool of these he-ec-ec junctions) suggest that ArhGEF11, in the wild type condition, should be slowing down the recycling of junctional components at tri-junctions, which hypothetically should contribute to increase adhesion strength. This may be required also to stabilize the junction-cytoskeleton interface involved in controlling the contraction/shrinkage of HE cells along the longitudinal axis, a hypothesis that is compatible with the mutant phenotype observed in this study, that is the increase in frequency of more elongated HE cells and the decrease in HE cell progression throughout EHT (see *Figure 7*); 2 (right cartoon, cyan arrow and bottom cartoons a-c), the intercalation of an endothelial cell to isolate two adjacent hemogenic cells or two daughter cells after mitosis (not depicted). This is mandatory for EHT progression and completion which requires adjoining endothelial cells to protrude membrane extensions that will anastomose to seal the aortic floor (see *Lancino et al., 2018*). The accumulation of adjoining cells of rather small length and apparently impaired in EHT progression that we describe in *Figure 7* upon MO interference (that may also indicate impairment in abscission completion) suggests that the ArhGEF11 + exon 38 peptide encoding variant is more specifically involved in controlling the remodeling of the he/he interface that leads to endothelial cell intercalation (bottom cartoons, the remodeling of he-he-ec junctions is leading to he-ec-ec junctions and takes place between b and c). The increase in the recycling parameters that we measure in this interfering condition (mobile pool and early speed of recovery) indicates that the activity of ArhGEF11, and in particular of its + exon 38 peptide encoding variant, negatively controls junctional recycling. Hypothetically, the junctional adhesion strengthening triggered by reducing the recycling of junctional components at the he/he interface may be required during the early phase and progression of intercalation to support increase in membrane tension and environmental constraints (intercalation contributes to reducing the length/surface of the he/he membrane interface preceding junctional remodeling (cartoons b and c); this reducing membrane interface is in addition submitted to the shear stress imposed by blood flow).

and the zebrafish, EHT cells appear to contain large vacuolar structures (which is also the case for HE cells in the zebrafish embryo, as we show); recently, this has been precisely described in chick and quail embryo models (*Sato et al., 2023*). In amniotes, in which EHT cells are round, these vacuoles appear to emanate from the abluminal membrane (facing the sub-aortic space) and not from the luminal membrane. In addition, these structures disappear before EHT completion, based on water chase by aquaporin activity *Sato et al., 2023*; it has been suggested that they contribute to exert mechanical force to initiate the detachment of the cell from the endothelial layer. We do not know the membranes from which the pseudo-vacuolar structures that we observe originate, particularly in the HE (they may be formed by transcytosis from the basal membrane, by endocytosis from the luminal membrane, or coming directly from the biosynthetic pathway). However, we can exclude that they contribute to the rounding of EHT pol- cells, as is the case for avians, since we do not see them during EHT pol- emergence. For EHT pol+ cells, according to our results (see *Figure 2A* and our hypothetical model *Figure 2B*), the most plausible scenario is that the apical/luminal membrane is established by fusion of cytosolic vacuolar structures or redistribution of internal pseudo-vacuoles connected to the luminal membrane. Subsequently, the dynamic properties of the EHT pol+ cell luminal/apical membrane allows its rapid evolution after EHT completion (see *Figure 2—figure supplement 1*). We propose that this is compatible with the activity of aquaporins in late EHT pol+ cells and/or shortly after their release, for water chase and vacuolar regression as proposed in our hypothetical model (see *Figure 1—figure supplement 1B* and the legend for details). Water chase activity would then be a common theme controlling the morphology and dynamics of EHT cells throughout vertebrates, although not for the same purposes.

## Characteristics of the HE and complexity of pre-hematopoietic stem cell emergence

Heterogeneity in the identity and hematopoietic potential of cells emerging from the HE in the AGM has been recently shown in co-culture systems and in vivo in the mouse (*Ganuza et al., 2017*; *Dignum et al., 2021*) and in the zebrafish model (*Tian et al., 2017*; *Xia et al., 2023*); however, knowledge on the fundamental cell biological and molecular events leading to this complexity is still lacking. Here, we highlight functional aspects that may significantly contribute to this complexity and that take root in the cellular plasticity of the HE. We bring evidence suggesting that HE cells that become competent to initiate the EHT time-window (around 28–30 hpf) tune their apicobasal polarity status which endows them with the ability to support two emergence processes with radically different morphodynamic characteristics; eventually, this would lead to differentially fated cells, which remains to be determined.

While the ability of the HE to regulate its polarity features may be inherited from molecular cues involved in its upstream intra-aortic specification (such as for example NOTCH *Robert-Moreno et al., 2008*; *Gama-Norton et al., 2015*; *Bonkhofer et al., 2019*, TGFβ *Monteiro et al., 2016*, Gata2 *Butko et al., 2015*; *Daniel et al., 2019*), or any other of the more upstream factors involved in hemogenic specification (*Zhao et al., 2022*), our results suggest that Runx1 is taking part in this regulation, consistently with its hematopoietic and EHT-inducing activity (*North et al., 1999*; *Kalev-Zylinska et al., 2002*; *Kissa and Herbomel, 2010*; *Lancrin et al., 2012*). Our experiments using an interfering form of Runx1 (dt-Runx1) that produces a relatively weak and subtle phenotype in comparison to morpholino treatment that prevents aortic cells to convert into HE cells (*Bonkhofer et al., 2019*) or genetic mutation and gene knockout abrogating hematopoiesis (*North et al., 2009*; *Sood et al., 2010*; *Gao et al., 2018*), provided the opportunity to observe that, at 30 hpf (the timing point at which a majority of HE cells are still paving the aortic floor), elongated HE cells appear to be maintained and to exhibit apicobasal polarity, with the absence of apparent cytosolic vacuolar structures (as has been described in the mouse in Runx1 interfering condition *North et al., 1999*); this correlates with the subsequent accumulation of EHT pol+ cells. This is in favour of the idea that the polarity status of HE cells has a direct incidence on the cell type that will arise, with a stronger chance to evolve as an EHT pol+ cell if apical and basal domains are already determined (which is the case for HE cells upon dt-Runx1 expression) and a lesser chance if these domains are not established and the pool of intracellular vacuoles/pseudo-vacuoles is reduced (see the hypothetical model *Figure 2B*).

A key question arising from our observations is what would control the balance orienting toward an EHT pol+ or an EHT pol- type of emergence. Cell division may be orienting toward EHT pol- emergence type (which would also be the reason why these cells have a round-shape morphology in comparison to EHT pol+ cells). The frequency of EHT pol+ versus EHT pol- cells that we have estimated and that is of approximately 2/1 is not incompatible with this possibility since divisions take place in the HE at a relatively high frequency. Indeed, it has been estimated that 50% of HE cells that constitute the aortic floor at 28 hpf will divide and will undergo emergence thereafter during the 28–72 hpf time window (*Zhao et al., 2022*). However, we observe that both EHT pol+ and EHT pol- cells can divide during the emergence process (see also our former work *Lancino et al., 2018* for EHT pol+ cells), which is followed by the full-recovery of their respective morphodynamic characteristics. Hence, this excludes the possibility of a short term and purely mechanical effect of cell division on emergence morphodynamics. In link with cell division, however, there exists the possibility that the pool of large vesicular structures potentially awaiting for exocytosis in HE cells having lost their apicobasal polarity may become asymmetrically inherited after division. This would endow the cell remaining with the largest pool to undergo EHT pol+emergence after exocytosis and expansion of the apical/luminal membrane. Our results *Figure 2* would support this possibility but what controls this asymmetry would remain to be established.

Still, regarding the control of the balance orienting toward an EHT pol+ or an EHT pol- type of emergence, the intrinsic nature of the HE, that is its acquaintance with the aortic differentiation program (*Bonkhofer et al., 2019*), raises the possibility of a partial recovery of an aortic cell phenotype in the case of EHT pol+ emergence, thus supporting its very unique mechanobiology. Therefore, the regulation of local signaling between aortic and HE cells may tune the balance toward the one or the other fate; this could be the case, for example, for adjoining HE cells in comparison to isolated ones that would then receive unequal signaling cues owing to the difference in homotypic versus heterotypic contacting surfaces. In this scenario, NOTCH signaling which is determinant for aortic and

HE specification (*Gama-Norton et al., 2015*) may play a significant role, particularly via the ratio of the Jag1/Dll4 ligands that is fundamentally involved in HE versus aortic specification (*Robert-Moreno et al., 2008*; *Bonkhofer et al., 2019*).

Finally, we cannot totally exclude that EHT pol+ and EHT pol- cells would originate from different precursors. On this line, it has been shown that the aortic wall can home a transient veinous-fated progenitor cell population that will lead to the underlying vein via selective sprouting from 21 to 23 hpf (*Herbert et al., 2009*). This option would require a massive expansion of a minor fraction of the cell population remaining in the aortic floor since it is expected that most of it has been moving out toward the sub-aortic region, which is not in favour of this possibility. In addition, recent scRNAseq analysis of floor and roof aortic cells at 21 and 28 hpf has led, for both timing points, to two well-defined clusters defined only by aortic and hematopoietic signatures (*Zhao et al., 2022*).

## Spatially restricted control of Pard3ba mRNAs by Runx1

Recent work has identified Runx1 targets in the zebrafish embryo at 29 hpf *Bonkhofer et al., 2019*; however, the results do not point at any of the conventional apicobasal polarity organizers as being direct targets, although Pard3 and Pard6 mRNAs are clearly downregulated in the HE in comparison to aortic roof cells. Our work now strengthens the idea that understanding the mechanisms under-laying the EHT may require more than simply deducing from expression of mRNAs in the HE (including the corresponding proteins), as exemplified by members of the Pard3 family. Also, annotation of protein isoforms in reference databases needs to be improved so as to gain precise and accurate insight on mechanistic processes.

Our results reveal the unique sensitivity of Pard3ba mRNAs expression to Runx1 interference, in comparison to Pard3aa and Pard3ab. Also, the dt-Runx1 mutant phenotype with enhanced expression of Pard3ba mRNAs in MERs highlighted and strengthened the existence of these specific aortic regions in wild type animals, which could not have been so obviously pointed at because of its rela-tively low and variable expression. Importantly, the expression of Pard3ba mRNAs is mainly detected in AECs and we cannot for sure say that hemogenic/EHT cells also express this mRNA because very few spots are detected and those may be at the limit of the technical background (although we measured a tendency to increase their detection upon dt-Runx1 expression, indicating that they may indeed be expressed in HE/EHT cells but at very low levels, in wild type embryos).

The sensitivity of AECs on Runx1 interference is somehow unexpected since Runx1 is rather expected to be involved in HE maturation as well as during EHT. This raises the possibility of a direct signaling cross talk between HE/EHT cells and aortic adjoining cells. In this scenario, HE/EHT cells exposed to Runx1 signaling would instruct AECs that subsequently regulate their apico-basal polarity features (via Pard3ba expression); this would then feedback on HE/EHT cells. Another plausible expla-nation is that AECs have been exposed to interference on Runx1 activity upstream of the EHT time-window, at a timing point at which aortic and HE cell types have been sharing a common ancestor, a possibility that has been debated and reinforced in the zebrafish model (*Bonkhofer et al., 2019*; *Zhao et al., 2022*). Of notice, our dt-Runx1 mutant is expressed under the vascular kdrl promoter that may be active sufficiently early (*Liao et al., 1997*) to allow efficient interference with endogenous Runx1 activity on a relatively long time scale; this may take place in a more or less restricted population of precursor cells, according to the spatially restricted localization of Pard3ba MERs.

The increase of Pard3ba mRNAs detection, upon dt-Runx1 mutant expression, suggests that endogenous Runx1 negatively controls this specific mRNA in wild type embryos. This is consistent with our results since there are two cell types/stages during the EHT process for which we observe a negative regulation of apico-basal polarity, that is the maturing HE and EHT pol- cells, as visual-ized upon expression of eGFP-podxl2 and mKate2-podxl2 (the fluorescent fusion proteins are parti-tioned equally between the luminal/apical and the basal membranes). Altogether, this reinforces the idea that Runx1 negatively controls apico-basal polarity in HE and EHT pol- cells; at least in part, it does so via Pard3ba that we hypothesize acts as a tuner, consistently with the high sensibility of its mRNA expression to Runx1 activity and the recurrent variability in expression that we are measuring. Owing to the apparent stochasticity of the EHT process in time and space (emergence takes place asynchronously during the time window of the EHT that ranges from approximately 28–72 hpf and proceeds from place to place, all along the floor of the dorsal aorta), this may be what contributes to the experimental variability that we are measuring at a given time using qRT-PCR and RNAscope (for

all determinations of Pard3ba mRNAs in comparison to the other members of the family; of notice, variations are even larger for the dt-Runx1 mutants, which is consistent with the high sensitivity to dt-Runx1 expression levels that are expected to be relatively different from one transgenic embryo to the other).

Altogether, our results are consistent with the hypothesis that the variability in expression of Pard3ba mRNAs reflects the response to intrinsic fluctuations of the system over time and space and reinforce the idea that these conditions, mediated by Runx1 and environmental constraints imposed by the blood flow, tune apico-basal polarity. This tuning scenario would also explain the specific case of EHT pol+ cells that would then proceed as such upon exposition to positive apico-basal signaling when Runx1 activity is decreased, consistently with the tendency to accumulate EHT pol+ cells upon expression of our dt-Runx1 mutant. Unfortunately and presently, we cannot show a spatial proximity between EHT pol+ cells and Pard3ba MERs in situ (or the opposite for EHT pol- cells) owing to technical limitations with fixed embryos that do not allow the maintenance of EHT cell shapes. Resolving this issue in the future will be very informative, also to place cellular morphodynamics and gene expression regulation throughout the EHT, with spatial and temporal resolution.

## The complexity of pre-HSC emergence and heterogeneity of hematopoietic stem cell and progenitor populations

Currently, we do not know what would be the features that would endow the two cell types to behave differently after their release. Since post-EHT cells have to migrate in transient developmental niches (i.e. the CHT, the thymus, the pronephric region), these features may provide them with more or less ability to migrate in the sub-aortic space and pass throughout the wall of the underlying vein to conquest more distant niches (*Murayama et al., 2006*), or to remain in a local AGM niche and be exposed to specific signaling there, in the proximity to the aorta as it is the case for hematopoietic clusters in mammals and avians (*Ciau-Uitz et al., 2014*; *Jaffredo and Yvernogeau, 2014*). Beside their potential differences in colonizing different niches which would impact on their subsequent fate, these cells may also contribute to more or less transient waves of hematopoietic stem cells and progenitors that would be specific to the developmental period such as, for example, a restricted sub-set of T-lymphocytes that was described before (*Tian et al., 2017*). In the specific case of the Tian et al. work, the AGM was shown to produces transient T-lymphocytes proposed to be independent of long-lived HSCs according to a gradient that increases from the anterior to the posterior part of the trunk region, as well as from the aorta in the tail (the posterior blood island in the tail region that was proposed to give rise to hematopoietic precursors related to mammalian erythro-myeloid progenitors, *Bertrand et al., 2007*). Since the EHT pol+ and EHT pol- cells that we describe in our work are equally produced from the aorta in the trunk and in the tail regions (thus irrespective of the antero-posterior axis of the aorta), the one or the other of the two EHT types most probably does not directly relate to the first wave of T-lymphopoiesis arising from the aortic endothelium described in *Tian et al., 2017*.

## The cell biology behind emergence control and morphodynamics

The intrinsic capacity of the HE, that we show here to be able to support two highly distinct emergence processes, in the zebrafish embryo, is very unique. Indeed, not only are they taking place contemporarily, but also can the two cell types proceed throughout emergence at a distance shorter than 50 μm, which accommodates the intercalation of a single adjoining endothelial cell. This may reflect the remarkable properties of the vascular system that needs to allow cellular extrusion and transmigration. The latter is particularly relevant in the context of developmental hematopoiesis since hematopoietic precursors that travel via blood circulation need to pass throughout the aortic wall to conquest their distant niches.

In the present work, we provide some of the mechanistic insights into the specificities of the two emergence types and that rely on two fundamental aspects of cellular biology, i.e apicobasal polarity establishment and junctional mobility. The establishment and maintenance of apicobasal polarity in a cell extruding from its environment, as is the case for EHT pol+ cells, is at odds with the fundamental mechanisms of cell extrusion irrespective of the context into which the process is taking place (*Nieto et al., 2016*; *Gudipaty and Rosenblatt, 2017*; *Pei et al., 2019*; *Staneva and Levayer, 2023*); this places the emergence of EHT pol+ cells as an extreme case of extrusion and it is somehow unexpected to observe this type of emergence when the aortic wall can extrude EHT pol- cells that appear

to emerge contemporarily, apparently according to a more conventional mode. During the EHT time-window, Runx1 is directly controlling the emergence program by regulating Rho-dependent cytoskeletal functions in HE cells derived from mouse ES lines (*Lie-A-Ling et al., 2014*) and inducing the expression of Gfi1ab in the zebrafish (*Bonkhofer et al., 2019*) that belongs to the family of transcriptional repressors of the arterial program expressed in the HE (Gfi1 and Gfi1b in the mouse, *Lancrin et al., 2012*). This program downregulates key molecules involved in the maintenance of the aortic endothelium among which Cdh5 (VE-Cadh) and the downregulation of cadherins is a hallmark of extrusion. Thus, for EHT cells, and particularly EHT pol+ cells, downregulation of VE-Cadh should be compensated by other mechanisms that allow for the maintenance of aortic integrity. In this context of adhesion downregulation, the JAMs (that are expressed endogenously in the vascular system and that were used in this study) appear to be a well-suited substitute. As part of tight junction complexes, they can strengthen adhesion and, in addition, since they are also involved in cell migration (in particular leukocyte trans-endothelial migration, see *Ebnet, 2017*), they could in theory support the extrusion of EHT pol- cells that do so according to a seemingly migration-type emergence mode (inferred from our time-lapse sequences) more or less related to trans-endothelial migration. In the case of EHT pol+ cells, we propose that JAMs may significantly contribute to the re-establishment of apicobasal polarity and to the strengthening of adhesion while EHT pol+ cells should undergo strong mechanical tension (owing to cellular bending). Interestingly, Jam3b appears to be a direct and positively regulated target of Runx1 (*Bonkhofer et al., 2019*), which is supporting its aforementioned potential functions during EHT.

In regard to the different modes of junctional recycling between aortic and either EHT pol+ or EHT pol- cells for which be bring evidence using FRAP, we hypothesize that differences take root in the regulation of cell polarity features, such as for example the organization of an apical endosome in the case of EHT pol+ cells (which may be why recycling of eGFP-Jam molecules appears to be faster in EHT pol+ cells in comparison to EHT pol- cells, in relation to the recycling pathways undertaken by JAM molecules). In addition, since JAMs interact with Pard3 (*Ebnet et al., 2003*) this recycling, that would provide renewed JAM/Pard3 complexes at membrane junctions and sustained activation of the Pard3/Pard6/aPKC complex (*Ebnet, 2017*), may contribute to the maintenance/reinforcement of EHT pol+ cell polarity throughout the entire emergence process. Also, the regime of recycling may be regulated by an adjacent aortic endothelial cell that expresses Pard3ba and that controls trans-cellular JAMs pairing via signaling involving apico-basal polarity features. Finally, this turnover may be adapted to the mechanical tensions EHT pol+ cells are exposed to, in comparison to EHT pol- cells, owing to their specific morphodynamic features (ex: cellular bending); these tensions may be sensed by tri-cellular junctions which would be preferential sites for the renewal of junctional components whose half-life is reduced owing to submission to mechanical forces.

## The cell biology behind emergence control and mechanics: the function of ArhGEF11/PDZ-RhoGEF in the dynamic interface between aortic and HE cells

Finally, regarding inter-cellular junctions and the control of their dynamics all along the EHT process, we also addressed the dynamic interface between aortic and HE cells. As for EHT cells, we focused more specifically on tri-junctional interfaces that have been proposed to sense tension and transmit the information to other types of junctional complexes (*Bosveld et al., 2018*); in the EHT context, the contracting hemogenic endothelium is additionally exposed to high mechanical tension owing to wall sheer stress imposed by the blood flow and that was proposed to regulate emergence efficiency (*Lundin et al., 2020*; *Campinho et al., 2020*; *Chalin et al., 2021*). In our work, we highlight one essential aspect of controlling junctional recycling which is during cell-cell intercalation which concerns either contacting HE cells lying on the aortic floor or HE/EHT cells after cytokinesis (as mentioned before, cell division is rather frequent during the EHT time window *Zhao et al., 2022*). In this context, we addressed the function of ArhGEF11/PDZ-RhoGEF which provides a functional link between cell polarity (with its PDZ-binding motif and its RGS domain that couples to G-protein coupled receptor signaling to regulate planar cell polarity *Nishimura et al., 2012*), cell adhesion (it binds to the tight junction associated regulatory protein ZO-1, *Itoh et al., 2012*), and cytoskeletal regulation/contraction (*Itoh et al., 2012*; *Itoh, 2013*), all three essential mechanistic components of morphogenetic movements and tissue remodelling. Since morphodynamic events leading to cell extrusion and to the

control of cell migration do not only rely on the regulation of mRNAs levels but also on alternative splicing (*Pradella et al., 2017*) our interest for investigating the function of ArhGEF11 in EHT was raised by the fact that the protein undergoes changes in its biological properties via the alternative splicing of a small exon which leads to a modification of its C-terminus. We identified a splicing event in this region (concerning exon 38) for a potential zebrafish isoform ortholog. However, the precise function of this variant for the zebrafish protein, that was shown in the case of the mammalian isoform to induce cell migration and motility (*Itoh et al., 2017*) and to bind to the Pak4 kinase thus leading to the subsequent destabilization of tight junction complexes via loss of RhoA activation (*Lee et al., 2018*), remains to be established. On this line, our results consistently indicate that preventing exon 38 insertion interferes with EHT progression (as well as deleting the C-terminus of the protein as shown with our CRISPR mutant) and appears to impair intercalation of aortic cells between adjoining HE cells. Hence, the + exon 38 isoform, by regulating the dynamic interplay between HE and endothelial cells, would favour one of the essential steps of the EHT process leading to its completion and requiring the complete isolation of EHT cells to ensure the ultimate sealing of the aortic floor (see our former work *Lancino et al., 2018*). It remains to be established, as our measurements indicate, why the ArhGEF11 + exon 38 isoform appears to slow down junctional recycling which is counterintuitive with increasing the HE/endothelial dynamic interface. However, upon interference, we have been following Jam3b recycling which is only a sub-population of tight junction components that should be at play in the system. Finally, the EHT process on its whole may be submitted to a complex regime of alternating contraction and stabilization phases as we have shown previously in the context of EHT pol+ emergence and that may reflect the necessity to adapt to mechanical constraints imposed by the environment as well as the rearrangements of the HE/EHT/aortic cellular interface (*Lancino et al., 2018*). In this context, back and forth regulatory mechanisms, particularly involved in the control of the RhoA-Myosin II signaling axis partially regulated by ArhGEF11, may locally and at specific timing points change the turnover of junctional molecules, thus blurring the correlation between gross phenotype (here the organization of the HE/aortic interface) and very local and dynamic molecular events (for a complement to the discussion, see also the legend of the model presented *Figure 8C*).

Overall, our work highlights the complexity of pre-hematopoietic stem cell emergence as well as some of the essential molecular and mechanistic aspects involved, at the interface with aortic endothelial neighbors. We show that the aorta, in the zebrafish embryo, produces two fundamentally different types of EHT cells and propose that this results from specific functional features of the HE regulated by the transcription factor Runx1. Among these features, apico-basal polarity and its regulation appear to be essential for the spatial and temporal control of EHT cell emergence which is an asynchronous process. This would support the production of cell types potentially endowed with different cell fate potential, subsequently reinforced by the type of niche into which these cells will establish homing. It now remains to be established if the different cell types that we describe are indeed leading to cells that will eventually hold specific hematopoietic fate and properties.

## Materials and methods

**Key resources table**

| Reagent type (species) or resource | Designation | Source or reference | Identifiers | Additional information |
|---|---|---|---|---|
| Strain, strain background (zebrafish *Danio rerio*) | AB | Zebrafish International Resource Center (ZIRC) | ZFIN: ZDB-GENO-960809–7 | |
| Strain, strain background (zebrafish *D. rerio*) | Tg(kdrl:HsHRAS -mCherry): s916Tg | *Chi et al., 2008* | ZFIN ID: ZDB-ALT-090506–2 | referred to as Tg(kdrl:Ras-mCherry) |
| Strain, strain background (zebrafish *D. rerio*) | Tg(6.0itga2b:EGFP) | *Lin et al., 2005* | ZFIN ID: ZDB-ALT-051223–4 | referred to as Tg(CD41:eGFP) |
| Strain, strain background (zebrafish *D. rerio*) | Tg(kdrl:eGFP) | *Jin et al., 2005* | ZFIN ID: ZDB-ALT-050916–14 | |
| Strain, strain background (zebrafish *D. rerio*) | Tg(kdrl:nls-mKate2) | N/A | N/A | |
| Strain, strain background (zebrafish *D. rerio*) | Tg(kdrl:Gal4;UAS:RFP) | N/A | N/A | |

*Continued on next page*

*Continued*

| Reagent type (species) or resource | Designation | Source or reference | Identifiers | Additional information |
|---|---|---|---|---|
| Strain, strain background (zebrafish *D. rerio*) | Tg(kdrl:Gal4;UAS:RFP;4xNR:eGFP-Podxl2) | This paper | N/A | See Materials and Methods 'Transient and stable transgenesis' |
| Strain, strain background (zebrafish *D. rerio*) | Tg(kdrl:mKate2-Podxl2) | This paper | N/A | See Materials and Methods 'Transient and stable transgenesis' |
| Strain, strain background (zebrafish *D. rerio*) | Tg(kdrl:eGFP-Jam2a) | This paper | N/A | See Materials and Methods 'Transient and stable transgenesis' |
| Strain, strain background (zebrafish *D. rerio*) | Tg(kdrl:eGFP-Jam3b) | This paper | N/A | See Materials and Methods 'Transient and stable transgenesis' |
| Strain, strain background (zebrafish *D. rerio*) | Tg(kdrl:Gal4;UAS:RFP;4xNR:dt-runx1-eGFP) | This paper | N/A | See Materials and Methods 'Transient and stable transgenesis' |
| Strain, strain background (zebrafish *D. rerio*) | Tg(ArhGEF11[CRISPR-Cterdel-/+]) | This paper | N/A | See Materials and Methods 'CRISPR methodology and GEF11 mutant transgenic line screening' |
| Recombinant DNA reagent | pG1-flk1-MCS-tol2 | *Jin et al., 2005* | N/A | |
| Recombinant DNA reagent | pEGFP-C1 | Clontech | N/A | |
| Recombinant DNA reagent | Transposase pCS-zT2TP | *Suster et al., 2011* | N/A | |
| Recombinant DNA reagent | pmKate2-f-mem | Evrogen | Cat# FP186 | |
| Antibody | goat polyclonal anti-rabbit Alexa Fluor 488 | Invitrogen | Cat# A11070 | 1/400 |
| Antibody | rabbit polyclonal anti-GFP | MBL | Cat# 598 | 1/300 |
| Antibody | mouse monoclonal anti-HA | Sigma | Cat# 12ca5 | 1/50 |
| Antibody | goat polyclonal anti-mouse HRP-conjugated | Thermo Fisher Scientific | Cat# G-21040 | 1/300 |
| Sequence-based reagent | Morpholino sih | *Sehnert et al., 2002* | ZDB-MRPHLNO-060317–4 | CATGTTTGCTCTGATCTGACACGCA |
| Sequence-based reagent | Morpholino ArhGEF11 exon 38 | This paper | N/A | GAAATAAATGAAGCCCCACCTCCGT, see Materials and Methods 'Morpholinos and injections' |
| Sequence-based reagent | Oligo for CRISPR: Alt-R CRISPR-Cas12a (Cpf1) crRNAs –1 | This paper | N/A | TATCACACACACATCACCTTCTA, see Materials and Methods 'CRISPR methodology and GEF11 mutant transgenic line screening' |
| Sequence-based reagent | Oligo for CRISPR: Alt-R CRISPR-Cas12a (Cpf1) crRNAs –2 | This paper | N/A | TTTCTCAGCGCTCCTGACAGATG, see Materials and Methods 'CRISPR methodology and GEF11 mutant transgenic line screening' |
| Sequence-based reagent | Oligo for CRISPR: Alt-R CRISPR-Cas9 crRNAs | This paper | N/A | AGCCAATCGTCTGAGGACGG, see Materials and Methods 'CRISPR methodology and GEF11 mutant transgenic line screening' |
| Sequence-based reagent | Oligo for CRISPR: Alt-R tracrRNA | IDT | Cat# 1072532 | see Materials and Methods 'CRISPR methodology and GEF11 mutant transgenic line screening' |
| Sequence-based reagent | Oligo for WISH: RNAscope Probe- Dr-myb-C3 | ACD-Biotechne | Cat No. 558291 C3 | Cat No. 558291 C2 |
| Sequence-based reagent | Oligo for WISH: RNAscope Probe- Dr-pard3ab-C3 | ACD-Biotechne | Cat No. 1282521 C3 | |
| Sequence-based reagent | Oligo for WISH: RNAscope Probe- Dr-pard3ba-C2 | ACD-Biotechne | Cat No. 1309581 C2 | |
| Sequence-based reagent | Oligo for WISH: RNAscope Probe- Dr-pard3aa-C2 | ACD-Biotechne | Cat No. 1305321 C2 | |
| Software, algorithm | Imaris | Oxford Instruments | https://imaris.oxinst.com/ | v10.1.0 |
| Software, algorithm | CRISPRscan | *Moreno-Mateos et al., 2015* | | N/A |

*Continued on next page*

*Continued*

| Reagent type (species) or resource | Designation | Source or reference | Identifiers | Additional information |
|---|---|---|---|---|
| Software, algorithm | Zen Pro2 software | Zeiss | https://www.zeiss.com/microscopy/en/products/software/zeiss-zen.html | N/A |
| Software, algorithm | QUANTSTUDIO DESIGN & ANALYSIS 2 | Thermofisher | https://apps.thermofisher.com/apps/da2/ | N/A |
| Software, algorithm | MetaMorph software | Molecular Devices | https://www.moleculardevices.com/products/cellular-imaging-systems/acquisition-and-analysis-software/metamorph-microscopy | N/A |
| Software, algorithm | Acquisition-Imaging analysis-LAS X | Leica | http://www.leica-microsystems.com | N/A |
| Software, algorithm | Image Analysis-Fiji | NIH | https://imagej.net/Fiji | N/A |
| Software, algorithm | Image Analysis Icy | *de Chaumont et al., 2012* | http://icy.bioimageanalysis.org/ | N/A |
| Software, algorithm | Image Analysis TubeSkinner plugin for Icy | *Lancino et al., 2018* | https://icy.bioimageanalysis.org/plugin/tubeskinner/ | N/A |
| Software, algorithm | Figures-Illustrator CC 2017.1.1 | Adobe | http://www.adobe.com/cn/ | N/A |
| Software, algorithm | Active contour plugin | *Dufour et al., 2011* | https://icy.bioimageanalysis.org/plugin/active-contours/ | N/A |
| Software, algorithm | NIS software | Nikon | https://www.microscope.healthcare.nikon.com/en_EU/products/software/nis-elements | N/A |
| Software, algorithm | Rstudio | *RStudio Team, 2020* | http://www.rstudio.com/ | N/A |
| Software, algorithm | ggplot2 | *Wickham, 2016* | https://ggplot2.tidyverse.org | N/A |
| Software, algorithm | readr, version 2.1.5 | *Wickham et al., 2024a; Wickham et al., 2024b* | https://github.com/tidyverse/readr | N/A |
| Software, algorithm | cowplot | *Wilke, 2020* | https://wilkelab.org/cowplot/ | N/A |
| Software, algorithm | dplyr, version 1.1.4 | *Wickham et al., 2023a; Wickham et al., 2023b* | https://github.com/tidyverse/dplyr | N/A |
| Software, algorithm | wesanderson, version 0.3.6 | *Ram and Wickham, 2018* | https://github.com/karthik/wesanderson | N/A |
| Software, algorithm | matrixStats, version 1.3.0 | *Bengtsson, 2024a* 10/04/2024 10:05:00 | https://github.com/HenrikBengtsson/matrixStats | N/A |
| Software, algorithm | stringr, version 1.5.1 | *Wickham, 2023a; Wickham, 2023b* | https://github.com/tidyverse/stringr | N/A |
| Software, algorithm | ggsci, version 3.0.3 | *Xiao, 2024a; Xiao, 2024b* | https://github.com/nanxstats/ggsci | N/A |
| Software, algorithm | ggpubr | *Kassambara, 2020* | https://rpkgs.datanovia.com/ggpubr/ | N/A |
| Software, algorithm | ggstatsplot | *Patil, 2024* | https://indrajeetpatil.github.io/ggstatsplot/index.html | N/A |
| Software, algorithm | ggbeeswarm, version 0.7.2 | *Clarke and Sherrill-Mix, 2023* | https://github.com/eclarke/ggbeeswarm | N/A |
| Software, algorithm | DT, version 0.33 | *Xie et al., 2024* | https://github.com/rstudio/DT | N/A |
| Software, algorithm | SciViews | *Grosjean, 2019* | https://www.sciviews.org/SciViews/ | N/A |
| Software, algorithm | nls.multstart, version 1.3.0 | *Padfield and Matheson, 2023a; Padfield and Matheson, 2023b* | https://github.com/padpadpadpad/nls.multstart | N/A |

List of primers: see *Supplementary file 1*.

## Contact for reagent and resource sharing

Further information and requests for resources and reagents should be directed to and will be fulfilled by the corresponding Author, Anne A. Schmidt (anne.schmidt@pasteur.fr).

## Zebrafish husbandry

Zebrafish (*Danio rerio*) of the AB background and transgenic fish carrying the following transgenes *Tg(kdrl:ras-mCherry)* (*Chi et al., 2008*); *Tg(kdrl:nls-mKate2)*; *Tg(kdrl:Gal4;UAS:RFP)*; *Tg(kdrl:eGFP)* (*Jin et al., 2005*); *Tg(CD41:eGFP)* (*Lin et al., 2005*); and the fish lines generated in this study: *Tg(kdrl:Gal4;UAS:RFP;4xNR:eGFP-podxl2)*; *Tg(kdrl:mKate2-Podxl2)*; *Tg(kdrl:eGFP-Jam2a)*;

*Tg(kdrl:eGFP-Jam3b); Tg(kdrl:Gal4;UAS:RFP;4XNR:dt-runx1-eGFP); Tg(ArhGEF11^CRISPR-Cterdel-/+)* were raised and staged as previously described (*Kimmel et al., 1995*). Adult fish lines were maintained on a 14 hr light/10 hr dark cycle. Embryos were collected and raised at either 28.5 or 24 °C in N-Phenylthiourea (PTU, Sigma-Aldrich, Cat# P7629)/Volvic source water (0.003% final) to prevent pigmentation complemented with 280 µg/L methylene blue (Sigma-Aldrich, Cat# M4159). Embryos used for imaging, extracting mRNA, or for WISH ranged from developmental stages 28-to-60 hpf and larvae used for imaging were of 3–5 dpf, precluding sex determination of the animals. The fish maintenance at the Pasteur Institute follows the regulations of the 2010/63 UE European directives and is supervised by the veterinarian office of Myriam Mattei.

## mRNA extraction and cDNA synthesis

Total RNA was extracted from whole 48 hpf embryos for cDNA cloning or from pooled trunks at the desired developmental stages (30–32 hpf, 48–50 hpf time windows for qRT-PCR experiments; ~30 individuals per tube). Briefly, embryos were anesthetized using balneation in embryo medium supplemented with tricaine methanesulfonate (MS-222, Sigma-Aldrich Cat# A5040), at a final concentration of 160 µg/ml. RNA was extracted via organic extraction using TRIzol reagent (Invitrogen Cat# 15596026) according to the manufacturer's guideline. gDNA contaminant was removed using TURBO DNase (Invitrogen Cat# AM2238) treatment according to the manufacturer's guideline. Total RNA was stored at –80 °C. Reverse transcription of mRNA was performed using SuperScriptIV (Invitrogen Cat# 18090010) with OligodT primer. cDNA samples were stored at –20 °C until further processing (for quantitative real-time PCR, PCR, cloning and sequencing).

## Transient and stable transgenesis

The *kdrl* promoter (*flk*) (*Jin et al., 2005*) was used to drive endothelial expression of mKate2-podxl2, eGFP-Jam2a and eGFP-Jam3b, with eGFP (Clontech) and mKate2 (Evrogen) cDNAs amplified using overlapping primers for Gibson cloning. eGFP-podxl2, ArhGEF11(PDZ-PRD-RGS-eGFP) and dt-runx1-eGFP were amplified for cloning into a pG1-4XNR vector (built from the 4XNR (non-repetitive) 4 X UAS sequence less susceptible to methylation than the 14 X UAS described in *Akitake et al., 2011*). For the sequence of all the designed cloning primers, see the Key Resources Table.

Podocalyxin-like 2 (Podxl2), ArhGEF11/PDZ-RhoGEF, JAM2a and JAM3b full length sequences were amplified from pools of 48 hpf whole embryo cDNA (see before for mRNA extraction and cDNA synthesis methodologies). All constructs generated in this study were obtained using the Gibson assembly assay (NEB, Cat # E2611S).

For eGFP-Jam2a and eGFP-Jam3b cloning, we used the JAM2a signaling peptide sequence (included in the 5-prime amplification primers and encoding for the signal peptide: mlvcvsllili-hsvpvspvtvssr) and, for both constructs, the eGFP sequence was inserted in frame upstream of the sequence encoding for the transmembrane domain; for the eGFP-Jam2a construct, the eGFP was inserted upstream of amino-acid D221 (DLNVAA) of the NCBI Reference Sequence: NP_001091734.1; for the eGFP-Jam3b construct, the eGFP was inserted upstream of amino-acid D243 (DINIAG) of the NCBI Reference Sequence: NP_001076332.2. For transient ubiquitous and inducible expression, the eGFP-Jam2a and eGFP-Jam3b fusion constructs were inserted into a KpnI pre-digested pG1 plasmid containing tol2 sites and the HSP70-4 promoter. For stable transgenesis using the *kdrl* (*flk1*) promoter, constructs cloned into the pG1-HSP70 plasmids were extracted using XhoI/NotI restriction enzymes and inserted into the XhoI/NotI pre-digested pG1-flk1 vector using DNA T4 ligation (NEB, Cat # M0202S).

For the Podxl2 N-ter deletion mutant that has been used for establishing stable *Tg* fish lines, the cDNA encoding for the peptide sequence starting with amino-acid G341 (GGTEYL; fragment 341–587) of the NCBI Reference Sequence: XP_692207.6 was fused with either eGFP or mKate2 cDNAs. The sequence encoding for the signal peptide of the human CD8 (malpvtalllplalllhaarpsq-frvs) was introduced upstream (5-prime) of the eGFP or mKate2 sequences (included in the 5-prime primers designed for the Gibson cloning strategy). For the construct designed to obtain the *Tg(kdrl:Gal4;UAS:RFP;4xNR:eGFP-podxl2)* fish line, two overlapping fragments were amplified for cloning into a pG1-tol2_4xNR vector after digestion with NcoI, with the following couples of primers: 4XNR_CD8-eGFP_fw and delPodxl2_eGFP_rev; delPodxl2_fw and Podxl2_pG1-4XNR_rev. For the construct designed to obtain the *Tg(kdrl:mKate2-Podxl2)* fish line two overlapping fragments were

amplified for cloning into a pG1-flk1 vector after digestion with EcoR1, with the following couples of primers: pG1-flk1_CD8_mKate2_fw and delPodxL2-mKate2_rev; mKate2-delPodxL2_fw and Podxl2_PG1-flk1_rev.

For the ArhGEF11(PDZ-PRD-RGS-eGFP) construct, the cDNA encoding for the peptide ending upstream of the DH domain at Valine 709 of the amino-acid sequence (GenBank: AY295347.1; PALDEDV; fragment 1–709) was fused upstream of the sequence encoding for eGFP, for expression using the pG1-4xNR vector. The cDNA was amplified using the GB_4xUAS-gef11_fw; GB_eGFP-RGS_ rev couple of primers.

For the construct designed to obtain the *Tg(kdrl:Gal4;UAS:RFP;4xNR:dt-runx1-eGFP)* fish line, the Runx1 sequence amplified omits the ATG (replaced by an nls) and ends downstream of the Runt domain (ending at amino-acid sequence PAHSQIP). The construct allows for the expression of the dt-runx1 protein under the control of the 4xNR driver using the same strategy as for the above PodxL2 constructs. The dt-runx1 cDNA construct was amplified from the zebrafish Runx1 cDNA sequence (*Kalev-Zylinska et al., 2002*) using the couple of primers GB-4xnr-nlsRunx1_fw; GB-eGFP-T2A-2xHA-delRunx1_rev. The sequence of these primers each allowed for the introduction, in the protein sequence, of a nls (encoded by the fw primer) and of a 2xHA-T2A peptide (encoded by the rev primer that overlaps with the 5-prime sequence of eGFP).

For all constructs, cloning reactions were used to transform NEB 5-alpha Competent *E. coli* (NEB, Cat # C2987) and resistant clones then used for miniprep plasmid preparations (Nucleospin plasmid, Macherey-Nagel, Cat # 740588.50), followed by sequencing. Plasmid DNA from clones containing correct insertions where then used to transform *E. coli* TOP10 competent cells (Invitrogen, Cat # C404010) and purified using the endotoxin free NucleoBond Xtra Midi kit (Macherey Nagel, Cat# 740420.10). Transgenesis was then performed by co-injecting 1 nl of plasmid (at 25 ng/μl) mixed with tol2 transposase mRNA (at 25 ng/μl) transcribed from linearized pCS-zT2TP plasmid using the mMESSAGE mMACHINE SP6 kit (Ambion, Cat# AM1340). For stable transgenesis, embryos were screened for fluorescence between 24 hpf and 48 hpf and raised to adulthood. Founders with germ-line integration were then isolated by screening for fluorescence after outcrossing with AB adults and for establishment of the F1 generation.

## CRISPR methodology and GEF11 mutant transgenic line screening
### CRISPR/Cas9 mutagenesis and isolation of founders
A sgRNA was designed against the splice donor site at the end of exon 38 with the aim to interfere with the production of the splicing variant including its encoding sequence, using the CRISPRscan web tool (*Moreno-Mateos et al., 2015*). The sgRNA was obtained by annealing for 5 min at 95 °C equal volumes of 100 μM specific Alt-R crRNAs (IDT, Cat# sequence: AGCCAATCGTCTGAGGACGG ) with 100 μM Alt-R generic tracrRNA (IDT, Cat# 1072532). Cas9 Nuclease Reaction Buffer (NEB Cat# B0386A) was added to obtain a final 45 μM sgRNA stock solution. For the generation of the sgRNAs–Cas9 complex, a mix containing 18 μM sgRNA and 12 μM Cas9 protein (EnGen Spy Cas9-NLS S. pyogenes, NEB Cat# M0646T) was incubated for 10 min at room temperature. For mutagenesis, 1 nl of the sgRNA-Cas9 complex was injected into one-cell stage embryos, that were subsequently raised until 48 hpf. Bulk genomic DNA (gDNA) was extracted from 60 injected and 60 non-injected control embryos. gDNA extraction proceeded as follows: pooled samples were incubated for 3 hr at 55 °C in lysis buffer (100 mM NaCl, 20 mM Tris-HCl pH 8, 25 mM EDTA, 0.5% SDS and 2 μg/μl proteinase K), purified using phenol:chloroform. Precipitation was done in 0.1 volume of 3 M NaAc and 2.5 volume of 100% ethanol for 20 min at –20 °C. Samples were centrifuged for 30 min at 4 °C before washing with 70% ethanol and resuspension in water. gDNA was used for PCR, using DNA polymerase Platinum SuperFi (Invitrogen Cat# 12351010) and the SM31-SM32bis couple of primers (TTTCACTTTCTCTCTG CCTCTTACA; AACTCTGTCCAGATGATTGAGGAGC) flanking the targeted region. PCR products were run on gel electrophoresis before gel extraction of bands of interest (with sizes corresponding to either a wild-type allele or a deleted allele) (Nucleospin gel and PCR clean-up, Macherey-Nagel Cat# 740609.50). Fragments were cloned into blunt-end TOPO vector (Invitrogen Cat# 45024), transformed into *E. coli* TOP10 competent cells and grown on selective medium. Twelve colonies for each PCR product were sent for sequencing. Mutant sequences contained either a 7 nucleotides deletion, or a mix of 1, 4 or 7 nucleotides deletions located at the same site (at the end of exon 38, see *Figure 7—figure supplement 4A* for the 7 nucleotides deletion), attesting of the proper functioning of our guide.

A stable homogeneous mutant line was generated by injecting the sgRNA:Cas9 complex into one-cell stage *Tg(kdrl:Gal4;UAS:RFP)* embryos similarly to what was described above. A total of 100 embryos were raised until adulthood. Adult founders were identified by crossing adult F0 with AB fishes, and single embryo genotyping was performed on their progeny (10 embryos per adult fish). Briefly, single embryos were incubated in 40 μl 25 mM NaOH 0.2 mM EDTA and heated for 10 min at 95 °C, before cooling for 10 min at 4 °C and addition of 40 μl 40 mM Tris-HCl pH5 to stop the reaction. PCR was performed using 5 μl of template and specific primers (couple SM31-SM34 –TTTCACTTTCTC TCTGCCTCTTACA; ATAAATGAAGCCCCACCTCCGTCC – for wild-type (WT) allele and couple SM31-SM40 – TTTCACTTTCTCTCTGCCTCTTACA; ATGAAGCCCCACCTCAGACGATTGGC– for mutant allele). Presence of WT or mutant allele for each single embryo was assessed by the presence/absence of amplification band on gel electrophoresis. Four F0 founders were isolated, that had a transmission rate in the germline ranging from 20% to 80%. Progeny of the F0 founders and subsequent generations were raised separately until adulthood, and genotyped. After complete anesthesia (in fish water supplemented with 160 μg/ml tricaine methanesulfonate), a small tissue sample was collected from the caudal fin, placed into DNA extraction buffer (10 mM Tris pH8.2, 10 mM EDTA, 200mMNaCl, 0.5% SDS, 200 μg/ml proteinase K) and incubated 3 hrs at 50 °C. DNA precipitation and resuspension was performed as described above. Seven μL gDNA was used as template for PCR amplification using the primer couple SM31-SM32bis. Sequenced PCR products showed that all founders harbored the same 7 nucleotides deletion (see *Figure 7—figure supplement 4A*). All experiments relative to *Figure 7* and *Figure 8* were performed using the progenies of one heterozygous mutant F3 couple. For these experiments, genotyping on single embryos was performed on gDNA as mentioned above. Only WT homozygous and mutant homozygous embryos were kept for analysis.

## Alternative Cpf1 approach

As an alternative approach to the CRISPR/Cas9 deletion, we attempted to generate a second mutant line, using to CRISPR/Cpf1(Cas12a) system to delete a 315 nucleotides region encompassing exon 38 (with specific guides located in the introns before and after exon 38). The sequence of guides that have been tested are: TATCACACACACATCACCTTCTA and TTTCTCAGCGCTCCTGACAGATG. However, due to the structure of the intronic regions (repeat rich) and the necessary presence of specific PAM sequence, we only generated off-target deletions of intronic regions, without deletions of exon 38, leading us to focus on the CRISPR/Cas9 approach.

## Morpholinos and injections

Morpholinos were obtained from GeneTools (see Key Resources Table for sequences). The *sih* Tnnt2 translation start codon and flanking 5-prime sequence MO (*Sehnert et al., 2002*) as well as the ArhGEF11 exon 38 splice blocking MO (overlapping the exon 38/intron 38–39 boundary) were resuspended in ddH2O to obtain stock solutions at 2 mM and 1 mM, respectively. The *sih* MO (1.5 ng) and ArhGEF11 exon 38 MO (3 ng) were injected into one-cell stage zebrafish embryos, after dilution in ddH2O. For *sih* morphants, embryos that were used in the experiments were checked for absence of heart beating 24 hr after injection as well as before being used for dissection and RNA extractions at the 30–32 and 48–50 hpf time-windows. For measuring the efficiency of the ArhGEF11 exon 38 MO, total RNA was extracted from pools of injected and control embryos followed by mRNA reverse transcription. PCR on cDNA flanking the exon 38 was performed using DNA polymerase Platinum SuperFi (Invitrogen Cat# 12351010) and analyzed on gel electrophoresis before extraction, cloning and sequencing to verify the absence of the DNA sequence encoded by exon 38 (see *Figure 7—figure supplement 4A*).

## Whole mount single molecule fluorescent in situ hybridization (RNAscope)

Forty-eight hpf *Tg(kdrl:eGFP)* and *Tg(kdrl:Gal4;UAS:RFP;4xNR:dt-Runx1-eGFP)* embryos were fixed in 4% formaldehyde (Electron Microscopy Sciences, Cat#15712) diluted in PBS/0.1% tween20 (PBST) for 2.5 hr at room temperature, rinsed with PBST and kept in 100% methanol at –20 °C until use. On the day of the experiment, embryos were rehydrated upon sequential incubations in 75%/50%/25% MeOH/ddH20 (each for 10 min at room temperature (RT)). Embryos were then incubated for 10 min in PBST. In subsequent steps of the RNAscope procedure, the Multiplex Fluorescent Reagent Kit v2 was

used (Cat# 323100), that included H2O2; probe diluent (PD); wash buffer (WB); AMP1, AMP2, AMP3 buffers; HRP-C1, HRP-C2, HRP-C3 reagents; TSA buffer; HRP blocker. All incubations were performed in Eppendorf tubes, with a maximum of 20 embryos/tube, in a dry heating block. Embryos were then incubated with H2O2 for 10 min at RT, washed for 15 min in PBST at RT, incubated for 7 min at 37 °C with proteinase K in PBST (1/2000 from a glycerol stock at 20 mg/ml, Ambion, Cat#10259184), washed in PBST for 2x10 min at RT, and incubated for at least 2 hr at 40 °C in PD. Embryos were then incubated overnight at 40 °C with the Pard3aa-C2, Pard3ab-C3, Pard2ba-C2 RNAscope probes.

Next morning, embryos were washed for 2x10 min in WB. From that step on, extreme caution should be taken because embryos will become totally transparent, owing to the harsh conditions of the WB (we recommend that all washing steps, upon buffer removal, are performed under a binocular loupe). Embryos were then incubated sequentially with AMP buffers during 30 min for AMP1 and AMP2 and 15 min for AMP3, at 40 °C, with 10 min washes at RT between each AMP buffer incubation. Embryos were then incubated for 15 min at 40 °C with RNAscope Multiplex FL v2 HRP-C2 for Pard3aa and Pard2ba probes, HRP-C3 for the Pard3ab probe, washed for 10 min at RT with TSA buffer, incubated for 30 min at 40 °C with OPAL-570 (PNFP1488001KT) in TSA buffer (dilution 1/500), washed 2x10 min at RT in WB, incubated for 15 min at 40 °C in HRP blocker and finally washed 2x10 min in WB and 2x10 min in PBST at RT. Embryos were then embedded in 1% agarose and imaged by confocal microscopy, ideally immediately and no more than 2 days after mounting. Since eGFP can maintain its fluorescence property after fixation, Pard3 RNAscope spots could be relatively easily superposed to the fluorescence of eGFP expressed in endothelial cells.

Images need to be analyzed with 3D-rendering (for example, with Imaris) to ensure the localization of RNAscope spots in a given cells.

## Whole mount chromogenic in situ hybridization

For the sequence of all the primers used to amplify probes (relative to PDZ-RhoGEFs, *Figure 7— figure supplement 2*), see the Key Resources Table. Whole-mount chromogenic in situ hybridization was performed as described in *Lancino et al., 2018*. Probes were synthetized using the manufacturer recommendation (T7 RNA polymerase, Promega, Cat# P2075, DIG-nucleotides, Jena Bioscence, Cat# NU-803-DIGX).

Images were captured with the Zeiss Axio ZOOM V16 microscope with the Zen Pro2 software, with a brightfield transmitted light optics. Post-processing steps were performed using the Extended-Depth Focus method to combine in focus regions from multiple z-planes and convert into in a transmitted light z-stack to generate a unique in-focus image.

## Immunofluorescence detection of the HA-tagged dt-runx1 deletion mutant

Whole mount immunostaining was performed on hand-dechorionated 48 hpf *Tg(dt-runx1)* embryos to assess the localization of the 2XHA-tagged dt-runx1 protein in aortic cells. Briefly, embryos were fixed in 4% methanol free PFA (Polysciences Cat# 040181) for 3 hr at room temperature (RT) and washed in PBS/0.1% tween20 (PBT). Embryos were treated in successive baths of milliQ water supplemented with 0.1% tween20, then in cold acetone for 10 min at –20 °C and again in milliQ water plus 0.1% tween20. They were then washed in 1 X HBSS (Invitrogen Cat# 14025), and permeabilized for 45 min at RT in 1 X HBSS, 0.1% tween20, 5 mM CaCl2 and 0.1 mg/ml collagenase (Sigma-Aldrich Cat# C9891). Embryos were then rinsed in PBSDT and incubated in blocking solution 1×sheep serum (SS, Sigma Cat# S2263) for at least 4 hr at RT. They were then incubated overnight at 4 °C with primary antibodies in blocking solution: rabbit anti-GFP (MBL Cat# 598; 1/300) and mouse anti-HA (Sigma, Cat# 12ca5; 1/50). On the next day, embryos were washed several times in PBSDT and endogenous peroxidase activity was inactivated by treatment with 6%H2O2 (Sigma-Aldrich, Cat# H1009) for 1 hr at 4 °C. Embryos were then incubated in the NGS for at least 4 hrs and incubated overnight at 4 °C with secondary antibodies: goat anti-rabbit Alexa Fluor 488 (Invitrogen Cat# A11070; 1/400) and goat anti-mouse HRP-conjugated (Thermo Fisher Scientific Cat# G-21040; 1/300). Finally, embryos were rinsed several times first in PBSDT and after in PBT before HRP fluorescent revelation. Embryos were incubated for 45 min at RT in the dark in imidazole buffer (0.1 M imidazole (Sigma-Aldrich, Cat# I5513) in PBT supplemented with 1% H2O2) with Cy3 Tyramide Reagent (Cy3 NHS sigma Cat# PA13101, tyramide sigma Cat# T-2879, dimethyl formamide sigma Cat# T-8654 and triethylamine sigma Cat#

T-0886). Final washes in PBT and in 6% H2O2 for POD inactivation were performed before embryos were mounted in low-melting agarose in 1 X PBT for fluorescence confocal imaging.

## Quantitative real-time PCR gene expression analysis

### Preparation of samples

In a first line of experiments, expression of pard3 family genes by qRT-PCR as well as of some hematopoietic markers myb and runx1 was evaluated in our tissue of interest at 48–50hpf. Total RNA was obtained from FACS sorted vascular cells of control embryo using an incross of *Tg(Kdrl:gal4;UAS:RFP)* as well as non-vascular cells (as described below). Cells were collected from either whole embryos or from dissected trunks. Subsequently, to determine the impact of runx1 interference on pard3 family genes, total RNA was obtained from dissected trunk regions of 48-50hpf embryos from either incross of control *Tg(Kdrl:gal4;UAS:RFP)* or mutant *Tg(Kdrl:dt-runx1-eGFP)* fish lines. Analyses were carried on three biological replicates.

### Cell dissociation and fluorescent activated cell sorting

Single-cell suspension for FACS sorting was prepared using an optimized protocol adapted from multiple sources (*Manoli and Driever, 2012*; *Samsa et al., 2016*; *Bresciani et al., 2018*). All steps were performed with cooled solutions (4 °C), and samples were kept on ice throughout the processing to preserve the viability of the cells. Briefly, embryos were anesthetized using balneation in embryo medium supplemented with tricaine methanesulfonate (MS-222, Sigma-Aldrich Cat# A5040), at a final concentration of 640 µg/ml. Embryos were placed into a petri dish, rinsed in PBS supplemented with tricaine, and a majority of the medium was removed to prevent movement. Embryos were cut using a needle, and were then pooled in tubes and placed on ice until cell dissociation. Embryos trunks or whole embryos (around 120 or 60 respectively) were washed twice in PBS before centrifugation (300 x *g* for 1 min at 4 °C) and the supernatant was discarded. Embryos were incubated in 1 ml TrypLE medium (Life Technologies Cat# 12605–010) for 10 min on ice, with gentle pipetting every 3 mins, first with a P1000 pipette and subsequently with a P200 pipette. Samples were then centrifuged for 7 min at 300 g to remove the TrypLE and the cells were resuspended in 500 µl FACSmax medium (Genlantis Cat# T200100). Ultimately, samples were passed through a 40 µm cell strainer moistened with FACSmax buffer onto a 35 mm cell culture dish using a syringe plunger. The cell strainer and the dish were washed with 300 µl of FACS max and the flow-through was transferred to a FACS tube and kept on ice until sorting.

Cell sorting was performed on a BD FACS AriaIII cell sorter. For all samples, gating was done on SSC-A vs. FSC-A to collect cells, and then on FSC-A vs. FSC-H to keep only single cells. For control embryos, RFP positive cells as well as RFP negative cells were collected. For dt-runx1 expressing mutant, all GFP positive cells (both RFP positive and RFP negatives) were collected, as well as RFP single positives cells (not used for qPCR) and double negative cells as internal controls. FASC sorted RFP single positives cells from dt-runx1 expressing mutants were not used as control for the study of gene expression for 2 reasons. First, due to the mosaicism of the line, very few cells were RFP single positive compared to the GFP positive cells. Second, RFP single-positive cells from dt-runx1 mutant embryos would have been indirectly exposed to runx1 interference (they would have been potentially in direct contact with their neighboring endothelial cells that express the dt-runx1 mutant) and thus would not be a true no interference condition.

### RNA extraction from FACS-sorted cells

Total RNA was extracted from sorted cell fractions using the RNeasy Qiagen kit (Qiagen, Cat # 74104). Briefly, cells were sorted directly into 100 µl lysis buffer, and RNA extraction was done immediately according to the manufacturer's instructions. RNA was eluted twice with 15 µl RNA free H2O. Reverse transcription of mRNA was performed using SuperScriptIV (Invitrogen Cat# 18090010) with OligodT primer. cDNA samples were stored at –20 °C until further processing (for quantitative real-time PCR).

### Real-time qPCR

For the sequence of all the designed qRT-PCR primers, see the Key Resources Table. Primer couples were individually tested to assess their specificity as well as their efficiency. qRT-PCR was performed

using Takyon Rox SYBR 2 x Master mix blue dTTP kit (Eurogentec Cat# UF-RSMT-B0701), with starting concentrations of template of around 10 ng/µl and primer concentrations of 0.5 µM, on a Quant-Studio3 system (Applied Biosystems). Each reaction was performed in technical triplicates and in three biological replicates. Ct was determined automatically by the QUANTSTUDIO DESIGN & ANALYSIS 2 software on the Thermofisher cloud and exported to be analyzed manually. Standard deviation was calculated within the technical triplicates, and when superior to 0.3, obvious outliers were removed (Dixon's Q test). The delta-delta-Ct method (double normalization to endogenous control — zebrafish elongation factor ef1α — and to control sample) was used to compare gene expression in control and altered conditions. The delta-Ct method (normalization to endogenous control ef1 α) was used to investigate expression of genes of interest in whole embryo and trunk vascular cells versus non-vascular cells.

## In vivo confocal imaging

Embryos, dechorionated manually, were anesthetized using tricaine (Sigma Aldrich, Cat# A5040). They were then embedded on the side position in 1% low melting agarose (Promega, Cat# V2111) in a glass bottom 60µ-Dish (35 mm high; Ibidi, Cat# 81156). To avoid movements and pigmentation during image acquisitions, 1 x tricaine /1 x PTU Volvic water was incorporated to the low melting agarose and, after solidification, 1 ml of 1 x tricaine /1 x PTU Volvic water was added before sealing the dish.

Embryos were imaged with 2 confocal microscope systems. For the results presented *Figure 1*, z-stacks and time-lapse sequences were obtained using a Leica TCS SP8 inverted confocal microscope as described in *Lancino et al., 2018*.

For all the other confocal microscopy results, embryos were imaged using an Andor (Oxford Instruments) spinning disk confocal system (CSU-W1 Dual camera with 50 µm Disk pattern and single laser input (445/488/561/642 nm), LD Quad 405/488/561/640 and Tripl 445/561/640 dichroic mirrors), equipped with a Leica DMi8 fluorescence inverted microscope and CMOS cameras (Orca Flash 4.0 V2+ (Hamamatsu)). Imaging was performed using a 40 x water immersion objective (HC PL APO 40 x/1.10 Water CORR CS2), a LEDs light source (CoolLED pE-4000 16 wavelength LED fluorescence system), and the acquisitions were piloted with the support of the MetaMorph software.

RNAscope is situ hybridization were imaged similarly to live sample, using a spinning disk confocal microscope (see in vivo confocal imaging paragraph above) after embedding the embryos in 1% low-melting agarose (Promega, Cat# V2111) in PBS 1 X in a glass bottom 60µ-Dish (35 mm high; Ibidi, Cat# 81156).

## Morphological and morphometric analysis of aortic and hemogenic cells

For the morphometric analysis of the aorta and aortic cells, large z-stack of 48–55 hpf Tg(kdrl:eGFP-JAM3b; kdrl:nls-mKate2) or Tg(kdrl:eGFP-JAM2a; kdrl:nls-mKate2) encompassing the whole aortic depth were acquired with optimal z-resolution (0.3 µm z-step). For each embryo, three contiguous z-stacks of 330 µm width were acquired, allowing us to image the entirety of the AGM region, from the anterior to the posterior end of the elongated yolk.

### Samples

For ArhGEF11 splicing morpholino experiment, three splicing morpholino injected embryos and two control non-injected siblings were kept for the analysis.

For ArhGEF11 CRISPR Cter deletion and when performing the experiments, imaging was done, for putative mutants, on embryos that had shown a delay in initiation of circulation at 24 hpf, regained circulation at 48 hpf albeit with a clearly visible oedema in the cardiac region (non-altered embryos were kept as sibling controls). All imaged embryos whose signal allowed us to generate segmentable 2D-cartography from the whole length of the aorta were kept and processed further for genotyping. Genotyping confirmed that the circulation problems at 24 hpf and the remaining oedema at 48 hpf corresponded to embryos bearing the mutation.

For CRISPR mutant phenotype analyses, 2 mutant embryos and 2 wild type siblings were kept, after genotyping.

### Image analysis

The following image analysis was performed on Icy software.

For each segment, we generated a 2D-cartography of the eGFP-Jam signals at intercellular contacts of aortic cells, using the Tubeskinner plugin, according to the protocol described in *Lancino et al., 2018*. Briefly, the aorta was rotated to face the lumen, a circular ROI fitting the interior of the aorta was initialized by the experimenter on the first plane, and for each plane sequentially, the algorithm will more accurately fit the ROI to the aortic perimeter and collect the fluorescence intensity all along its longitudinal axis. The collected intensity was then projected on a 2D-plane, where the Y-axis of the 2D-cartography corresponds to the perimeter of the aorta, and the X-axis corresponds to the X-axis of the original z-stack.

Before further processing, a metadata table recapitulating the dates and conditions of experiments for each embryo was generated, and a random label was assigned to rename each acquisition. This allowed for the later part of the analysis, that involves semi-automated segmentation and classification by the user, to be blind and avoid potential bias.

Each 2D-cartography was then manually pre-segmented using the ROI Polygon tool to draw the contour of each aortic cell, using the original 3D z-stack as reference. Due to technical limitation (decreased fluorescence signal collection on the farthest lateral side of the aorta), only about 2/3 of the 2D-cartographies could be segmented (comprising the floor, the right lateral side of the aorta and the roof). The manual segmentation was then improved using the 'Active Contour' plugin, that improved the fit of the cellular contour to the fluorescence signal and slightly dilated the cell contours until the ROIs formed a joint paving. All morphometric descriptors were extracted for each cell. The number of neighbors was automatically generated for each cell by counting the number of touching adjacent ROIs.

Additionally, cells were manually classified into cell types according to morphometric parameters using as reference the original 3D z-stack and in particular the nls-mKate2 nuclear signal. EHT-undergoing cells were positioned on the floor of the aorta, have reduce antero-posterior axis length, luminal membrane inward bending for EHT pol+ cells and luminal membrane outward bending for EHT pol- cells, and a thickened nucleus. Hemogenic cells whose criteria were established in *Lancino et al., 2018* (elongated in the antero-posterior axis, with lateral membranes not trespassing the equatorial plane of the aortic wall), are positioned on the floor of the aorta, systematically have a reduced width perpendicular to the blood flow and, importantly, a thickened nucleus that protrudes out of the aortic plane. Finally, endothelial cells have a flattened nucleus positioned high on the lateral side or on the roof of the aorta.

## Measurement of the aortic perimeter

An accurate measurement of the aortic perimeter was obtained using the z-stack generated for the aortic cells' morphometric analysis. Briefly, the z-stack was rotated to face the lumen of the aorta and a ROI fitting the aortic perimeter was drawn by the user and optimized by the 'Cell Contour' plugin. The length of the perimeter was then extracted from the ROI statistics.

## Fluorescence recovery after photobleaching (FRAP) measurements and analysis

Measurement of junctional turn-over was performed on 48–55 hpf *Tg(kdrl:eGFP-JAM3b; kdrl:nls-mKate2)* embryos using a Nikon Ti2e spinning disk microscope equipped with a sCMOS camera (Photometrics, Prime 95B, pixel size 11 µm, 1,200×1,200 pixels, QE 95%) and a 40 x water objective (Numerical Aperture 1.15, Working Distance 0.6 mm, xy pixel size of 0.27 µm). Embryos were mounted as previously described.

### Samples

For ArhGEF11 splicing morpholino phenotypic analysis, 10 different embryos (injected with the splicing morpholino) obtained from 3 separate experiments and 30 control embryos (non-injected siblings or control morpholino injected siblings) obtained from 6 separate experiments were analyzed.

For EHT analysis, 16 embryos over 4 separate experiments were analyzed.

For ArhGEF11 CRISPR/Cas9 Cter deletion phenotypic analysis, 8 mutant embryos and 12 wild type siblings over 3 separate experiments were analyzed.

Overall, the exact number of interfaces analyzed is written on each figure panel.

## Acquisitions

For each acquisition, a wide two channels z-stack of the aorta was generated to capture the whole aortic volume and generate a 2D-cartography of the aortic environment (Icy, plugin Tubeskinner). This 2D-cartography of the cellular contours and the z-stack with the cellular contours and nuclei were used jointly to identify un-ambiguously cellular and junctional types and define one or multiple single point regions of interest (ROIs) for subsequent bleaching and fluorescence recovery measurement.

The sequence of acquisitions before/after bleach focused only on the eGFP-Jam3b signal to speed up the acquisition and improve the temporal resolution. The 5 steps of the acquisition can be recapitulated as follows and we obtained/performed: (1) a single z-stack of 20 µm centered on the plane of focus, (2) a 30 s time-lapse acquisition on the plane of focus to measure average pre-bleach intensity with an image every second, (3) a high-intensity stimulation of the ROIs (25% 300ms 488 FRAP laser exposure, leading to the bleaching of the Jam3b-eGFP signal), (4) a 3 min time-lapse acquisition on the plane of focus to measure fast post-bleach recovery with an image every second, and (5) a 7-min time lapse acquisition of 8 µm z-stack centered around focus plane with a 20 s time interval to measure slow post-bleach recovery. Each step was saved separately as.nd2 files.

## Signal quantification and recovery analysis

File were converted from.nd2 to.tiff using FIJI batch converter, scale factor 1.00, bilinear interpolation, no average when downsizing, images read using bioformat.

To collect fluorescence intensity, ROIs for each stimulation point was manually drawn using Fiji and named with an individual label using Fiji ROI manager. A metadata table was manually curated with all information for each ROI (junction type, condition, date of experiment, embryo number). An additional ROI was created to collect background intensity for subsequent corrections. Acquisitions with clear embryo movement or drifting were either completely removed from the dataset (~5% of the acquisitions) or only kept for early recovery analysis (see below) if the drift only occurred in the last step (fifth) of the acquisition.

Fluorescence intensity for each ROI was collected and saved automatically as a.csv file for each acquisition using a homemade Fiji macro.

All further analysis were performed using R for automatization.

Briefly, for each ROI: the fluorescent signal was normalized to the background signal ($I_{background\ normalized} = I_{roi} - I_{background}$) then normalized to mean intensity pre-bleach ($I_{normalized} = I_{background\ normalized\ ROI\ post-bleach}$ / $I_{background\ normalized\ ROI\ pre-bleach}$), and finally scaled $I_{full\ scaled\ normalized} = (I_{normalized} - min\ (I_{normalized}))/ (1- min\ (I_{normalized}))$.

A single exponential equation was used to fit the data using the nls:multistart package for parameter optimization:

$$F(t) = A * (1 - e^{-t*\tau})$$

With A the maximum amplitude of the curve, t the time, and $\tau$ a constant defining the growth rate of the curve.

The fitted curve allowed to extract quantitative information describing the recovery of each type of junction in different conditions: the mobile fraction of the bleached pool (corresponding to A, the amplitude of recovery) and the time of half recovery, related to the speed of recovery, that can be extracted from $\tau$:

$$T_{half-recovery} = ln(0.5)/(-\tau)$$

A more precise description of the recovery speed was extracted from the observation of the early time period (until 30 s post bleach), were the fluorescence intensity increases uniformly. A first-degree polynomial equation was fitted to the data (using lm function from the stats package), and the coefficient corresponding to the slope of the fitted curve was used as a metric for speed of fluorescence recovery.

## RNAscope image analysis
### Myb
Phenotypic analysis of the impact of the dt-runx1 mutant on hematopoiesis was evaluated by investigating the in situ expression of myb, using the single-molecule in situ hybridization RNA-scope method,

performed in control *Tg(Kdrl:eGFP)* and dt-runx1 mutant embryos *Tg(Kdrl:Gal4;UAS:RFP;4xNR:dt-runx1-eGFP)* (analysis was performed on four and three biological replicates (embryos) for control and mutants respectively). For each embryo, two consecutive aortic segments were analyzed (each spanning over 5–6 intersegmentary vessels). It is roughly equivalent to the length to three consecutive aortic segments that we use for our live analysis each spanning over 3–4 intersegmentary vessel. Discrepancies between live and fixed samples are probably due to the morphological alterations caused by the fixation and permeabilization steps of the RNAscope protocol. 3D-rendering of the aorta and the sub-aortic space were generated using the Imaris Software (Oxford Instruments, version 10.1.0). Both cell volume (GFP signal) and RNAscope spots were simultaneously and automatically segmented using the "Cells" tool from Imaris, that segment individual cells (generates an outer surface) as well as RNA-scope signal within cells. Cells were subsequently manually classified, based on their localization and the shape of the cell. Roof endothelial cells are located dorsally to the aortic floor, and are flat and elongated. Due to morphological alteration of the tissues and the impossibility to discriminate EHT pol-, EHT pol+ cells and the uncharacterized hemogenic cells we observed in vivo, our hemogenic / EHT cells category comprises both characterized hemogenic cells (elongated cells with thick nuclei, located on the aortic floor), uncharacterized hemogenic cells and putative EHT-undergoing cells. Sub-aortic cells were classified based on their localization, and they comprise a mix of endothelial cells from the underlying posterior cardinal vein (visible only in the control embryos and not in the dt-runx1 mutant embryos, in which arterial structures are more specifically labeled compared to veinous structures) as well as newly generated hematopoietic cells. Finally, the 'Statistics' tool from Imaris allowed us to extract cell and RNA spots count information. The data plotting and statistical analysis was performed using R software.

### Pard3

Pard3 mRNAs (encoding for Pard3ba, Pard3ab, Pard3aa) in situ expression analysis was done similarly to that of myb, except that a second RNAscope signal segmentation was performed independently of the 'Cells' segmentation, using the 'spots' tool. This allowed to capture the majority of the Pard3 signal, that is expressed in aortic cells, but with a discrepancy in the intra-cellular localization of RNA transcript, that could not be detected when restricting the segmentation of RNAscope spots to spots located inside the GFP signal. After RNAscope segmentation using the 'Spots' tool, manual curation allowed us to select specifically RNAscope spots located in aortic tissues for subsequent analysis. The 'Statistics' and 'object-object Statistics' tools were used to extract relevant spatial and count information for subsequent analysis. A key information that we used for the spatial analysis was the average distance of Pard3ba spots to Hemogenic / EHT cells, as well as the cumulative number of spots within a distance from Hemogenic / EHT cells, to investigation spatial correlation between Hemogenic / EHT cells and Pard3ba expression. Analysis was performed in control *Tg(Kdrl:eGFP)* and dt-runx1 mutant embryos *Tg(Kdrl:Gal4;UAS:RFP;4xNR:dt-runx1-eGFP)* (5 and 6 biological replicates (embryos) for control and mutants respectively, with 2 consecutive aortic segments analyzed per embryo).

### Dt-runx1 phenotype analysis – cell count

For the live dt-runx1 phenotype analysis, large z-stack of 48–55 hpf dt-runx1 expressing mutant (eGFP positive) and siblings control (eGFP negative) *Tg(Kdrl:Gal4;UAS:RFP;4xNR:dt-runx1-eGFP)* embryos, encompassing the whole aortic depth were acquired with optimal z-resolution (0.3 µm z-step). Three and seven biological replicates (embryos) for control and mutants were analyzed, respectively. For each embryo, three contiguous z-stacks of 330 µm width were acquired, allowing us to image the entirety of the AGM region. The resulting phenotype was assessed by manually counting EHT pol+and pol- cells (taking into account their position, ventral or lateral) as well as hemogenic cells. Classification was done manually, based on morphological criterias. EHT pol+ cells presented the typical EHT pol+ morphology, with clear luminal membrane invagination, that could be observed even in laterally emerging cells. EHT pol- cells displayed a clear rounded ovoid, with a clear shortening of the antero-posterior axis. Hemogenic cells correspond to endothelial cells located on the aortic floor, that are elongated along the antero-posterior axis, have a narrowed left-right axis and a thickened nucleus. Non characterized hemogenic cells were also included in this category: they correspond to cells that had potentially already advanced in the emerging process but whose morphology did not fit any of

the typical EHT pol- and pol+ characteristics (no clear invagination nor clear concave protrusion in the aortic lumen).

### Thymus phenotype analysis

To question one aspect of the hematopoietic phenotype of our dt-runx1 mutant, dt-runx1 expressing mutants (GFP +RFP-) as well as control siblings (RFP +GFP-) were imaged at 5dpf to analyze their developing thymus in vivo (analysis was performed on five mutant and five control larvae). Subsequent image analysis was performed using the Imaris software (Oxford Instruments). Individual thymic cells were automatically segmented using the 'Surface' tool. Cell count per thymus as well as morphometric information for each cell (volume) was automatically retrieve using the Imaris 'Statistics' tool, allowing us to compare the distribution of cell volume in mutant versus control embryos, as well as the total cellular volume of the thymus (sum of individual cell volumes) for each larva.

## Impact of ArhGEF11 exon 38 splicing morpholino on hematopoiesis

After morpholino injection at the one cell stage, double transgenic *Tg(CD41:eGFP; kdrl:nls-mKate2)* embryos were raised until 48hpf and imaged using spinning disk confocal microscopy. The integrality of the dorsal aorta was imaged (corresponding to 3 contiguous segments), on 8 and 6 biological replicates (embryos) for controls and morphants respectively. Quantification of the number of hemogenic cells (double CD41:eGFP and kdrl:nls-mKate2 positive cells) as well as of hematopoietic cells (simple positive CD41:eGFP and kdrl:nls-mKate2 negative cells) was performed using a in house Fiji/R script. For the segmentation, only the eGFP channel is handled. An issue with the *Tg(CD41:eGFP)* background is the presence of circulating thrombocytes, that move faster than the speed of acquisition and thus generate multiple artefactual 2D objects (only presents in single Z-planes, compared to non-moving cells whose depth spans over 15–20 z-sections). These artefactual objects are removed during the pre-processing steps, by using a Faster Temporal Median algorithm (*Jabermoradi et al., 2022*). This algorithm, by calculating median value over rolling windows of 20 frames approximate cell depth and substracting it to the pixel values, removes the 2D outliers. The rest of the pre-processing steps consists of fluorescent signal filters to ease the subsequent segmentation (contrast enhancement, background substraction, 3D gaussian blur). The cells are segmented using a TopHat algorithm and touching objects are separated using a 3D distance Watershed (*Legland et al., 2016*). Once the objects are segmented, the total fluorescence intensity in both channels as well as the background fluorescence intensities are collected using the raw images (without any signal filters). The data are subsequently processed using R for plotting, threshold selection for classification as well as statistical comparisons.

## Statistical analysis

Statistical tests used are described for each figure panel in the legend. Since our samples are usually small, the test used is an unpaired two samples non-parametrical Wilcoxon mean comparison test, without a priori knowledge on sample distribution or homoscedasticity. In the context of FRAP experiments (relative to *Figure 6* and *Figure 8*), for which we had larger sample size, we still decided not to assume that our data were normally distributed, because of the inherent variability of the biological events that are observed. Indeed, the EHT process is heterogeneous in time, with successive phases of contraction and stabilization, during which we can imagine that the dynamics of junctional recycling could vary. The distribution of speed and amplitude of recovery could thus correspond to a normal distribution, a bi- or multi-modal distribution, or a uniform distribution. In this case, the use of non-parametrical tests and the large variability we observed in the data reduces the power of statistical tests and our ability to confidently detect significant differences. In some cases, when our p-values were larger than the arbitrary threshold of 0.05, we refer to tendencies whose significance is legitimated by the overall consistency of the data (see for example between the results presented *Figure 7* and *Figure 8*).

Correlation analysis (relative *to Figure 4F*) was performed by calculating Pearson correlation coefficient and associated p-value (Student T-test) using R.

## Software availability

All graphics and plots were generated using RStudio and packages including: ggplot2 (*Wickham, 2016*), readr (*Wickham et al., 2024a*), dplyr (*Wickham et al., 2023a*), stringr (*Wickham, 2023a*), ggstatsplot (*Patil, 2024*), ggpubr (*Kassambara, 2020*), wesanderson (*Ram and Wickham, 2018*), cowplot (*Wilke, 2020*), ggsci (*Xiao, 2024a*), ggbeeswarm (*Clarke and Sherrill-Mix, 2023*), viridis (*Garnier, 2024*), DT (*Xie et al., 2024*), SciViews (*Grosjean, 2019*), nls.multstart (*Padfield and Matheson, 2023a*), matrixStats (*Bengtsson, 2024b*).

## Acknowledgements

We wish to warmly thank Philippe Herbomel for his encouragements and for ensuring the funding of this study. We thank our fish facility members Y Rolin and K Sebastien for their daily help and commitment. We thank Sandrine Schmutz and Gaëlle Letort for their expertise and invaluable assistance. We acknowledge the help of the Cytometry and Biomarkers (UTechS CB) service platform for expert assistance with FACS analysis, of the Image Analysis Hub, and of the Photonic BioImaging (UTechS PBI) service platform of the Institut Pasteur for their respective contributions. We wish to thank R Patient (University of Oxford) for the Hsp70 promoter. This work was supported by Institut Pasteur, CNRS, grants to P Herbomel from the Fondation pour la Recherche Médicale (#DEQ20160334881), the Fondation ARC pour la Recherche sur le Cancer, and the Agence Nationale de la Recherche Laboratoire d'Excellence Revive (Investissement d'Avenir, ANR-10-LABX-73), and a grant to AS from the Institut Pasteur (PTR 439-2021). LT was recipient of a PhD fellowships from the Collège Doctoral, Sorbonne Université, and from the Labex Revive (ANR-10-LABX-73).

## Additional information

### Funding

| Funder | Grant reference number | Author |
| --- | --- | --- |
| Institut Pasteur | PTR 439-2021 | Anne A Schmidt |
| Sorbonne Université | PhD fellowship from Collège Doctoral | Léa Torcq |
| Agence Nationale de la Recherche | Labex Revive (ANR-10-LABX-73) | Léa Torcq |

The funders had no role in study design, data collection and interpretation, or the decision to submit the work for publication.

### Author contributions

Léa Torcq, Conceptualization, Resources, Data curation, Formal analysis, Validation, Investigation, Visualization, Methodology, Writing – original draft; Sara Majello, Conceptualization, Data curation, Formal analysis, Validation, Investigation, Visualization, Methodology; Catherine Vivier, Data curation, Investigation, Visualization, Methodology; Anne A Schmidt, Conceptualization, Data curation, Formal analysis, Supervision, Funding acquisition, Validation, Investigation, Visualization, Methodology, Writing – original draft, Project administration, Writing – review and editing

### Author ORCIDs

Anne A Schmidt ⓘ https://orcid.org/0000-0002-8326-0937

### Ethics

The fish maintenance at the Pasteur Institute follows the regulations of the 2010/63 UE European directives and is supervised by the veterinarian office of Myriam Mattei. All transgenic fish lines established in this study obtained approval from the French Ministry of Education and Research, with the reference APAFIS#l 7634-2018112213111950 vl.

Reviewer #2 (Public Review): https://doi.org/10.7554/eLife.91429.3.sa1

Author response https://doi.org/10.7554/eLife.91429.3.sa2

## Additional files

### Supplementary files
• Supplementary file 1. Materials and methods - Supplementary Table.
• MDAR checklist

### Data availability
4 datasets (raw images) relative to Figure 3, Figure 4 (and corresponding Figure 4—figure supplement 3) and Figure 7 (and corresponding Figure 7—figure supplement 5, 6) are deposited to the open external repository Zenodo (https://zenodo.org/). The following datasets were generated for this work: Figure 3—Source data 1 Raw images (z-stacks) relative to Figure 3, dt-runx1 phenotype analysis. https://doi.org/10.5281/zenodo.10932245. Figure 4—Source data 2 Raw images (z-stacks) relative to Figure 4 and Figure 4—figure supplement 3, Pard3 mRNA expression in control and mutant conditions. https://doi.org/10.5281/zenodo.10937428. Figure 7—Source data 3 Raw images (z-stacks and 2D cartographies) relative to Figure 7A and Figure 7—figure supplement 5, morphometric analysis of aortic cells in control and ArhGEF11 morpholino splicing conditions. https://doi.org/10.5281/zenodo.10937430. Figure 7—Source data 4 Raw images (z-stacks and 2D cartographies) relative to Figure 7B and Figure 7—figure supplement 6, morphometric analysis of aortic cells in control and ArhGEF11 CRIPSR mutant conditions. https://doi.org/10.5281/zenodo.10937434.

The following datasets were generated:

| Author(s) | Year | Dataset title | Dataset URL | Database and Identifier |
|---|---|---|---|---|
| Torcq L, Majello S, Vivier C, Schmidt A | 2024 | Tuning apicobasal polarity and junctional recycling in the hemogenic endothelium orchestrates the morphodynamic complexity of emerging pre-hematopoietic stem cells—Source data 1 relative to Figure 3 | https://doi.org/10.5281/zenodo.10932245 | Zenodo, 10.5281/zenodo.10932245 |
| Torcq L, Majello S, Viver C, Schmidt A | 2024 | Tuning apicobasal polarity and junctional recycling in the hemogenic endothelium orchestrates the morphodynamic complexity of emerging pre-hematopoietic stem cells—Source data 2 relative to Figure 4 | https://doi.org/10.5281/zenodo.10937428 | Zenodo, 10.5281/zenodo.10937428 |
| Torcq L, Majello S, Vivier C, Schmidt A | 2024 | Tuning apicobasal polarity and junctional recycling in the hemogenic endothelium orchestrates the morphodynamic complexity of emerging pre-hematopoietic stem cells—Source data 3 relative to Figure 7 – ArhGEF11 morpholino splicing interference | https://doi.org/10.5281/zenodo.10937430 | Zenodo, 10.5281/zenodo.10937430 |

*Continued on next page*

*Continued*

| Author(s) | Year | Dataset title | Dataset URL | Database and Identifier |
|---|---|---|---|---|
| Torcq L, Majello S, Vivier C, Schmidt A | 2024 | Tuning apicobasal polarity and junctional recycling in the hemogenic endothelium orchestrates the morphodynamic complexity of emerging pre-hematopoietic stem cells—Source data 4 relative to Figure 7 – ArhGEF11 CRISPR interference | https://doi.org/10.5281/zenodo.10937434 | Zenodo, 10.5281/zenodo.10937434 |

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
