## [Editor Report · eLife assessment]

This **important** study presents a detailed characterization of two distinct cellular morphologies of haematopoietic stem cells undergoing endothelial to haematopoietic transition in zebrafish. It brings new information on how regulation of apico-basal polarity influences cellular behaviour, shape, and interaction with neighbouring cells. The evidence supporting the existence of these two distinct morphologies is **convincing**, using state-of-the-art confocal microscopy and image analysis of 2D-cartography.

---

## [Referee Report · Reviewer #2 (Public Review)]

In this study, Torcq and colleagues make carefull observations of the cellular morphology of haemogenic endothelium undergoing endothelial to haematopoietic transition (EHT) to become stem cells, using the zebrafish model. To achieve this, the used an extensive array of transgenic lines driving fluorescent markers, markers of apico-basal polarity (podocalixin-FP fusions) or tight junction markers (jamb-FP fusions). The use of the runx truncation to block native Runx1 only in endothelial cells is an elegant tool to achieve something akin to tissue-specific deletion of Runx1. Overall, the imaging data is of excellent quality. They demonstrate that differences in apico-basal polarity are strongly associated with different cellular morphologies of cells undergoing EHT from HE (EHT pol- and EHT pol+) which raises the exciting possibility that these morphological differences reflect heterogeneity of HE (and potentially HSCs, but this is not addressed in this manuscript) at a very early stage. They then overexpress a truncated form of Runx1 (just the runt domain) to block Runx1 function and show that more HE cells abort EHT and remain associated with the embryonic dorsal aorta. The revised version identifies pard3ab as differentially distributed in dtRunx mutants and correlates that distribution with a potential regulatory role on cell polarity. No direct evidence for their role in EHT is presented.

The manuscript has now been streamlined and reference to figures made much clearer. It provides for a clearer reading, and clearly a well thought out discussion of HE, polarity and the regulation of the EHT process. The evidence for the different cellular morphologies of cells undergoing EHT is strong, and the main claim that tuning apico-basal polarity and junctional recycling underlie morphological complexity of EHT (rather than of HSCs) is well supported by the data.

---

## [Author Response]

The following is the authors’ response to the original reviews.

**Public Reviews:**

We wish to thank the Reviewers for their critical analysis of the article and for their suggestions and comments.

In addition and beside the point-by-point answer to the Reviewers, we wish here to emphasize on three essential points that have been raised:First, we never intended (nor pretended) to address the incidence of the two EHT cell emergence processes on downstream fate, after release from the aortic floor (see for example the last paragraph of our initially submitted manuscript). We only wished to bring evidence on cell biological heterogeneity of the HE, particularly relying on cell polarity control and polarity reestablishment/reinforcement in the case of EHT pol+ cells, thus leading to emergence morphodynamic complexity. In the general context of cell extrusion in which all polarity features are generally downregulated, these are remarkable features.

Second, we inform the Reviewers that we have performed a major revision of the work on the Pard3 proteins issue the outcome of which, hopefully, substantiates significantly the idea of a tuning of cell polarity features in the HE and all along the EHT time-window, for supporting EHT pol- and EHT pol+ types of emergence. To achieve this, we entirely revised the experimental strategy to increase specificity and sensitivity of detection of Pard3 protein isoforms expressed in the vascular system, based on endothelial FACS-sorting, qRT-PCR and single-molecule whole mount in situ hybridization using RNAscope. Importantly, we wish to stress that, by addressing Pard3 proteins, we initially aimed at substantiating our observations on the localization of our podxl2 construct (del-podxl2) used to label apical membranes. Hence, we sought to bring correlative evidence on the variation of expression of polarity proteins at early and later time points of the EHT time-window (suggesting tightly regulated expression control of polarity determinants, possibly at the mRNA level). This was clearly written and justified in the text, lines 227 or 303 of the initial manuscript. Also, this may have led to identify (a) specific isoform(s), including splicing variants as initially addressed.

As the Reviewers will see, while performing the revision of our work, we now have been able to point at a specific isoform of Pard3, namely Pard3ba, whose mRNA expression level, in aortic cells and at the single cell resolution, is uniquely and specifically enhanced in cells contacting emergence ‘hot spots’. Using our Runx1 mutant fish line (dt-Runx1), we also show that expression of Pard3ba mRNAs, in these specific aortic regions, is sensitive to interference with Runx1 activity (i.e dt-Runx1 increases Pard3ba expression). Altogether, our new results strongly support our idea, initially proposed, on the regulation of polarity features during EHT; they indicates intercellular coordination, throughout cooperative cross-talk between aortic and HE/EHT cells. This is compatible with the idea of a ‘tuning’ of apico-basal polarity during the entire EHT time-window (including maturation of the HE to become competent for emergence and the emergence process per se whose morphodynamic complexity relies on regulating apico-basal polarity associated functions (ex: for controlling the specific junctional recycling modes of EHT pol+ and EHT pol- cells, as we suggest using JAM proteins that we have chosen owing to their function in the recruitment of Pard3 proteins for apico-basal polarity establishment)). This complements nicely our work and highlights the relevance of studying the interplay between aortic and HE/EHT cells (which we have started to dissect in the second part of our manuscript). Further work is obviously required to address local, dynamic variations of mRNAs encoding for this specific isoform of Pard3 as well as specific interference with its functions at the spatial and temporal levels (hence on live tissues), which is far beyond the scope of our currently submitted work.

Finally, this emphasizes the importance of the aortic context, at the mesoscopic level, in the regulation of the EHT.

Third, based on these major points and Reviewers suggestions, we propose to take into account the fact that the heterogeneity in emergence morphodynamics was not highlighted and propose the following title:

‘Tuning apicobasal polarity and junctional recycling in the hemogenic endothelium orchestrates the morphodynamic complexity of emerging pre-hematopoietic stem cells’

Regarding Results and Figures, the previous Figures 3 and 4 have been entirely revised, with the support of Supplement Figures (3 and 4 supplement figures, respectively as well as a supplement video to Figure 3). Supplement Figures have also been included to the revised version, for nearly all results that appeared as data not shown (Figure 1 – figure supplement 2: illustrating the maintenance of EHT pol+ and EHT pol- cells after division; Figure 1 – figure supplement 3: illustrating the expression of the hematopoietic marker CD41 by EHT pol+ and EHT pol- cells). Also, a new supplemental figure, Figure 7 – figure supplement 7, has been added to substantiate the impact of interfering with ArhGEF11/PDZ-RhoGEF alternative splicing on hematopoiesis. Finally, a Figure for the Reviewers is added at the end of this file that shows that virtually 100% of aortic floor cells that we consider as hemogenic cells are positive for the hematopoietic marker Gata2b which is upstream of Runx1 (using RNAscope which allows achieving cellular resolution unambiguously).

**Reviewer #1 (Public Review):**
Summary:In this research article, the authors utilized the zebrafish embryo to explore the idea that two different cell types emerge with different morphodynamics from the floor of the dorsal aorta based on their apicobasal polarity establishment. The hypothesis that the apical-luminal polarity of the membrane could be maintained after EHT and confer different functionality to the cell is exciting, however, this could not be established. There is a general lack of data supporting several of the main statements and conclusions. In addition, the manuscript is difficult to follow and needs refinement. We present below some questions and suggestions with the goal of guiding the authors to improve the manuscript and solidify their findings.

Here, we wish to emphasize that we do not make the hypothesis that ‘…the apical-luminal polarity of the membrane could be maintained after EHT …’ but that the apico-basal polarity establishment/maintenance controls the type of emergence and their associated cell biological features (EHT pol+ and EHT pol- cellular morphodynamics, establishment of membrane domains). Hence, our work suggests that these emergence modes, as a consequence of their intrinsic characteristics and differences, might have an impact on cellular behavior after the release (to place the work in the broader context of hematopoietic cell fate and differentiation). More specifically, the difference in the biological features of the luminal versus abluminal membrane for the two EHT types (ex: membrane signaling territories, membrane pools devoted to specific functions), might endow the cells with specific functional properties, after the release. What happens to those cells thereafter, except for illustrating the evolution of the luminal membrane for pol+ EHT cells, is beyond the scope of this paper. Here, we analyze and characterize some of the cell biological features of the EHT process per se (the emergence from the aortic floor), including the dynamic interface with adjoining endothelial cells.

Strengths:New transgenic zebrafish lines developed. Challenging imaging.Weaknesses:(1) The authors conclude that the truncated version of Podxl2 fused to a fluorophore is enriched within the apical site of the cell. However, based on the images provided, an alternative interpretation is that the portion of the membrane within the apical side is less stretched than in the luminal side, and therefore the fluorophore is more concentrated and easier to identify by confocal. This alternative interpretation is also supported by data presented later in the paper where the authors demonstrate that the early HE is not polarized (membranes are not under tension and stretched yet). Could the authors confirm their interpretation with a different technique/marker like TEM?

The argument of the apparent enrichment, or exclusion, of a marker depending on membrane stretching (and hence molecular packing) would be valid for any type of molecule embedded in these membranes, including of course endogenous ones (this is one of the general biophysical principles leading to the establishment of membrane domains, structurally and functionally speaking); hence, using another marker would not solve the issue because it would depends on its behavior in regard to packing (in particular lipid packing), which is difficult to anticipate and is a topic in its own (especially in this system that has been poorly investigated in regard to its biophysical and biochemical properties in vivo (including its exposure to the hemodynamics)).

If we follow the logic of the Reviewer, it appears that it is not consistent with our results on the maturing HE. Indeed, in our dt-Runx1 mutants, mKate2-podxl2 is enriched at the luminal membrane of HE cells (HE cells are elongated, and the two membrane domains have a relative equal surface and bending); in comparison, HE cells have the same morphology in control animals than in mutants but, in controls, eGFP-podxl2 and mKate2-podxl2 are equally partitioned between the luminal and abluminal membranes (see Figure 3 – figure supplement 2 (for mKate2-podxl2) and Figure 2 – figure supplement 1 and 2 (for eGFP-podxl2)). In addition, we took care while designing the eGFP and mKate2 fusions to keep the natural podxl2 sequence containing critical cysteine residues to maintain assembly properties and distance from the transmembrane segment (hence the fluorescent protein per se is not directly exposed to membrane stretching).

Finally, electron microscopy is not the approach to use for this issue because requiring tissue fixation which is always at risk because modifying significantly membrane properties. On this line, when we fix embryos (and hence membranes, see our new Figure 4 and its Supplemental Figures), we do not appear to maintain obvious EHT pol+ and pol- cell shapes. In addition, to be conclusive, the work would require not TEM but immuno-EM to be able to visualize the marker(s), which is another challenge with this system.

(2) Could the authors confirm that the engulfed membranes are vacuoles as they claimed, using, for example, TEM? Why is it concluded that "these vacuoles appear to emanate from the abluminal membrane (facing the sub-aortic space) and not from the lumen?" This is not clear from the data presented.

The same argument regarding electron microscopy mentioned on the point before is valid here (in addition, it would require serial sectioning in the case it would be technically feasible to make sure not to miss the very tinny connection that may only suggest ultimate narrowing down of the facing adjacent bilayers, which is quite challenging). The term vacuole which we use with caution (in fact, more often, we use the term pseudo-vacuoles in the initial manuscript, lines 140, 146, 1467 legend to Figure 1 – figure supplemental 1 or apparent vacuole-like in the same legend lines 1465 and 1476) is legitimate here because we cannot say that they are portions of the invaginated luminal membrane as we could be accused not to show that these membranes are still connected to the luminal surface; we are here at the limit of the resolution that in vivo imaging is allowing for the moment with this system, and we drive the attention of the Reviewer on the fact that we are reaching here a sub-cellular level which is already a challenge by itself.

In addition, if there would not be at some point vacuoles (or pseudo-vacuoles) formed in this system (membrane-bounded organelles), it would be difficult to conceive how, after release of the cell, the fluid inherited from the artic lumen would efficiently be chased from these membranes/organelles (see also our model Figure 1 – figure Supplement 1B).

Why is it concluded that "these vacuoles appear to emanate from the abluminal membrane (facing the sub-aortic space) and not from the lumen?" This is not clear from the data presented.

This is not referring to our data but to the Sato et al 2023 work. For EHT undergoing cells leading to aortic clusters in mammals and avians, vacuolar structures indeed appear to emanate from the ab-luminal side facing the sub-aortic space (we cannot call it basal because we do not know the polarity status of these cells). In the Revised version of the manuscript, we have moved this paragraph referring to the Sato et al work to the Discussion, which gives the possibility to expand a bit on this issue, for more clarity (see the second paragraph of our new Discussion).

(3) It is unclear why the authors conclude that "their dynamics appears to depend on the activity of aquaporins and it is very possible that aquaporins are active in zebrafish too, although rather in EHT cells late in their emergence and/or in post-EHT cells, for water chase and vacuolar regression as proposed in our model (Figure 1 - figure supplement 1B)." In our opinion, these figures do not confirm this statement.

This part of the text has been upgraded and moved to the Discussion (see our answer to point 2), to take Reviewers concern about clarity of the Results text section and allowing elaborating a bit more on this issue. We only wished to drive the attention on the described presence of intracellular vacuolar structures recently addressed in the Sato el al 2023 paper showing EHTcell vacuoles that are proposed to contribute to cellular deformation during the emergence. We take this example to rationalize the regression of the vacuolar structures described Figure 1 - figure supplement 1B, which is why we have written ‘… it is very possible that aquaporins are active in zebrafish too’; the first part of the sentence refers to the Sato et al 2023 paper.

(4) Could the authors prove and show data for their conclusions "We observed that both EHT pol+ and EHT pol- cells divide during the emergence"; "both EHT pol+ and EHT pol- cells express reporters driven by the hematopoietic marker CD41 (data not shown), which indicates that they are both endowed with hematopoietic potential"; and "the full recovery of their respective morphodynamic characteristics (not shown)?".

To the new version of our manuscript, we have added new Supplemental information to Figure 1 (two new Supplemental Figures):

Figure 1 - figure Supplement 2 that illustrates that both EHT pol+ and EHT pol- cells divide during the emergence as well as the maintenance of morphology for both EHT cell types. We wish also to add here that the maintenance of the EHT pol+ morphology is the most critical point, showing that dividing cells in this system do not necessarily lead to EHT pol- cells.Figure 1 - figure Supplement 3 that shows that both EHT cell types express CD41.

(5) The authors do not demonstrate the conclusion traced from Fig. 2B. Is there a fusion of the vacuoles to the apical side in the EHT pol+ cells? Do the cells inheriting less vacuoles result in pol- EHT? It looks like the legend for Fig. 2-fig supp is missing.

As said previously, showing fusion here is not technically possible, but indeed, this is the idea, which fits with the images corresponding to timing points 0-90 minutes (Figure 2A), showing (in particular for the right cell) a large pseudo-vacuole whose membrane is heavily enriched with the polarity marker podxl2 (based on fluorescence signal in a membrane-bounded organelle that, based on its curvature radius, should be more under tension then the more convoluted EHT pol+ cell luminal membrane). Also, EHT pol – cells may be born from HE cells that either inherit from less intracellular vesicles after division or that are derived from HE cells that are less – or not - exposed to polarity-dependent signaling see our data presented in the new Figure 4 and the new version of the Discussion (see paragraphs ‘Characteristics of the HE and complexity of pre-hematopoietic stem cell emergence’ and ‘Spatially restricted control of Pard3ba mRNAs by Runx1’).

Finally, the cartoon Figure 2B is a hypothetical model, consistent with our data, and that is meant to help the reader to understand the idea extrapolated from images that may not be so easy to interpret for people not working on this system. In legend of Figure 2 that describes this issue in the first version of our manuscript (lines 1241-1243), we were cautious and wrote, in parentheses: ‘note that exocytosis of the large vacuolar structure may have contributed to increase the surface of the apical/luminal membrane the green asterisk labels the lumen of the EHT pol + cell’.

The legend to Figure 2 – figure supplement 1 is not missing (see lines 1492 – 1499 of the first manuscript). The images of this supplement are not extracted from a time-lapse sequence and show that as early as 30hpf (shortly after the beginning of the EHT time-window – around 28hpf), cells on the aortic floor already exhibit podxl2-containing pseudo-vacuolar structures (which we propose is a prerequisite for HE cell maturation into EHT competent cells; see also Figure 2 – figure supplement 2).

(6) The title of the paper "Tuning apico-basal polarity and junctional recycling in the hemogenic endothelium orchestrates pre-hematopoietic stem cell emergence complexity" could be interpreted as functional heterogeneity within the HSCs, which is not demonstrated in this work. A more conservative title denoting that there are two types of EHT from the DA could avoid misinterpretations and be more appropriate.

There was no ambiguity, throughout our initial manuscript, on what we meant when using the word ‘emergence’; it refers only to the extrusion process from the aortic floor.

Reducing our title only to the 2 types of EHT cells would be very reductionist in regard to our work that also addresses essential aspects of the interplay between hemogenic cells, cells undergoing extrusion (EHT pol+ and pol- cells), and their endothelial neighbors not to mention what we show in terms of the cell biology for the maturing HE and the regulation of its interface with endothelial cells (evidence for vesicular trafficking, specific regulation of HE-endothelial cell intercalation required for EHT progression etc … ). However, and to take this specific comment into account, we propose a slightly changed title saying that there are emergences differentially characterized by their morphodynamic characteristics:

‘Tuning apicobasal polarity and junctional recycling in the hemogenic endothelium orchestrates the morphodynamic complexity of emerging pre-hematopoietic stem cells’

(7) There are several conclusions not supported by data: "Finally, we have estimated that the ratio between EHT pol+ and EHT pol- cells is of approximately 2/1". "We observed that both EHT pol+ and EHT pol- cells divide during the emergence and remain with their respective morphological characteristics". "We also observed that both EHT pol+ and EHT pol- cells express reporters driven by the hematopoietic marker CD41 (data not shown), which indicates that they are both endowed with hematopoietic potential." These conclusions are key in the paper, and therefore they should be supported by data.

Most of the requests of the Reviewer in this point have already been asked in point 4 and were added to the revised version.

Regarding the EHT pol+/pol- ratio, we will keep the ratio to approximately 2/1. The Reviewer should be aware that quantification of EHT cells is a tricky issue and a source of important variability, as can be assessed by the quantifications that we have been performing (see for example figures in which we compare the dt-Runx1 phenotype with Ctrl). This is inherent to this system, more specifically because the EHT process is asynchronous, ranging from approx. 28 hpf to 3 days post fertilization (we have even observed EHT at 5 dpf). We systematically observed heterogeneity in EHT numbers and EHT types between animals and also between experiments (some days we observe EHTs at 48 hpf, others more around 55 hpf or even later). In addition, emergence also proceeds on the lateral side of the aorta and, while it is relatively easy to identify EHT pol+ cells because of their highly characterized morphology, it is more difficult for EHT pol- cells that can be mistaken to round HE cells preparing for division. In the current revision of our work, we provide additional facts and potential explanations on the mechanisms that control this asynchrony and the apparent stochasticity of the EHT process (see results of new Figures 3 and 4).

**Reviewer #2 (Public Review):**
In this study, Torcq and colleagues make careful observations of the cellular morphology of haemogenic endothelium undergoing endothelial to haematopoietic transition (EHT) to become stem cells, using the zebrafish model. To achieve this, they used an extensive array of transgenic lines driving fluorescent markers, markers of apico-basal polarity (podocalixin-FP fusions), or tight junction markers (jamb-FP fusions). The use of the runx truncation to block native Runx1 only in endothelial cells is an elegant tool to achieve something akin to tissuespecific deletion of Runx1. Overall, the imaging data is of excellent quality. They demonstrate that differences in apico-basal polarity are strongly associated with different cellular morphologies of cells undergoing EHT from HE (EHT pol- and EHT pol+) which raises the exciting possibility that these morphological differences reflect the heterogeneity of HE (and therefore HSCs) at a very early stage. They then overexpress a truncated form of Runx1 (just the runt domain) to block Runx1 function and show that more HE cells abort EHT and remain associated with the embryonic dorsal aorta. They identify pard3aa and pard3ab as potential regulators of cell polarity. However, despite showing that loss of runx1 function leads to (late) decreases in the expression of these genes, no evidence for their role in EHT is presented. The FRAP experiments and the 2d-cartography, albeit very elegant, are difficult to interpret and not very clearly described throughout the text, making interpretation difficult for someone less familiar with the techniques. Finally, while it is clear that ArhGEF11 is playing an important role in defining cell shapes and junctions between cells during EHT, there is very little statistical evidence to support the limited data presented in the (very beautiful) images.

As mentioned in the response to reviewer 1, we revised our whole strategy for the analysis of the role of Pard3 proteins in regulating the emergence of hematopoietic precursors. Our new data, obtained using refined gene expression analysis by qRT-PCR on FACS sorted populations and by in situ gene expression analysis at the single-cell resolution using RNAscope, show first that a unique Pard3 isoform (Pard3ba) is sensitive to runx1 activity, and that its expression is specifically localized in aortic cells contacting hemogenic(HE)/EHT cells. We show a clear correlation between the densification of Pard3ba mRNAs and the presence of contacting HE/EHT cells, suggesting a key role for Pard3ba in a cross talk between aortic and hemogenic cells. Furthermore, we show that our dt-runx1 mutant impacts on the maturation of HE cells; when this mutant is expressed, we observe, in comparison to control, an accumulation of HE cells that are abnormally polarized as well as unusually high numbers of EHT pol+ cells. This strongly suggests that the polarity status of HE cells controls the mode of emergence. Overall, our work shows that regulation of apico-basal polarity features is essential for the maturation of the HE and the proper proceeding of the EHT.

We made efforts to explain more clearly the FRAP experiments as well as the analysis of 2Dcartography throughout the text to facilitate readers comprehension. 2D-cartography are an invaluable tool to precisely discriminate between endothelial and hemogenic cells, and their usage was essential during the FRAP sessions, to point at specific junctional complexes accurately. Performing FRAP at cellular junctions during aortic development was extremely challenging technically and the outcome subjected to quite significant variability (which often leads to quantitative results at the limit of the statistical significance, which is why we speak of tendencies in our results section reporting on this type of experiments). Apart from constant movement and drifting of the embryos which are sources of variability, the EHT process per se is evolving over time and does so at heterogeneous pace (for example, the apical closure of EHT pol+ cells is characterized by a succession of contraction and stabilization phases, see Lancino et al. 2018) which is an additional source of variability in the measurements. Despite all this, our data collectively and consistently suggest a differential regime of junctional dynamics between EHT cell types and support the critical function of ArhGEF11/PDZ-RhoGEF in the control of junctional turnover at the interface between HE and aortic cells as well as between HE cells to regulate cell-cell intercalation.

There is a sense that this work is both overwhelming in terms of the sheer amount of imaging data, and the work behind it to generate all the lines they required, and at the same time that there is very little evidence supporting the assertion that pard3 (and even ArhGEF11) are important mediators of cell morphology and cell fate in the context of EHT. For instance, the pard3 expression data, and levels after blocking runx1 (part of Figure 3 and Figure 4) don't particularly add to the manuscript beyond indicating that the pard3 genes are regulated by Runx1.

We thank the reviewer for the comment on the Pard3 data particularly because it led us to reconsider our strategy to address with more precision and at the cellular resolution the potential function of this protein family during the time-window of the EHT. As summarized in the header of the Public Review, we identified one specific isoform of Pard3 in the zebrafish - Pard3ba – whose sensitivity to runx1 interference and spatial restriction in expression reinforce the idea of a fine control of apico-basal polarity features and associated functions while EHT is proceeding. Our new data also reinforce the interplay between HE/EHT cells and their direct endothelial neighbors.

WeaknessesThe writing style is quite convoluted and could be simplified for clarity. For example, there is plenty of discussion and speculation throughout the presentation of the results. A clearer separation of the results from this speculation/discussion would help with understanding. Figures are frequently presented out of order in the text; modifying the figures to accommodate the flow of the text (or the other way around) - would make it much easier to follow the narrative. While the evidence for the different cellular morphologies of cells undergoing EHT is strong, the main claim (or at least the title of the manuscript) that tuning apico-basal polarity and junctional recycling orchestrate stem cell emergence complexity is not well supported by the data.

We refined our text when necessary, in particular taking care of transferring and substantiating the arguments that appeared in the Results section, to the Discussion. We also made efforts, on several occasions and for clarity, to describe more precisely the results presented in the different panels of the Figures.

As mentioned in the header of the text of the Public Review and the response to the 6th point of the Public Review of Reviewer 1, we modified slightly the title to avoid ambiguity. In addition, we added a new paragraph to the beginning of our discussion that summarizes the impact of our findings and, we believe, legitimates our title.

**Recommendations for the authors:**

**Reviewer #1 (Recommendations For The Authors):**
(1) Embryonic stages should be indicated in all images presented for clarification.

We thank the reviewer for this point, we added stages when missing on the figures (Figure 1, Figure 1 - Figure supplement 1, Figure 2, Figure 2 - Figure supplement 1, Figure 5, Figure 6, Figure 6 - Figure supplement 1, Figure 7 - Figure supplement 3, Figure 7 - Figure supplement 5, Figure 7 - Figure supplement 6)

(2) In which anatomical site/s were images from Fig 1C and D taken? The surrounding environment looks different, for example, cells in Fig1D seem to be surrounded by other cells, resembling the endothelial plexus at the CHT, while the cells in Fig. 1C seem to be in the dorsal aorta. Is there a spatial difference depending on where cells are budding off? The authors state that there are no differences, but no quantification or data demonstrating that statement is provided.

As mentioned in the figure legend (lines 1206-1209 of the original manuscript), images for Figure 1C and 1D were both taken at the boundary between the end of the AGM and the entry in the caudal hematopoietic tissue. As the images were acquired from different embryos, the labelling of the underlying vein differs between the two panels, with veinous tissues being more sparsely labelled in panel C than in panel D. These images were chosen to illustrate the clearly opposite morphology between the two EHT types that we describe. However, for the rest of the paper, all images and all analysis were exclusively acquired / performed in the dorsal aorta in the AGM, in a region spanning over approximately 10-12 inter-segmentary vessels, starting from the end of the elongated yolk up to the start of the balled yolk. In light of the work from the lab of Zilong Wen showing that only cells emerging anteriorly exhibit long-term replenishment potential (Tian et al. 2017), we specifically chose to limit our comparative analysis to the AGM region and did not quantitatively investigate emergences occurring in the caudal region of the aorta. Additionally, although we routinely observe both types of emergences occurring in the caudal region of the dorsal aorta, we did not quantify the frequency of either EHT events in this region.

Finally, the EHT pol+ cells that we show Figure 1C are of the highest quality obtained ever; one reason is that these two cells emerge at the entry of the CHT which is a region a lot easier to image at high resolution in comparison to the trunk because the sample is less thick and because we are less perturbed by heart beats.

(3) Which figure shows "EHT pol- cells were observed in all other Tg fish lines that we are routinely imaging, including the Tg(Kdrl:Gal4;UAS:RFP) parental line that was used for transgenesis, thus excluding the possibility that these cells result from an artefact due to the expression of a deleted form of Podxl2 and/or to its overexpression."? It would be informative to include this figure.

Other examples of EHT pol- cells were shown Figure 5C as well as Figure 6B using the Tg(kdrl:Jam3b-eGFP; kdrl:nls-mKate2) fish line, that was routinely used for junctional dynamic analyses by FRAP. Furthermore, we add now a new figure (New Figure 1 – figure supplement 3), to illustrate the presence of EHT pol- cells using the Tg(CD41:eGFP) transgenic background, additionally illustrating that EHT pol- cells are CD41 positive.

(4) Are the spinning disk confocal images a single plane? Or maximum projections? Sometimes this is not specified.

We made sure to take into account this remark and went through all figures legends to specify the type of images presented (Figure 1 – figure supplement 1, Figure 2, Figure 2 – figure supplement 1, Figure 2 – figure supplement 2, Figure 7 – figure supplement 3) and also, when relevant, we added this information directly to the figure panels (Figure 6A – 6B).

(5) Could the expression data by RT-qPCR for the Pard3 isoforms be shown? Additionally, it would be appreciated if this expression data could be complemented using Daniocell (https://daniocell.nichd.nih.gov/).

As mentioned in the first paragraph of our response to Public Reviews, and based on reviewers’ comments, we revised our strategy for the investigation of pard3 proteins expression in the vascular system, for their potential role in EHT and sensitivity to runx1. First, we used FACS sorting as well as tissue dissection to enrich in aortic endothelial cells and perform our qPCR analyses (see the new Figure 4 – figure supplement 1A and Figure 4 – figure supplement 3A for the strategy). As asked by the reviewers and for more transparency, we show the expression relative to the housekeeping gene ef1a in our different control samples (new Figure 4 – figure supplement 1C). Furthermore, we used single-molecule FISH to precisely characterise in situ the expression of several of the Pard3 isoforms (Pard3aa, Pard3ab and Pard3ba, which, based on qPCR, were the most relevant for our investigation in the vascular system) (see lines 386 to 412 in text relative to Figure 4 – figure supplement 2). This new addition nicely shows the different pattern of expression of 3 of the Pard3 zebrafish isoforms in the trunk of 2dpf embryos, outlining interesting specificities of each isoform expression in different tissues.

We thank the reviewer for this suggestion to complement our data with the published Daniocell dataset. However, and potentially due to the poor annotation of the different pard3 genes on public databases, gene expression information was absent for two of our isoforms of interest (pard3aa and pard3ba), that we ultimately show to be the most enriched in the vascular system in the trunk. Daniocell gene expression data for the Pard3ab isoform at 48hpf show expression in pronephric duct at 48-58hpf, as well as in intestine progenitors and neuronal progenitors, which is consistent with our in situ observations using RNAscope. However, pard3ab is poorly detected within the hematopoietic and vascular clusters. This observation is coherent with our data that do not show any enrichment of this isoform in vascular tissues compared to other structures. On the other hand, pard3bb does not seem to be particularly enriched in vascular/hematopoietic clusters at 48-58hpf in the Daniocell dataset, in accordance to what we observe with our qPCR. Finally, in the Daniocell dataset, all of the pard3 variants (pard3ab, pard3bb, PARD3 and PARD3 (1 of many)) seem to be either scarcely or not detected in the hematopoietic/vascular system. In our case, for all the isoforms we studied in control condition (pard3aa, pard3ab and pard3ba), and although the technic is only semi-quantitative due to the presence of an amplification step, RNAscope assays seem to indicate a very low expression in aortic cell with sometime as little as one mRNA copy per cell; this explains low detection in single-cell RNAseq datasets and is coherent with the Daniocell dataset.

(6) It would be informative to add in the introduction some information on apico-basal polarity, tight junctions, JAMs (ArhGEF11/PDZ-RhoGEF).

We modified the introduction so as to add relevant information on Pard3 proteins, their link with our JAMs reporters in the context of polarity establishment, as well as the role of ArhGEF11/PDZ-RhoGEF and its alternative splicing variants in regulating junctional integrity in the context of epithelial-to-mesenchymal transition (lines 99 to 127). This modification of the introduction also allowed us to lighten some parts of the result section (lines 222 to 224, 345 to 349 and 454 to 456 of the original manuscript).

**Reviewer #2 (Recommendations For The Authors):**
(1) There is lots of data (and lots of work) in this paper; I feel that the pard3 data doesn't substantially add to the paper, and at the same time there is data missing (see point 10, point 11 below for an example).

To add to the clarity and substantiate our findings on Pard3, we revised entirely our investigation strategy as mentioned in previous paragraphs. We refined the characterization of Pard3 isoforms expression in the vascular tissue, using both cell enrichment by FACS for gene expression analysis as well as single-molecule FISH (RNAscope) to access to spatial information on the expression of pard3 isoforms, reaching sub-cellular resolution.

This new strategy allowed us to show the unexpected localization of Pard3ba mRNAs in mRNAs enriched regions in the vicinity of HE/EHT cells (new Figure 4, and paragraph Interfering with Runx1 activity unravels its function in the control of Pard3ba expression and highlights heterogeneous spatial distribution of Pard3ba mRNAs along the aortic axis, see the new manuscript). Overall, the new spatial analysis we performed allowed us to substantiate our findings on Pard3ba and suggests a direct interplay between hemogenic cells and their endothelial aortic neighbors; this interplay supposedly relies on apico-basal polarity features that is at least in part regulated by runx1 in the context of HE maturation and EHT.

(2) Labelling of the figures could be substantially improved. In many instances, the text refers to a figure (e.g. Fig 6A), but it has several panels that are not well annotated (in the case of Fig 6A, four panels) or labelled sparsely in a way that makes it easy to follow the text and identify the correct panel in the figure. Even supplementary figures are sparsely labelled. Labelling to include embryonic stages, which transgenic is being used, etc should be added to the panels to improve clarity for the reader.

We revised the figures to added relevant information, including stages, types of images and annotations to facilitate the comprehension, including Figure 6A – 6B, Figure 5B – 5C (see response to Reviewer 1, first comment, for a more complete list of all revised figures, transgenic fish lines and embryonic stages annotations). Furthermore, we revised the integrality of the manuscript to fit as much as possible to the figures and added some annotations to more easily link the text to the figures and panels.

(3) The current numbering of supplementary figures is quite confusing to follow.

We revised the manuscript so as to make sure all principal and supplementary figures were called in the right order and that supplementary figures appearance was coherent with the unfolding of the text. For Figure 7 only, the majority of the supplemental figures are called before the principal figure, as they relate to our experimental strategy that we comment on before describing the results.

(4) Graphs in Fig 4, Fig 7 supplement 1 and some of the supplementary figures miss statistical info for some comparison (I assume when non-significant), and sometimes present a p-value of a statistical test being done between samples across stages - but these are not dealt with in the text. Throughout all graphs, the font size used in graphs for annotation (labelling of samples, x-axis, and in some cases the p values) is very small and difficult to read.

For Figure 7 - figure supplement 1, non-significant p-values of statistical tests were not displayed (as mentioned in the Figure legend, line 1614 of the original manuscript). For the new Figure 4, all p-values are displayed. For new Figure 4 - figure Supplement 1, statistical tests were only performed to compare RFP+ and RFP- cells in the trunk condition (3 biological replicates) and not in the whole embryo condition, for which we did not perform enough replicates for statistical analysis (biological duplicates).

(5) The results are generally very difficult to follow, with a fair amount of discussion included but then very little detail of the experiments per se.

We thank the reviewers for these comments that helped us improve the clarity of the manuscript.

The Results section was revised to move some of the paragraphs to the introduction (see response to Reviewer 1, 6th comment), and some of them to the Discussion (such as lines 149 to 156 or 410 to 416 in the first version of the manuscript referring to vacuolar structures or to the recycling modes of JAMs in EHT pol+ and EHT pol- cells).

(6) The truncated version of runx1 is introduced but its expected effect is not explained until the discussion. Related to this, is it expected that blocking runx1 with this construct (leading to accumulation of cells in the aorta before they undergo EHT) then leads to increased numbers of T-cell progenitors in the thymus? Abe et al (2005, J Immunol) have used the same strategy to overexpress the runt domain in thymocytes and found a decrease in these cells, rather than an increase. Can you explain this apparent discrepancy?

We thank the reviewer for this interesting point on the effect of runx1 interference. This phenotype (increased number of thymic cells) seems to be in agreement with the phenotype that was described in zebrafish using homozygous runx1 mutants (Sood et al. 2010 PMID: 20154212), in which the authors show an increase of lymphoid progenitors in the kidney marrow of adult runx1W84X/W84X mutants compared to controls as well as a similar number of intra-thymic lck:eGFP cells in mutants and controls. Notably, the T-lymphoid lineage seems to be the only lineage spared by the mutation of runx1. This could suggest that in this case either the T-lymphoid lineage can develop independently of runx1 or that a compensation phenomenon (for example by another protein of the runx family) occurs to rescue the generation of T-lymphocytes.

Although our data shows an impact on T-lymphopoiesis, we do not elucidate the exact mechanism leading to an increased number of thymic cells. In our case, we do not know the half-life of our dt-runx1 protein in newly generated hematopoietic cells when our transgene, expressed under the control of the kdrl vascular promoter, ceases to be produced after emergence. The effect we observe could be direct, due to the presence of our mutant protein after 3 days in thymic cells, or indirect, due to the impact of our mutant on the HE, that could lead to the preferential generation of lymphoid-biased progenitors. Similarly, we do not know whether the cells we observe at this stage in the thymus are generated from long-term HSC or short-term progenitors. Indeed, cell tracing analysis from the lab of Zilong Wen (Tian et al. 2017, see our Ref list) show the simultaneous presence of short-term PBI derived and longterm AGM derived thymic cells at 5dpf. Based on this, we can imagine for example that the sur-numerous cells we observe in the thymus are transient populations that could multiply faster in the absence of definitive populations. Conversely, based on our observation of an accumulation of EHT pol+ events, we can imagine that the EHT pol+ and EHT pol- cells are indeed differentially fated and that EHT pol+ may be biased toward a lymphoid lineage. We also know that at the stage we observe (5dpf), RNAscope assay of runx1 show that a vast majority of thymic cells do not express runx1 (our preliminary data), suggesting that the effect we observe would be an indirect one caused by upstream events rather than by direct interference with the endogenous expression of runx1 in thymic cells.

The article referred to by the reviewer (Sato et al. 2005, PMID: 16177090) investigates on the role of runx1 during TCR selection for thymic cell maturation and shows that runx1 signaling lowers the apoptotic sensitivity of double-positive thymocytes when artificially activated, leading to a reduced number of single-positive thymic cells. Furthermore, this paper references another study from the same lab (Hayashi et al. 2000, PMID: 11120804) that used the same strategy to study the role of runx1 on the positive and negative selection steps of T lymphocytes maturation. This paper, although showing that runx1 is important for later stages of T lymphocytes differentiation — the double-positive to single-positive stage maturation —, also shows a relative increase in the amount of double-negative and double-positive thymocytes, that could be coherent with our observations. Indeed, in our case, although we show an increased number of thymic cells, we do not know the relative proportion of the different thymocyte subsets. We could explain the increased number of thymic cells by increased number of DN/DP thymocytes that would not preclude a decrease in single-positive thymocytes. Finally, the cells we observe in the thymus of our dt-runx1 mutants may also be different lymphoid populations, namely ILCs, that would react differently to runx1 interference.

(7) Lines 154-155 refer to aquaporins but are missing a reference. This is a bit of speculation right in the results section and I struggled to understand what the point of it was.

To clarify the argument and ease the flow of the text, as suggested by the reviewers, we transferred this paragraph (lines 149 to 156 of the initial manuscript) to the Discussion section lines 763-789. We additionally made sure to add the missing reference (Sato et al. 2023, see our Ref list).

(8) Lines 173-175, indicating that both EHTpol+ and pol- express the CD41 transgenic marker - would be useful to show this data.

We provide a new supplement Figure (Figure 1 – figure supplement 3), where, using an outcross of the CD41:eGFP and kdrl:mKate2-podxl2 transgenic lines, we show unambiguously and for multiple cells that both polarized EHT pol+ cells and non-polarized EHT pol- cells are CD41 positive. In addition, but not commented on in the main text, we can also see that an HE cell, characterized by its elongated morphology (in the middle of the field), its thickened nucleus and its position on the aortic floor, is also CD41 positive.

(9) Lines 181-201 - it's not clear how HE cells were identified in the first place - was it just morphology? Or were they identified retrospectively?

HE cells were identified solely on morphology and spatial criteria (as mentioned in the Methods section, lines 1073-1082 and 1108-1111 of the first manuscript). Furthermore, a recent investigation by the lab of Zilong Wen (Zhao et al. 2022, see our Ref list) questioning the common origin of HE cells and of endothelial cells as well as their respective capacity to extrude from the aorta to generate hematopoietic cells showed, by single-cell tracing, that 96% of floor cells are indeed hemogenic endothelial cells. Furthermore, as mentioned in the response to the 8th point, we show in Figure 1 – figure supplement 3 that all floor cells express CD41. Finally, we also used an alternative method to validate the true hemogenic identity of aortic floor cells and show, using RNAscope, that virtually 100% of floor cells that we consider as typical HE cells are indeed expressing an hematopoietic transcription factor upstream of Runx1, namely Gata2b (see Author response image 1).

**Author response image 1. sa2fig1:** All cells from the aortic floor, at 48hpf, express the hematopoietic marker Gata2b. 48 hpf Tg(Kdrl:eGFP) fixed embryos were used for RNAscope using a probe designed to detect Gata2b mRNAs. Subsequently, images were taken using spinning disk confocal microscopy. The image in the top panel is a z-projection of the entire aortic volume of one embryo and shows the full portion of the dorsal aorta from the anterior part (left side, at the limit of the balled yolk) down to the urogenital orifice (UGO, right side). The 4 boxes (1 - 4) delineate regions that have been magnified beneath (2X). The 2X images corresponding to each box are z-projections (top views) or z-sections (bottom views). The bottom views allow to visualize the aortic floor and to mark its position on top views. Pink arrows point at HE cells (elongated in the anteroposterior direction) and at EHT cells (ovoid/round cells; EHT pol+ cell morphology is not preserved after fixation and RNAscope; thus, it cannot be distinguished from ovoid/round EHT pol- cells). Pink dots = RNAscope spots of various sizes. The green cells in the subaortic space that are marked by RNAscope spots are newly born hematopoietic stem and progenitor cells (see for example box 1). This embryo is representative of n = 5 embryos treated and imaged.

(1) Line 276 - the difference between the egfp-podxl2 and mKate-podxl2 - could that be due to the fluorophore used? Also, it would be good to label Fig 3 supplement 2 better and to see a control alongside the runt overexpression.

Line 276 does not point at a difference in control conditions between eGFP-podxl2 and mKatepodxl2 (see in new Figure 1 – figure supplement 3, Figure 2 or in new Figure 3 - figure supplement 2 several examples of non-polarized HE cells in control conditions using both fluorophores) but between control and dt-runx1 conditions, both expressing the mKate2podxl2 transgene. Similarly, the new example that we provide now in the CD41 figure (Figure 1 – figure supplement 3) clearly shows that mKate-podxl2 is enriched at the apical/luminal membrane of EHT pol+ cells while no such enrichment is observed for EHT pol- cells. The Reviewer should be informed that EHT cells are not always the most typical in shape, in particular because cells can be squeezed by underlying tissues and for example the vein; or from the luminal side by flow and tensions on the aortic wall because of heart beat (the more we image up in the trunk, the more difficult the imaging and the stability of cell shape during long time-lapse sequences). To also take into account the reviewer’s comments, we added for the new Figure 3 – figure supplement 2A a control condition next to the dt-runx1 condition.

(2) There is no quantitation data on the number of excess EHT pol+ cells in the DA, or in the thymus data (Figs 3 Supp1 and Fig 3 Supp 3). Can you quantify this data? This would better support the claim that tunin apico-basal polarity alters the morphology of the emerging HE cells.

We added quantifications relative to both the emergence process itself, showing the accumulation of HE and EHT pol+ cells (new Figure 3B), and on hematopoiesis per se (new Figure 3 – figure supplement 1). Indeed, we show a diminution in the number of newly generated cmyb+ cells in the sub-aortic space. Furthermore, we improved our quantification of the later phenotype on the thymus (new Figure 3 – figure supplement 3), using improved segmentation methods, that indeed validate the increase number of thymic cells that we described.

(3) The observed changes in pard3 isoforms are just reading out changes in their expression in the runt1 transgenics, rather than demonstrating a role in apico-basal polarity.

We entirely revised our strategy regarding Pard3 expression analyses (see also the text at the beginning of this file, for the Public Review). But we wish to stress on the point that we did not intend initially to show directly a role of Pard3 proteins in controlling apico-basal polarity in the system, we just intended to provide correlative evidence supporting our observations with the polarity marker podxl2 (by interfering with their function, as written in the text, apico-basal polarity - which is essential for aortic lumenization and maintenance -, would have been impaired, blurring interpretations).

During the revision, we obtained the unexpected finding, using RNAscope, that one Pard3 isoform, namely Pard3ba, is the one Pard3 that is expressed non-homogenously along the aortic axis and, in vast majority, by aortic cells and in the direct vicinity of emergence domains of the aortic floor (see the new Figure 4 and Figure 4 – figure supplements 2, 3).

This correlative relation between expression of Pard3ba in aortic endothelial cells neighbouring HE/EHT cells suggests, as we propose, that a cross talk occurs between hemogenic and aortic cells, and that this cross talk relies, at least in part, on the expression of key components of apico-basal polarity and their associated functional features. In addition, we show that junctional recycling differs between both EHT types, based on our observations on the different dynamics in the turnover of JAM molecules, in the two EHT types. As JAM molecules are also required for the recruitment of Pard3, which initiates the establishment of apico-basal polarity, these different dynamics suggest that the control of apico-basal polarity is involved in supporting the morphodynamic complexity of EHT cell types.

(4) There is a Fig 5, Supp 2 that is neither mentioned nor described anywhere in the manuscript.

Figure 5 - figure Supplement 2 is mentioned lines 366-370 of the original manuscript, to describe the initial validation that was performed for our eGFP-JAM constructs in multiple cell types using an ubiquitous heat-shock promoter. We developed our description of this supplemental figure in the new manuscript (lines 504 to 514).

(5) Lines 445-456 - these read like a bit of discussion, not results. There are other similar parts of the results section that also read like a discussion (e.g. 526-533)

Although we decided to keep this paragraph in the Results section, as it justifies the rationale behind the choice of ArhGEF11/PDZ-RhoGEF, we took the reviewers comment into account and, as mentioned in the response to reviewer 1 6th comment, lightened the Results section by transferring some of the paragraphs to the Introduction or Discussion sections.

(6) The description of Fig 7A (from line 505) is missing the stages at which the experiments were performed (also not labelled on the figure).

The stages at which the experiments were performed is stated in the figure legend (line 1366) as well as in the Methods section of the original manuscript (line 1033). We added the information on top of the panels A and B for more clarity.

(7) Some figures have multiple panels (e.g. Fig 7Aa'), so when referred to in the text, it remains unclear which panel is being referred to.

We modified the text so as to refer more clearly to the different panels when mentioned in the text, particularly with regards to Figure 7 and 8 but also for all the other figures.